# Strategies of Method Selection for Fine Scale PM$_{2.5}$ Mapping in an Intra-Urban Area Using Crowdsourced Monitoring

Shan Xu[1], Bin Zou[1], Yan Lin[2], Xiuge Zhao[3], Shenxin Li[1], Chenxia Hu[1]

[1]School of Geosciences and Info-Physics, Central South University, Changsha, Hunan, 410083, China
[2]Department of Geography & Environmental Studies, University of New Mexico, Albuquerque, New Mexico, 87131, United States
[3]Chinese Research Academy of Environmental Sciences, Beijing, 100012, China

*Correspondence to*: Bin Zou (210010@csu.edu.cn)

**Abstract.** Fine particulate matter (PM$_{2.5}$) is of great concern to the public due to its significant risk to human health. Numerous
methods have been developed to estimate spatial PM$_{2.5}$ concentrations in unobserved locations due to the sparse number of fixed monitoring stations. Due to an increase in low-cost sensing for air pollution monitoring, crowdsourced monitoring of exposure control has been gradually introduced into cities. However, the optimal mapping method for conventional sparse fixed measurements may not be suitable for this new high-density monitoring approach. This study presents a crowdsourced sampling campaign and strategies of method selection for hundred metre-scale level PM$_{2.5}$ mapping in an intra-urban area of
China. During this process, PM$_{2.5}$ concentrations were measured by laser air quality monitors through a group of volunteers during two 5-hour periods. Three extensively employed modelling methods (ordinary kriging (OK), land use regression (LUR), and regression kriging (RK) were adopted to evaluate the performance. An interesting finding is that PM$_{2.5}$ concentrations in micro-environments varied in the intra-urban area. These local PM$_{2.5}$ variations can be easily identified by crowdsourced sampling rather than national air quality monitoring stations. The selection of models for fine scale PM$_{2.5}$ concentration
mapping should be adjusted according to the changing sampling and pollution circumstances. During this project, OK interpolation performs best in conditions with non-peak traffic situations during a lightly-polluted period (hold-out validation R$^2$: 0.47–0.82), while the RK modelling can perform better during the heavy-polluted period (0.32–0.68) and in conditions with peak traffic and relatively few sampling sites (less than ~100) during the lightly-polluted period (0.40–0.69). Additionally, the LUR model demonstrates limited ability in estimating PM$_{2.5}$ concentrations on very fine spatial and temporal scales in this
study (0.04–0.55), which challenges the traditional point about the good performance of the LUR model for air pollution mapping. This method selection strategy provides empirical evidence for the best method selection for PM$_{2.5}$ mapping using crowdsourced monitoring, and this provides a promising way to reduce the exposure risks for individuals in their daily life.

## 1 Introduction

Fine particulate matter (PM$_{2.5}$) has been associated with an increased risk of morbidity and mortality in both the long-term and
the short-term (Beverland et al., 2012; Cohen et al., 2017; Di et al., 2017; Lelieveld et al., 2017). The persistent cumulative

effects from exposure in daily activities, especially daily travelling, are critical (Kingham et al., 2013; Hankey et al., 2017). If individuals could consciously choose the location and time of their outdoor activities based on detailed knowledge about the spatiotemporal variation in $PM_{2.5}$ concentration, then their health protection could be improved.

In situ measurement is the most reliable way to capture the $PM_{2.5}$ concentrations across every corner of a city in real time.

However, fixed monitoring stations in conventional air quality monitoring networks are sparse. As a result, site-based observations encounter challenges in capturing spatiotemporal variations of air pollutants, especially in intra-urban areas with unevenly distributed emission sources and dispersion conditions (Kumar et al., 2015; Zou et al., 2016; Apte et al. 2017). Spatial mapping methods, including air dispersion modelling, spatial interpolation, satellite remote sensing (RS), and empirical models, have been increasingly employed to estimate concentrations of $PM_{2.5}$ in unobserved locations over the past two decades (Jerrett

et al., 2005; Henderson et al., 2007; El-Harbawi, 2013; Kim et al., 2014; Rice et al., 2015; Fang et al., 2016; Zou et al., 2017; Zhai et al., 2018; Xu et al., 2018; Liu et al., 2018). The outputs of a dispersion model considerably depend on detailed emission inventories and meteorological information, which are not usually available for many cities. The coarse spatial resolution ($\geq$1-10 km) of satellite instruments and the data missing problem due to the cloud cover prohibit the widespread use of RS in $PM_{2.5}$ concentration mapping in urban environments (Zou et al., 2015; Apte et al., 2017).

Conversely, geostatistical and empirical models can estimate concentrations at high spatial resolution with a rather low requirement for data. The most commonly employed models are ordinary kriging (OK) interpolation and land use regression (LUR) modelling. Some studies have improved the estimating accuracy by combining these two technologies (Mercer et al., 2011; De Hoogh et al., 2018). While they have been successfully applied to map the spatial variability of $PM_{2.5}$ concentrations in various geographic areas, their accuracy varies as the concentration levels and sample sizes change (Wang et al., 2012;

Mercer et al., 2011; Lee et al., 2014; Zou et al. 2015; Gillespie et al., 2016; Choi et al., 2017; De Hoogh et al., 2018).

Due to an increase in low-cost sensing for air pollution monitoring, the real-time strategies for exposure control in cities have been further developed (Kumar et al., 2015). Crowdsourced monitoring that enables citizens to produce geospatial data is constantly growing and shows considerable potential (Heipke, 2010). Large and diverse groups of people who lack formal training can easily describe their environments with a mobile phone or smart phone and upload data via informal social

networks and web technology. Unlike traditional fixed monitoring stations that are usually mounted on roofs (i.e., 3 to 20 metres above the ground) for the sake of instrument protection, crowdsourced monitoring provides real-time $PM_{2.5}$ monitoring that reflects the real exposure for individuals who live and work on the ground. Although crowdsourced monitoring tends to produce observations with questionable quality, it enables us to obtain measurements of ambient air pollution in dense networks at relatively low cost. Some studies have employed these data to display the air pollution concentration and

investigate the exposure risks (Thompson, 2016; Miskell et al., 2017; Jerrett et al., 2017). These observations are still point measurements that are only representative of the limited area around the site and cannot satisfy the demand of obtaining the air pollution concentration whenever and wherever we want.

One way to address the previously mentioned challenge is to combine high-density crowdsourced observations with spatial mapping methods. An important investigation was performed by Schneider et al. (2017) in Oslo, Norway. They presented a

universal kriging technique for urban $NO_2$ concentration mapping that combines near-real-time crowdsourced observations of urban air quality with output from an air pollution dispersion model. However, high-density crowdsourced measurements may vary among urban microenvironments with different human daily activities and among sparsely distributed conventional in situ measurements. Using the elected mapping methods from previous studies to depict the variation in air pollution on a very fine spatial and temporal scale with new monitoring ways may cause the misclassification of exposure and an underestimation of risk. As the number of valid crowdsourced observations may significantly change due to instrument faults, human error, and other quality issues, the applicability of mapping methods to different sampling sizes needs sound scientific evidence.

In this study, we presented strategies of method selection for $PM_{2.5}$ concentration mapping based on crowdsourced datasets with varying size. The intra-urban crowdsourced sampling campaign was conducted in the city of Changsha, China, over two periods in different pollution scenarios. The performance of OK, LUR and regression kriging (RK) in estimating $PM_{2.5}$ pollution was evaluated and compared with an increasing number of training sites. The best performing method was employed to plot the variation in the hourly $PM_{2.5}$ concentration and identify the pollution hotspots in the intra-urban area. The results from this study will provide evidence for the method selection of $PM_{2.5}$ mapping using crowdsourced monitoring and significantly contribute to efficient air pollution mapping and exposure assessment in intra-urban areas.

## 2 Data and methods

### 2.1 $PM_{2.5}$ sampling

#### 2.1.1 Measurement instrument

The portable laser air quality monitor SDL307 (produced by NOVA FITNESS Co., Ltd.) is employed to perform sampling. The monitor manual can be downloaded from http://www.inovafitness.com/index.html. This monitor can be conveniently carried with a total size of $7.3 \times 7.3 \times 2.0$ cm (Fig. 1a) and has a resolution of 0.1 μg m$^{-3}$. It has two monitoring modes, Mode One and Mode Two. Under the Mode One, the monitor observes $PM_{2.5}$ concentration in real time and measurement updates and automatically stores every second, but only has a battery life of 5 hours; that is, there are 60-minutes, a total of 3600-seconds of measurements if we continuously observe $PM_{2.5}$ for one hour in Mode One. Under the Mode Two, the monitor repeats the procedure of observing and storing every second for 1 minute and then sleeping 5 minutes (i.e. 10-minutes, a total of 600-seconds of measurements per hour), and the battery life is 30 hours. According to the test report provided by the Center for Building Environment Test at Tsinghua University, the laser monitors were compared with the regulatory monitor in the lab in the air quality monitoring mode at concentration levels of 21, 63, 106, 212, 454 and 1060 μg m$^{-3}$ and the relative errors were rather low (within ±20%) and demonstrated similar patterns between concentration levels except for the 1060 μg m$^{-3}$ level (http://www.inovafitness.com/a/minyongchanpin/jianceyilei/2015/0522/31.html). The concentration of particulate matter is measured using the light-scattering method (Fig. 1b). The monitor contains a special laser module, and the signals are recorded by a photoelectric receptor when particulate matter passes through laser light. The count and size of particulate

matter are then analysed by a microcomputer after the signals are amplified and converted. Their mass concentrations are automatically calculated using a built-in algorithm based on the conversion factor between the light-scattering method and the tapered element oscillating microbalance technology.

To ensure the low inter-sensor variability (i.e. the measurement difference under the same condition) of sampling monitors, we placed 115 laser air quality monitors in the same environment and continuously observed them in the Mode Two for one week during each of the four seasons. If the absolute value of relative error between the observation of one monitor and the average observations of the other monitors exceeded 5%, this monitor fell into disuse. This procedure was conducted both indoors and outdoors. Subsequently, 86 monitors with rather stable performance and a small difference between each observation remained. In addition, we randomly selected 30 portable laser air quality monitors to compare with the national monitoring instruments to further guarantee the reliability of the sampling data. First, for ease of operation, three national air quality monitoring stations were selected. Second, for each station, 10 monitors were observed in the Mode Two (i.e. 600-seconds of measurements per hour) next to the national monitoring instrument (~15 metres above the ground in the study area) from 8:00 to 20:00 on December 20–22, 2015 and from 8:00 to 20:00 on December 29–31, 2015. The hourly $PM_{2.5}$ concentrations of the evaluation periods were the mean values of 10 minutes measurements. The weather on December 20–22 was overcast with patchy drizzle and light rain at times, and the relative humidity (RH) ranged from 77% to 94%, while the weather on December 29–31 was cloudy with some sunshine and a RH that ranged from 38%–67%.

The scatter plots and descriptive statistics of the valid hourly average $PM_{2.5}$ concentrations from the thirty laser air quality monitors and the three national monitoring instruments for December 20–22 and 29–31 were presented in Fig. 1c and Fig. 1d, respectively. The hourly average $PM_{2.5}$ concentrations for two types of instruments generally showed good agreement with a correlation coefficient $R^2$ of 0.89 and 0.90. The root-mean-square-errors (RMSE) for the former time period was lower than the RMSE for the latter time period (5.63 µg m$^{-3}$ vs. 5.94 µg m$^{-3}$), while the mean relative error (MRE) was higher than the MRE for the latter time period (6.37% vs. 3.82%). The latter time period demonstrated a smaller difference in hourly average $PM_{2.5}$ concentrations between laser air quality monitors and the national monitoring instruments with mean values and standard deviations (SD) of 72.99±16.45 µg m$^{-3}$ vs. 71.89±15.28 µg m$^{-3}$ and 129.93±18.33 µg m$^{-3}$ vs. 129.33±17.50 µg m$^{-3}$. The relationship between RH and the ratio of laser monitor measurements to the national instrument measurements for two evaluation periods was presented in the Supporting Information as Fig. S1a and Fig. S1b. The ratio of laser monitor measurements to the national instrument measurements roughly increased exponentially with the increase in the RH for December 20–22 ($R^2$=0.2186), while the ratio was uncorrelated with RH for December 29–31 ($R^2$=0.0457). When RH correction is made by empirical equation for December 20–22, the $R^2$ between hourly $PM_{2.5}$ concentration from laser monitor measurements and the national instrument measurements improved from 0.89 to 0.90.

### 2.1.2 Sampling design

The sampling area is located in the Changsha metropolitan area (112°49′–113°14′E, 27°58′–28°24′N), which covers an area of approximately 920 km$^2$ and seven districts (refer to Fig. 2). Changsha is the capital of Hunan Province with a population

that exceeds 7 million people. The area experienced high-level exposure to air pollutants due to an increase in anthropogenic activities and intensive energy consumption.

To ensure that the sampling sites exhibit a relatively even and typical distribution for different urban microenvironments (i.e., residential community, building site, school, and park), a series of rules were designed to determine the potential $PM_{2.5}$ sampling sites based on the distribution of potential emission sources (refer to Table 1). The data that support the sampling design consist of important points of interest (POI), dust surfaces, and main road networks. POI data includes industrial parks, enterprises, factories, depots, hospitals, schools, and parks. Dust surfaces refer to natural and artificial bare surfaces with vegetation that covers less than 10%, which easily produce atmospheric particulate matter, such as construction sites, stacked substance, and natural bare land. These data were collected from the Information Center of Land and Resources of Hunan Province. More than three observations of $PM_{2.5}$ concentrations are required every hour for each potential sampling site to improve the reliability of the sampling data. Given that the number of laser air quality monitors and the distance that a volunteer can walk in one hour are limited, only 2–4 sites can be set in the area in which a volunteer can cover during the sampling. Therefore, a total of 208 potential $PM_{2.5}$ sampling sites were selected. The centre of each area covered by a monitor were numbered in sequence (i.e., 1–86). The monitors were also numbered and labelled.

### 2.1.3 Sampling and data processing

Sampling was performed in two time periods in the winter of 2015 to examine the effect of air quality grades on the mapping results. The first period fell between 8:00 and 12:00 on December 24. In this period, the official air pollution levels were "Good" and "Moderate" (i.e., Period 1, lightly-polluted period). The weather was overcast with occasional rain or drizzle, and the relative humidity (RH) ranged from 95% to 98%. The second period extended between 14:00 and 18:00 on December 25, when an orange warning signal of haze (i.e., official air pollution level was "Heavily Polluted") was released by the Changsha Meteorology Bureau (i.e., Period 2, heavy-polluted period). The weather was cloudy with some sunshine, and the RH ranged from 39%–43%.

Before sampling started, every volunteer received one monitor and went to the corresponding area. At each potential monitoring site, the volunteer lifted the monitor (~2 metres above the ground) and constantly measured $PM_{2.5}$ concentration in the Mode One for 3 minutes. During this process, after the first 60 seconds, we observed the screen and uploaded the measurement using a smart phone application (App) that we developed to verify the reliability of the stored data. For each hour, we eliminated the sampling sites observed less than three times. As the sites take turns measuring $PM_{2.5}$ concentration, there are at least 3 minutes of measurements every 20 minutes of every sampling hour for those left sites. The valid observations for every sampling hour (i.e. 9–12 minutes, a total of 540–720 seconds measurements per hour) were then averaged at each site. The geographic coordinates of the sampling sites were also uploaded. As some volunteers quit after the sampling of the first period, the sampling sites in period 2 were concentrated in the central study area. A total of 179-208 samples were successfully collected at each hour in Period 1, and 105-118 samples were successfully collected in Period 2. The official

observations at 10 national monitoring stations in the study area were also obtained (China Environmental Monitoring Center, CEMC: http://106.37.208.233:20035/) and averaged for comparison purposes.

## 2.2 Mapping method selection

We divided sampling data into a training set and a validation set (hold-out validation) for each hour to evaluate the performance
of OK, LUR and RK with an increasing number of training sites. The training data sets were divided into groups based on the percentages of 20%, 30%, 40%, 50%, 60%, 70%, 80%, and 90% of the total number of monitoring sites. Therefore, a series of groups of training samples (36–42, 54–62, 72–83, 90–104, 107–125, 125–146, 143–166, 161–189 sites in Period 1 and 31–35, 42–47, 52–59, 63–71, 73–83, 84–94, 94–106 in Period 2) were extracted using the Subset Features Tool of ArcGIS (version 10.0). We repeated this process 100 times for each training set size for Period 1 and 50 times for Period 2. Statistics including
the coefficient $R^2$, RMSE and MRE between the predicted concentrations and observed concentrations of $PM_{2.5}$ in the independent validation set were employed to evaluate and compare their performance.

### 2.2.1 Ordinary kriging

OK estimates the target variable at an unsampled location as a linear combination of neighbouring observations. OK relies on a weighting scheme, where closer observations have a greater impact on the final prediction. The weighting scheme is dictated
by the variogram (Pang et al., 2010; Zou et al., 2015) and can be described as

$$\left. \begin{array}{l} Z^*(X_0) = \sum_i^n \omega_i Z(X_i) \\ \sum_i^n \omega_i = 1 \end{array} \right\}, \tag{1}$$

where $Z^*(X_0)$ is the estimation of an unknown sample point; $Z(X_i)$ and $\omega_i$ are the value of the $i^{th}$ known sample point surrounding the unknown sample point and its corresponding weight, respectively; and n is the number of known sample points.

### 2.2.2 Land use regression

LUR modelling predicts the air pollution concentration by linking measurements of monitoring sites and geographic elements around them using the least squares method. LUR is composed of predictor variable extraction and selection and regression modelling and validation.

Geographic factors including pollution sources (dust surface and pollution industries), road networks, and land use/cover were employed to indirectly characterise the $PM_{2.5}$ emissions in this study. These data were generated using multiple ring buffers
with different radii (50–1000 m) at each monitoring site. Meteorological data including wind speed, atmospheric pressure, relative humidity, and temperature of 107 sites in and around the sampling area, which may affect the dispersion of $PM_{2.5}$, were also obtained. Geographic factors were made available by the Information Centre of Department of Land and Resources of Hunan Province. Meteorological data were released by the Hunan Meteorology Bureau. All variables (Table 2) were

extracted using ArcGIS (version 10.0). The optimal buffer radius for the percentage of dust surfaces and land use, pollution industries density, and road density were defined based on the maximum Pearson correlation coefficients.

An automatic forward-backward stepwise regression procedure was employed to select the best fitting LUR models based on the screened-out predictors. The final LUR models in this study were determined based on the criteria of the lowest Akaike information criterion (AIC) value and the highest fitting $R^2$. The model structure can be expressed as

$$PM_{2.5,s} = a_0 + a_1 X_{1,s} + a_2 X_{2,s} + \cdots + a_n X_{n,s} + \mu, \tag{2}$$

where $PM_{2.5,s}$ is the estimation of the hourly averaged PM$_{2.5}$ concentration of site s, $X_{i,s}$ (i=1,2,$\cdots$,n) are independent variables, $a_0$ is a constant, $a_i$ (i=1,2,$\cdots$,n) are regression coefficients, and μ is the random error estimated using the least squares method. This process was conducted in R statistical software (version 3.3.2) (Fox and Weisberg 2011, R Core Team 2016).

### 2.2.3 Regression kriging

RK is a two-stage statistical procedure in this study. First, separate standard LUR models were developed based on crowdsourced observations in the training dataset for each hour. Second, the residuals for the LUR models was calculated and interpolated for each hour using OK technology. Finally, the estimations of the residuals at the validation sites were extracted and added to the LUR estimations.

In this study, OK was performed using the Geostatistical Analyst Tool of ArcGIS (version 10.0), and interpolated residuals were obtained using the Extract Values to Point Tool. The entire process was implemented with Python scripts.

### 2.3 PM$_{2.5}$ concentration mapping

The method that performed best with 90% training sites was chosen as the mapping method. Using this method, the spatial distributions of the PM$_{2.5}$ concentration for each hour were estimated with all samples. In this study, nearest neighbour distances between two sampling sites ranged from 15 to 60 metres for Period 1 and 54 to 98 metres for Period 2. Considering the resolutions of the potential predictors, 100 metres was used as the mapping grid size. The spatial distributions of the PM$_{2.5}$ concentration for each hour with measurements of 10 national monitoring stations were estimated using the same method for comparison.

### 3 Results

### 3.1 Descriptive statistics of PM$_{2.5}$ concentrations

Table 3 shows the descriptive statistics of hourly PM$_{2.5}$ concentrations for the crowdsourced sampling sites and the national monitoring stations. For Period 1, the mean values and SD of the PM$_{2.5}$ concentrations for the crowdsourced sampling sites ranged from (69.67±18.81) to (76.45±14.55) μg m$^{-3}$. These values were substantially higher than those for the national monitoring stations (i.e., (36.9±10.97) – (41.2±8.68) μg m$^{-3}$). The maximum and minimum values of crowdsourced PM$_{2.5}$

concentrations were higher than the national values. However, the mean values and SD of PM$_{2.5}$ concentrations of the crowdsourced sites are lower than those of the national stations in period 2. The former values ranged from (162.72±15.96) μg m$^{-3}$ to (171.89±21.5) μg m$^{-3}$, while the latter values ranged from (177.8±16.91) μg m$^{-3}$ to (188.3±22.4) μg m$^{-3}$. Although the minimum values of crowdsourced PM$_{2.5}$ concentrations were also lower than those of the national stations, the maximum values were higher. The average PM$_{2.5}$ concentrations of Period 2 were substantially higher than those of Period 1, and the highest values occurred when traffic and emissions from cooking had peaked (i.e., 12:00 and 18:00) for both periods.

Fig. 3 demonstrates the spatial variation in the PM$_{2.5}$ measurements over the two periods in the study area, and the spatial variations between different sampling sites and two periods can be obtained. For Period 1, the PM$_{2.5}$ concentrations gradually decreased from north to south and from west to east. Higher concentrations of PM$_{2.5}$ (> 75 μg m$^{-3}$) were observed at sampling sites in the northwest corner of the study area. The sampling sites in Changsha County with high levels of green vegetation cover had lower PM$_{2.5}$ concentrations compared with the sites in the inner city. For Period 2, conversely, sampling sites in the central and eastern parts of the study area had higher PM$_{2.5}$ concentrations than those in the western part. Monitoring sites had PM$_{2.5}$ concentrations higher than 150 μg m$^{-3}$ in most areas, with the exception of the western Yuelu district. Particularly, sampling sites in areas along the Xiangjiang River, especially in the higher education mega centre experienced extreme PM$_{2.5}$ pollution (> 210 μg m$^{-3}$).

### 3.2 Model performance for OK, LUR and RK

The box plots of Fig. 4 show the variation in the hold-out validation R$^2$ for the three mapping approaches in relation to the number of training sites. The average and standard deviation of the RMSE and MRE between the observed concentration and predicted concentration of PM$_{2.5}$ in the hold-out validation were presented in the Supporting Information (Table S3–S4). The average values and variability ranges of R$^2$ for OK, LUR and RK were positively associated with an increase in the number of training sites. RK performed best in Period 2 and at 8:00 and 12:00 of Period 1 with training sites less than ~100. The LUR demonstrated the poorest performance for both periods of the models tested.

For Period 1, the PM$_{2.5}$ estimating accuracy was generally highest at 9:00 and lowest at 12:00. The average validation R$^2$ ranges for different number training sites of OK at 8:00, 9:00, 10:00, 11:00 and 12:00 were 0.58–0.72, 0.56–0.78, 0.51–0.82, 0.47–0.71, and 0.24–0.48, respectively. Compared with OK, the accuracy of LUR was substantially lower. The ranges were 0.26–0.55, 0.29–0.54, 0.16–0.40, 0.16–0.36, and 0.24–0.34. The average R$^2$ for RK were weakly smaller than OK at 9:00, 10:00, and 11:00 with ranges of 0. 59–0. 69, 0. 50–0.66, and 0. 48–0.60, respectively. The average R$^2$ of RK at 8:00 and 12:00 were higher than OK when less than ~100 sampling sites were divided into training datasets (8:00: 0.65–0.69 vs. 0.58–0.68; 12:00: 0.40–0.44 vs. 0.24–0.41). For Period 2, the validation R$^2$ from high to low followed the sequence RK > OK > LUR. The average validation R$^2$ for a different number of training sites of OK were considerably lower in Period 1. The ranges at 14:00, 15:00, 16:00, 17:00 and 18:00 were 0.25–0.49, 0.34–0.50, 0.40–0.59, 0.27–0.39, and 0.18–0.27, respectively. The average R$^2$ of LUR were even lower; the lowest values were 0.08, 0.07, 0.15, 0.06, and 0.04, and the highest values were 0.22,

0.25, 0.42, 0.22, and 0.16, respectively. Combining OK and LUR, the performance of RK improved with an average $R^2$ that ranged from 0.43, 0.44, 0.43, 0.36, and 0.32 to 0.60, 0.68, 0.52, 0.54, and 0.57.

Fig. 5 shows scatterplots of holdout-validation results with 90% training sites. For Period 1, the lowest total $R^2$ of OK and the highest total $R^2$ of OK were 0.46 for 12:00 and 0.82 for 10:00 (Fig. 5a), respectively, while $R^2$ of RK were lower with the range of 0.44–0.68 (Fig. 5c); they were both higher than the LUR (0.29–0.53, Fig. 5b). Correspondingly, the RMSE and MRE from low to high were OK (5.95–10.36; 6.80%–9.91%) < RK (8.23–10.92; 9.80%–11.91%) < LUR (10.68–13.16; 12.91%–14.97%). For Period 2, however, the RK presented the highest accuracy with a $R^2$ that ranged from 0.45 (17:00) to 0.66 (14:00) (Fig. 5f). The OK ranked second ($R^2$: 0.27–0.54, Fig. 5d), while the LUR achieved the poorest performance ($R^2$: 0.06–0.36, Fig. 5e).

**3.3 Spatial patterns of crowdsourced PM$_{2.5}$ concentration**

Fig. 6a and Fig. 6b reveal the spatial distributions of OK interpreted PM$_{2.5}$ concentrations for Period 1 from the crowdsourced sampling sites and the national monitoring stations, respectively. Fig. 6c and Fig. 6d demonstrate the spatial distributions of the RK estimated PM$_{2.5}$ concentrations for Period 2. The crowdsourced hourly PM$_{2.5}$ concentration maps demonstrate more detailed intra-urban variations than the national monitoring maps, especially for Period 1.

For Period 1, crowdsourced PM$_{2.5}$ concentrations generally increased from south-east to north-west with multiple hot spots. In the central and south regions of the study area, areas with a larger number of factories that experience a relatively higher PM$_{2.5}$ concentration than other areas. The national monitoring PM$_{2.5}$ concentrations, however, were less than 55 μg m$^{-3}$ with limited spatial variation. For Period 2, with the exception of 14:00, the national monitoring PM$_{2.5}$ concentration maps showed high-east and low-west patterns. PM$_{2.5}$ concentrations of central Yuelu district were rather low (<175 μg m$^{-3}$). Crowdsourced PM$_{2.5}$ concentrations demonstrate extensive cold spots of PM$_{2.5}$ concentrations in southern Changsha County and the southern Kaifu district, while southern Yuelu and western Tianxin with a high-density of factories and roads were hot spots of PM$_{2.5}$ concentration.

**4 Discussion**

Aimed at efficiently mapping the PM$_{2.5}$ concentration in an intra-urban area at a fine scale using crowdsourced monitoring, a high-density crowdsourced sampling campaign and strategies of the popular mapping method selection with an increase in training sites were presented in China for the first time.

The number of sampling sites were 18 and 10 per 100 km$^2$ for Period 1 and Period 2, respectively. These data comprise a considerable improvement compared with a density of approximately 0.015 sites per 100 km$^2$ in the national air quality monitoring network in China. As expected, crowdsourced PM$_{2.5}$ measurements demonstrated detailed spatial variation among urban microenvironments, and these variations can hardly be disclosed by sparse national air quality monitoring stations. This finding suggests that crowdsourced sampling can effectively improve the density of PM$_{2.5}$ monitoring at a rather low monetary

cost and can be supportive of the short-term air pollution exposure assessment for epidemiologic studies at a fine scale. However, persuading the general public to continuously observe and upload $PM_{2.5}$ concentrations during their activities of daily living through a designed study is difficult. We employed a batch of volunteers to model their behaviours on the general public's behaviour and simultaneously collect data. This approach is a preliminary practice of crowdsourced monitoring and can be further developed and improved in the long-term exposure assessment at the fine scale in the future with the progress in low-cost wearable air quality monitors and automatic processing techniques of crowdsourced data.

During the sampling, as the number and the battery life of the monitors are limited, we cannot continuously observe $PM_{2.5}$ concentration at one site for the whole sampling period. For all potential sites in one area that a volunteer can cover, 27–36 minutes are needed to walk through three times. That means 24–33 minutes are left in one hour to observe 2–4 potential sites three times. Therefore, at each potential monitoring site, 3 minutes are designed for $PM_{2.5}$ concentration constantly monitoring every observation (i.e. Mode One). Although the hourly sampling concentrations are only averages from 9-12-minutes, a total of 540–720 seconds of measurements, we believe it is generally equal to the hourly averages for the following reasons: (1) the 3–4 times of 3-minutes observation happened at rather regular intervals, which can reflect the temporal variation of $PM_{2.5}$ measurements for one location during one hour to a certain extent. (2) there are 10-minutes, a total of 600-seconds of measurements per hour for each monitor during the comparison experiments with the national monitoring instruments; the numbers of measurements per hour for crowdsourced campaign and comparison experiments are close and averages from the 600-seconds of measurements during the comparison experiments demonstrate a relatively good agreement with the national hourly $PM_{2.5}$ observations ($R^2$: 0.89–0.90), which provides empirical evidence for this assumption. (3) several studies have employed sampling periods of 10-minutes level to measure the within-day variability of PM concentrations (Godri et al., 2010; Griffiths et al., 2018) and few studies have proven that the performance characteristics of some low-cost sensors on 1 min and 1 h scales were rather similar (Zheng et al., 2018). However, it needs to be noted that this assumption was proposed under the circumstance that the weather condition and emission sources of the study area demonstrate no extreme variation in short-term. It is inapplicable to some special cases such as a dust storm when the $PM_{2.5}$ can increase tens and even hundreds of times high concentration within a few minutes (Zhang et al., 2010).

The hourly $PM_{2.5}$ concentrations between crowdsourced sampling sites and national monitoring stations were rather different; this difference varied as the official air quality level changed. The crowdsourced $PM_{2.5}$ concentrations were substantially larger than the national concentrations in Period 1 (lightly-polluted) and slightly lower in Period 2 (heavy-polluted). One possible reason is that the national monitoring stations in the study area were installed on the roofs of mid-rise buildings (i.e., ~15 m) with ventilation and spaciousness, while crowdsourced sampling was conducted on the real ground (i.e., ~2 m). The $PM_{2.5}$ concentration may experience a drop above ~15m from the ground (Quang et al., 2012; Sajani et al., 2018). The change in the major pollution sources and meteorological conditions in the study area may contribute to the difference between two periods; the major contribution of local sources, especially the vehicle emission and the very high RH (95%–98%) during the lightly-polluted period, may cause the accumulation of $PM_{2.5}$ near the ground; and the sources of long-range transport of regional pollution during the heavy-polluted period can increase the concentration of $PM_{2.5}$ on the upper layer. Another possible reason

can be the inaccuracies in measurements the low-cost sensor and the use of optical particle detection may cause. For Period 1, the very high RH may lead to an overestimate of $PM_{2.5}$ concentration. The ratio of laser monitor measurements to the national instrument measurements generally increased exponentially with the increase in the RH for December 20–22 (i.e. the first evaluation period with a weather condition similar to that of lightly-polluted sampling period, RH: 77%–94%) ($R^2$=0.2186).

This is consistent with the findings of Zheng et al. (2018) and Badura et al. (2018), who discovered that low-cost sensors tend to overestimate the $PM_{2.5}$ concentrations when RH is high (>~80%). However, the agreement between laser monitor and the national instrument was rather good ($R^2$=0.89) and the improvement after RH correction was insignificant (0.01); the potential effect of RH on hourly $PM_{2.5}$ concentration during very high RH events could be consistent because of the low inter-sensor variability (i.e. the measurement difference under the same condition, <5%) of sampling monitors selected from the

preliminary experiments and the small spatial variability of RH in intra-urban area. In view of the above-mentioned reasons, we believe the hourly $PM_{2.5}$ concentration for lightly-polluted sampling period could generally disclose the air pollution variation of different urban microenvironments although the very high RH conditions (RH>95%) were not experienced in the evaluation period. For Period 2, the high concentration environment may cause the underestimate of low-cost sensor (Zheng et al., 2018; Johnson et al., 2018). Although the monitor experienced little high concentration environments (>160 $\mu g\ m^{-3}$) for

December 29–31 (i.e. the second evaluation period with a weather condition similar to that of heavy-polluted sampling period). the monitor was compared with the regulatory monitor at concentration levels of 106, 212 and 454 $\mu g\ m^{-3}$ in the test of Tsinghua University and the relative errors were rather low (within ±20%) and demonstrated similar patterns between concentration levels. In consideration of this and the low inter-sensor variability of sampling monitors, we assume the responses of sampling monitor to the national monitoring instrument in heavy-polluted sampling period is consistent in the study area and the spatial

variation of air pollution could be revealed to some extent. As the three methods we compared were performed with the same sampling dataset, the uncertainty in measurements associated with the monitor, RH and high concentration environments may cause a limited influence on the method comparison results. We therefore did not correct the measurements in this study. However, more efforts are needed in the intended use environment evaluating, uncertainty analysis and bias correction of low-cost sensors for applications that requires more accurate measurements in the future, such as very high-resolution mapping of

air pollution and accurate exposure assessment, especially under extreme weather conditions and very high and very low concentration environments.

Unlike previous studies that conducted performance comparisons of OK, LUR and RK in estimating air pollution concentration on an annual and seasonal scale based on measurements from sparse regulatory stations (Mercer et al., 2011; Lee et al., 2014; Zou et al. 2015; Choi et al., 2017; De Hoogh et al., 2018), this research is the first study to evaluate and compare their

performance with an increase in the number of training sites at an hourly scale using crowdsourced monitoring.

As expected, the performance of three methods improved with an increase in the number of training sites. Compared with former studies that normally developed in other fields (e.g., spatial variability analysis of soil components in the environmental sciences) (Li and Heap, 2014), this study further confirmed the better performance of OK interpolation with larger training data sets in air pollution estimation. We substantiated the findings of Johnson et al. (2010), who discovered that LUR models

developed with fewer sampling sites may perform poorly using real-ground $PM_{2.5}$ measurements. However, average hold-out validation $R^2$ (0.04–0.55) between the observed concentration and predicted concentration of $PM_{2.5}$ in this study were smaller than the results in Johnson et al. (2010) (0.29–0.67) and similar studies of $NO_2$ presented by Wang et al. (2012) and Gillespie et al. (2016) (0.44–0.85). The variations in the hourly average $PM_{2.5}$ concentration between two sampling sites were generally

sharper compared with the annual average values. The meteorological condition had a more sensitive role in the short-term transmission and diffusion of $PM_{2.5}$ than the long-term processes. These findings suggest that the most effective way to improve the accuracy of the mapping method continues to increase the number of sampling sites and confirm the necessity of developing high-density crowdsourced sampling for $PM_{2.5}$ monitoring. However, the increased variability ranges of $R^2$ and the standard deviation of RMSE and MRE with an increase in the number of training sites also suggest that the performance of these

methods was affected by more than sampling size. The spatial distribution of the samples, for example, may influence their estimating accuracy (Li and Heap 2014).

Contrary to the findings of Zou et al. (2015) and Choi et al. (2017) conduced at the annual scale, OK interpolation surprisingly showed a better performance in estimating the $PM_{2.5}$ concentrations compared with the LUR modelling with a substantially higher average $R^2$ and lower RMSE and MRE. RK also performed better than LUR (0.32–0.71 vs. 0.04–0.55), which is

consistent with the findings of Mercer et al. (2011) (0.67–0.75 vs. 0.48–0.74) and De Hoogh et al. (2018) (0.66 vs. 0.59). RK had the highest accuracy in Period 2 and at 8:00 and 12:00 of Period 1 with less than ~100 training sites. These results suggest that OK interpolation based on crowdsourced sampling is the best strategy for the $PM_{2.5}$ mapping in the intra-urban area when the official air pollution levels are "Good" and "Moderate" for non-peak traffic conditions in this study, while RK is the best strategy when the pollution levels are "Heavy-polluted". These findings challenge the traditional point on the LUR model's

good performance in air pollution mapping and verify that the applicability of mapping methods varies as the monitoring technology and sampling density change. In addition, the accuracy of OK and LUR were distinctly higher for Period 1 (0.24–0.82; 0.13–0.55) than for Period 2 (0.18–0.59; 0.04–0.42), while that for RK was rather stable (0.40–0.71 vs. 0.32–0.68). This finding indicates the robustness and generalisation capability of RK in estimating the $PM_{2.5}$ concentration.

Using the selected mapping method, the spatial distributions of the hourly $PM_{2.5}$ concentration based on crowdsourced

sampling data and national air quality observations were successfully plotted and compared. The former distribution provides more information about the intra-urban $PM_{2.5}$ variations than the latter distribution. The nearest-neighbour distances that range from 15 to 60 m between two crowdsourced sampling sites enable $PM_{2.5}$ concentration mapping to attain the hundred metre-scale level. In the lightly-polluted period, this phenomenon was more pronounced. These findings not only suggest the support of crowdsourced activities in $PM_{2.5}$ monitoring on a fine scale but also prompt us to pay more attention to the scenarios with

low-level air pollution. This outcome is critical to the long-term future of air pollution prevention and control and public health protection for China, since the main emphasis has gradually shifted from the control of heavy pollution to the prevention of exposure risks.

As the crowdsourced $PM_{2.5}$ concentrations maps revealed, areas with a larger number of factories and high-density of roads experienced relatively higher $PM_{2.5}$ concentrations, while areas with high levels of green vegetation cover had lower $PM_{2.5}$

concentrations. The relatively high concentration in the northwest corner of the study area with few factories in Period 1 may be attributed to the dust deposition from construction activities promoted by a high RH in this newly developed zone. This finding suggests that optimising the distribution of land use may improve the air quality to some extent and strengthening the control of local emission may be the primary way to reduce pollution in the lightly-polluted period. As the urban air quality

grade has an important effect on the spatial distribution of samples (spatial autocorrelation, and heterogeneity), which may also be affected by sample size, the mechanism for this influence is somewhat equivocal and needs further research.

## 5 Conclusions

This study presented strategies of method selection for efficient $PM_{2.5}$ concentration mapping with an increasing number of training sites using crowdsourced monitoring. The results confirmed that $PM_{2.5}$ concentrations in microenvironments varied

across the intra-urban area in China's cities. These variations can be clearly disclosed by the crowdsourced $PM_{2.5}$ sampling rather than the national air quality monitoring stations. The selection of models for fine scale $PM_{2.5}$ concentration mapping should be adjusted with changing sampling and pollution circumstances. During this project, ordinary kriging (OK) interpolation performs the best in conditions with non-peak traffic situations in the lightly-polluted period, while regression kriging (RK) can perform better in the heavy-polluted period and conditions with peak traffic and relatively few sampling sites

in the lightly-polluted period. Additionally, note that the land use regression (LUR) model demonstrates a limited ability in estimating $PM_{2.5}$ concentrations at very fine scale in this study. This method selection strategy provides empirical evidence for the method selection of $PM_{2.5}$ mapping using crowdsourced monitoring and a promising way to reduce the exposure risks for individuals in their daily lives.

*Author contribution*. SX performed the experiments and wrote the manuscript text. BZ supervised and designed the research

and helped with the manuscript. YL and XZ helped with the discussion and revisions. SL and CH participated in the data processing.

*Competing interests*. The authors declare that they have no conflicts of interest.

*Acknowledgements*. Special thanks go to the volunteers participated in this crowdsourced sampling campaign. Deepest gratitude goes to the editor and the anonymous reviewers for their careful work and thoughtful suggestions. This study was

supported by the National Key Research and Development Program of China (No. 2016YFC0206201/05), the National Nature Science Foundation of China (No. 41871317), and the Innovation Driven Program of Central South University (No. 2018CX016).

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

**Table 1.** Rules for potential PM$_{2.5}$ sampling sites selection.

| Code | Type | N | Rules |
|---|---|---|---|
| 1 | Vertex point | 5 | U1[a] = {X[c] | X∈(Vertex point of the boundary of sampling area ∩ Landmark)}. |
| 2 | Industrial park | 28 | A2[b] = {X | X∈((Industrial park ∪ (Metal & cement & power industrial factories agglomeration)) – High-tech industrial park)}; <br> U2 = {X | X has the largest number of factories within its 100 m buffer zone AND X∈A2}. |
| 3 | Dust surface | 13 | A3 = {X | X∈(POI ∩ Dust surface) AND area of dust surface ranks in the top 4 of each district}; <br> U3 = {X | Distance between X＞200 m AND X∈A3}. |
| 4 | Depot | 16 | U4 = {X | X∈(Coach station ∩ Railway station)}. |
| 5 | Scenic area | 27 | A5 = {X | X∈((Park – Neighbourhood park) ∩ well-known scenic area)}; <br> U5 = {X | Distance between X＞200 m AND X∈A5}. |
| 6 | Hospital | 11 | A6 = {X | X∈(Hospital ranks in the top 3 of each district ∪ Children's hospital ∪ Respiratory special hospital)}; <br> U6 = {X | Distance between X＞200 m AND X∈A6}. |
| 7 | Residential area | 12 | A7 = {X | Distance between X and U1＜200 m OR Distance between X and U3＜200 m, X∈ Residential area }; <br> U7 = {X | Distance between X＞200 m AND X∈A7}. |
| 8 | School | 15 | U8 = {X | Distance between X and U1＜200 m OR Distance between X and U3＜200 m, X∈School, in order of priority: Kindergarten＞Primary＞Secondary＞Universities)}. |
| 9 | Commercial area | 9 | U9 = {X | X is the building with the highest population density, X∈Commercial area}. |
| 10 | Other important POI | 8 | U10 = {X | X∈(Corresponding sampling site of national monitoring station ∪ Background site ∪ Museum)}. |
| 11 | Road | 56 | A11 = {X | X∈(Junction of (Expressway ∪ Main road))}; <br> U11 = {X | X is 50/100 metres away from A11 OR X∈A11}. |
| 12 | Supplementary point | 3 | U12 = {X | X∈ POI where four neighbouring grids have no site}. |

[a]U$_i$ (i=1, 2, …): ith subset of the set of potential PM$_{2.5}$ sampling sites.
[b]A$_i$ (i=1, 2, …): ith subset of the union of supporting data.
[c]X: element belongs to the set.

**Table 2.** Description of potential predictor variables for LUR.

| GIS dataset | Predictor Variables | Unit | Buffer size (radius in metres) |
|---|---|---|---|
| | Piling surface | % | |
| | Construction surface | % | |
| Dust surfaces | Rolling trample surfaces | % | 50, 100, 200, 300, 500, 1000 |
| | Bare surfaces | % | |
| | Total | % | |
| Pollution industries | Inverse distance to nearest industries | | NA |
| | Industries density | | 50, 100, 200, 300, 500, 1000 |
| | High-density residential area | % | |
| | Low-density residential area | % | |
| | Urban green land | % | |
| Land use | Other built-up area | % | 50, 100, 200, 300, 500, 1000 |
| | High-density forest | % | |
| | Low-density forest | % | |
| | Agricultural land | % | |
| Traffic | Inverse distance to a nearest major road | | NA |
| | Road density | | 50, 100, 200, 300, 500, 1000 |
| | Average wind speed | Meter/s | NA |
| Meteorology | Atmospheric pressure | Pa | NA |
| | Relative humidity | % | NA |
| | Temperature | Fahrenheit degree | NA |

**Table 3.** Descriptive statistics of $PM_{2.5}$ concentration ( $\mu g\ m^{-3}$ ).

| | | Mean | | Max | | Min | | Standard deviation | |
|---|---|---|---|---|---|---|---|---|---|
| | | SAMP[a] | NAT[b] | SAMP | NAT | SAMP | NAT | SAMP | NAT |
| | 8:00 | 69.67 | 39.8 | 128 | 58 | 36 | 27 | 18.81 | 10.46 |
| | 9:00 | 72.97 | 36.9 | 132 | 54 | 30 | 20 | 17.04 | 10.97 |
| Period 1 | 10:00 | 73.08 | 38.5 | 113 | 58 | 28 | 21 | 15.57 | 11.57 |
| | 11:00 | 74.12 | 39.4 | 106 | 54 | 30 | 27 | 13.96 | 8.78 |
| | 12:00 | 76.45 | 41.2 | 136 | 53 | 44 | 29 | 14.55 | 8.68 |
| | 14:00 | 167.91 | 188.3 | 220 | 207 | 145 | 165 | 14.43 | 14.48 |
| | 15:00 | 165.75 | 182 | 227 | 206 | 133 | 153 | 16.68 | 17.06 |
| Period 2 | 16:00 | 162.72 | 178.7 | 212 | 201 | 115 | 149 | 15.96 | 16.91 |
| | 17:00 | 167.69 | 177.8 | 266 | 209 | 136 | 146 | 18.92 | 20.49 |
| | 18:00 | 171.89 | 182.1 | 250 | 219 | 132 | 149 | 21.5 | 22.4 |

a: sampling sites of the crowdsourced sampling campaign.

5   b: national monitoring stations.

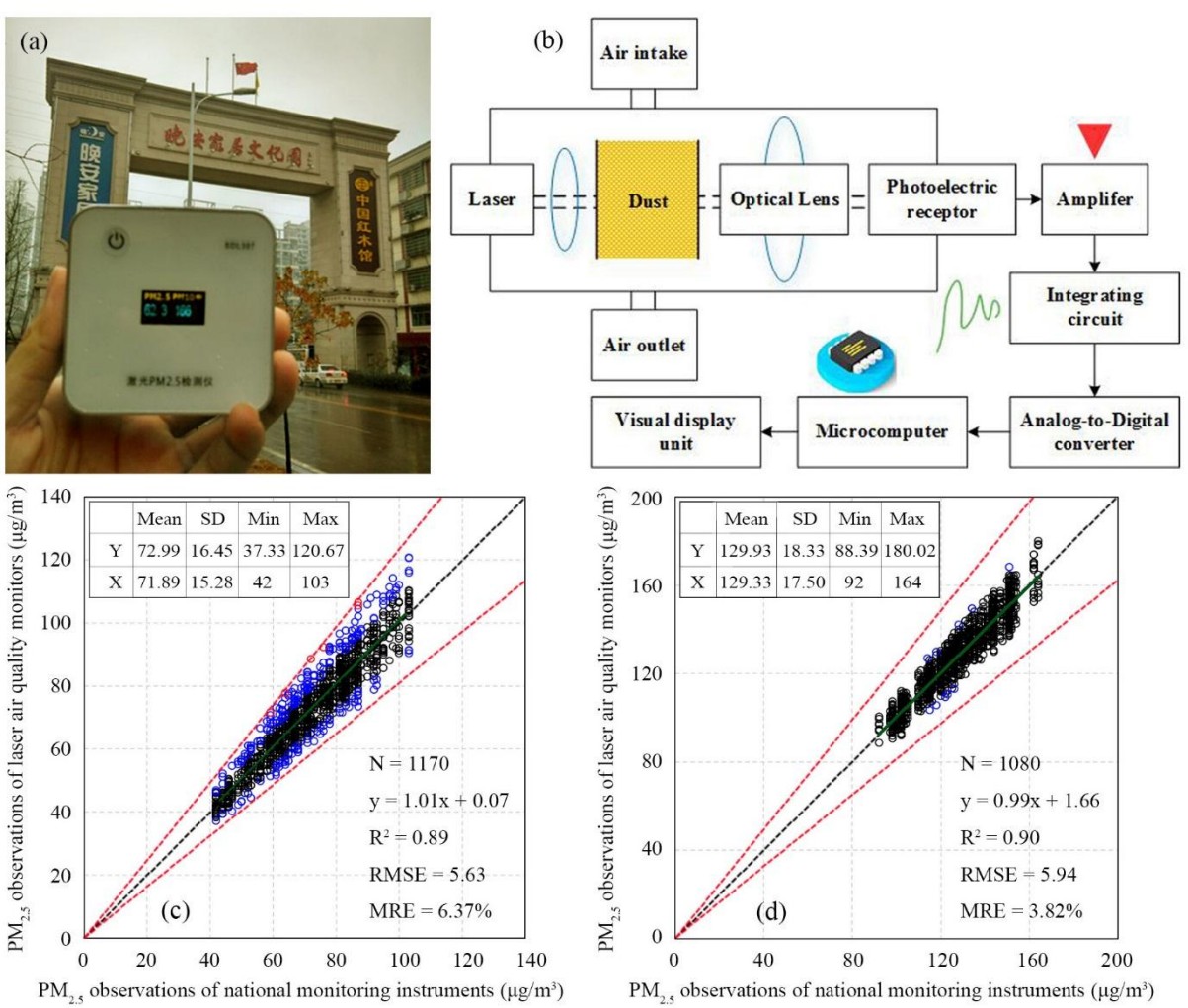

**Figure 1:** Laser air quality monitor: (a) exterior, (b) measuring principle, (c) (d) scatterplots of hourly PM$_{2.5}$ concentration from laser air quality monitor (Y) and national monitoring instrument (X). The black, blue and red dots indicate PM$_{2.5}$ observations with relative error of <10%, 10%−20%, and >20%, respectively. The black and red dotted line are the 1:1 line and 1:1.2 line as references.

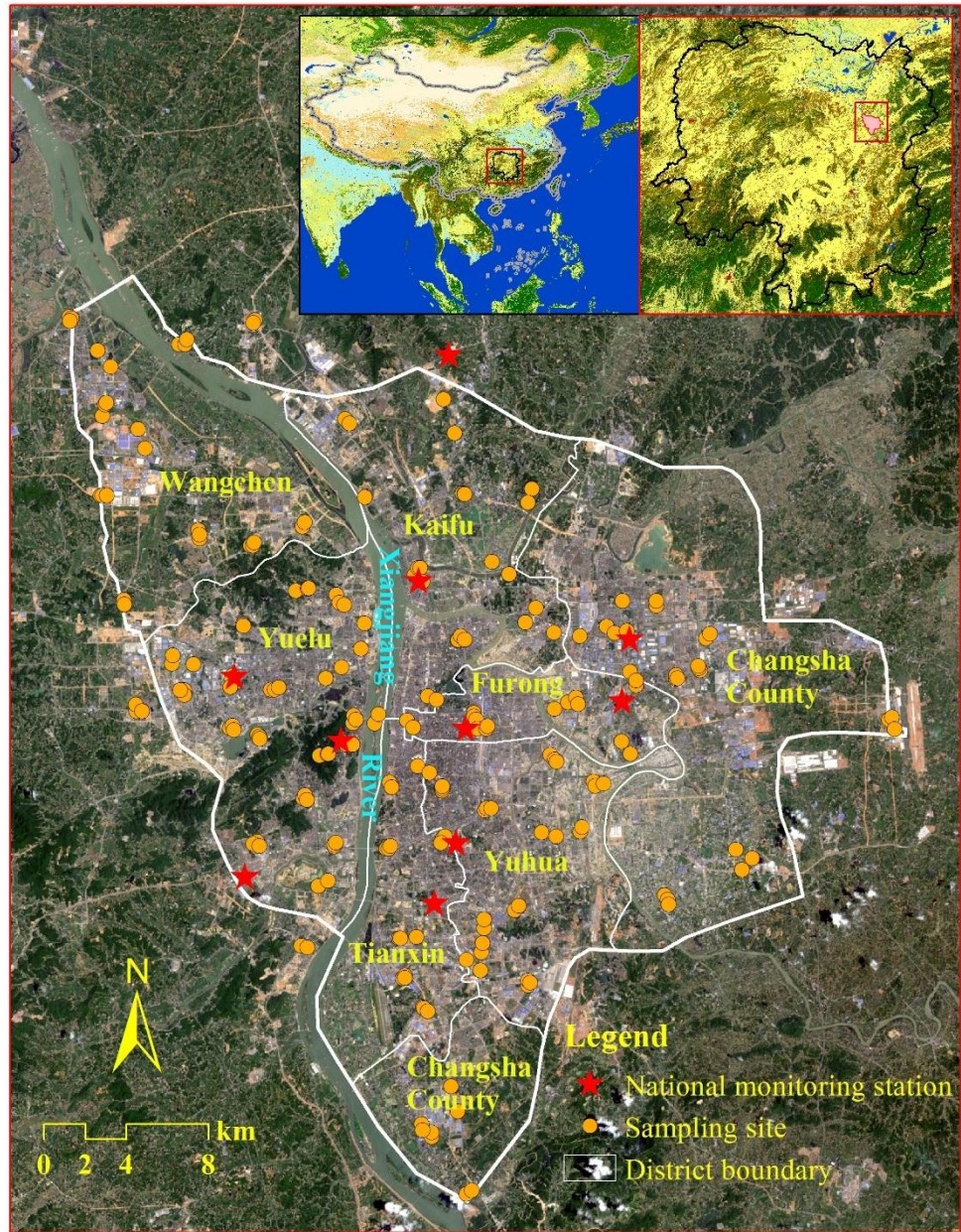

**Figure 2:** Sampling area and PM$_{2.5}$ sampling sites.

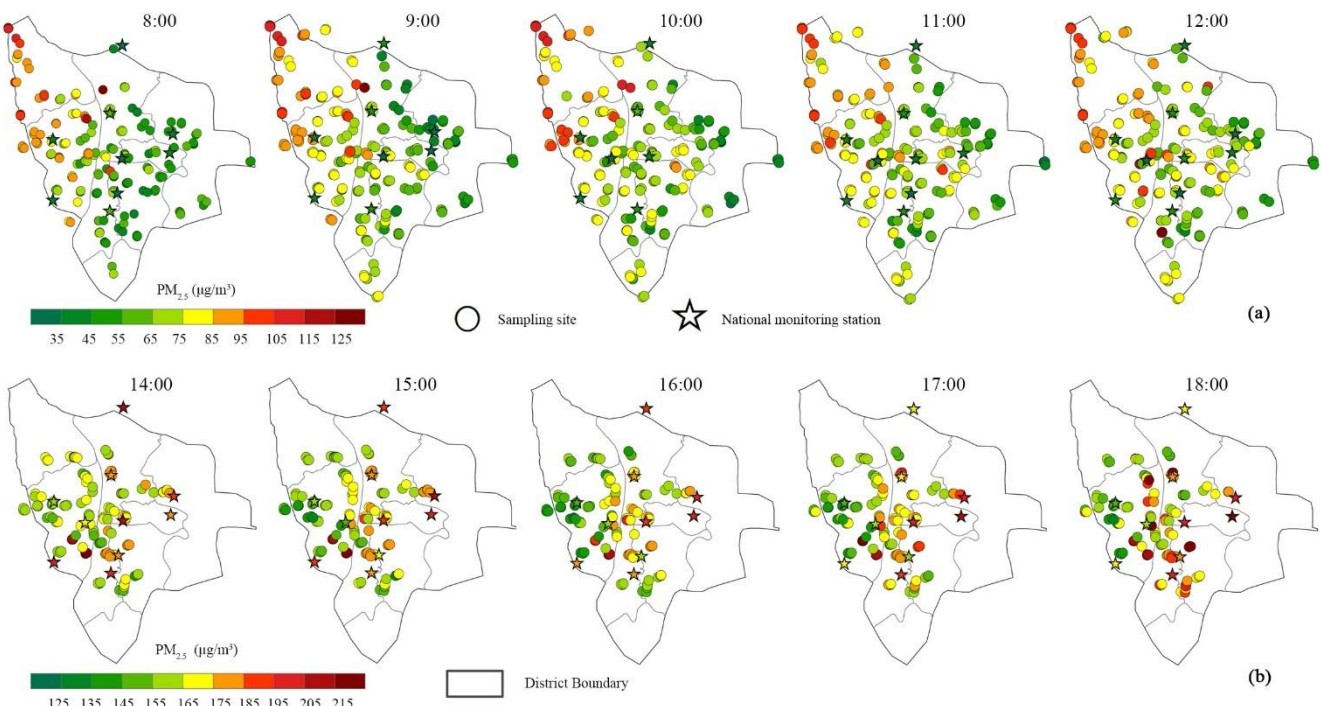

**Figure 3:** Spatial variation of PM$_{2.5}$ concentration of sampling sites: (a) Period 1, (b) Period 2.

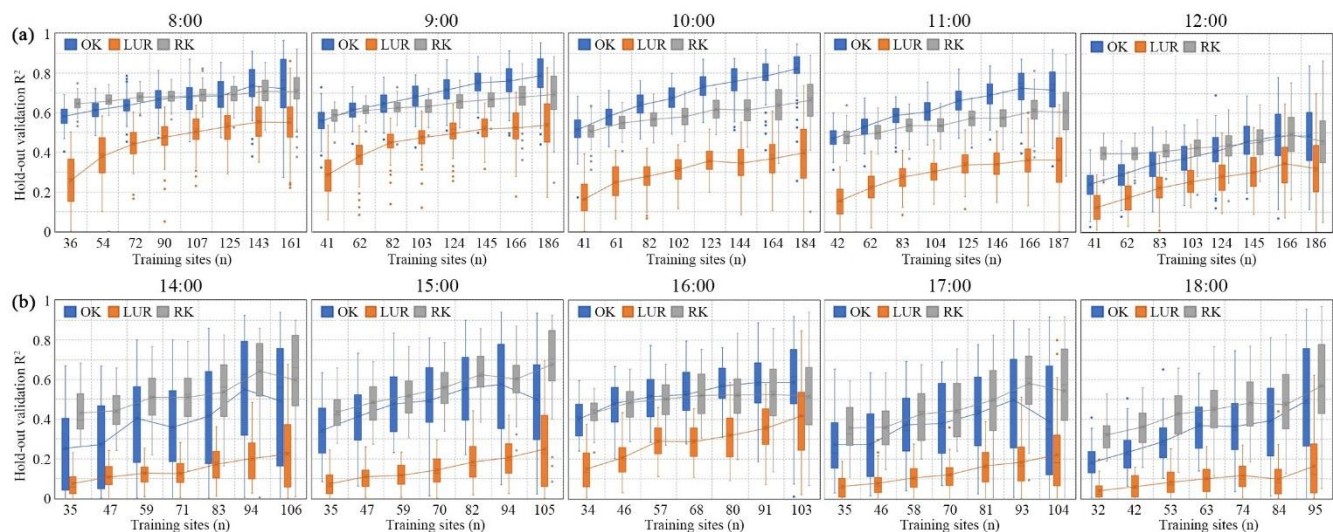

5   **Figure 4:** Box plots of hold-out validation R$^2$ between the observed concentration and predicted concentration of PM$_{2.5}$ for OK, LUR and RK with an increase in training sites: (a) Period 1; (b) Period 2. The boundaries of the boxes indicate the 75th percentile and 25th percentile (Q3 and Q1, respectively). The line within the box denotes the median (Q2), and the crosses denote the averages. The error bars above and below indicate the highest datum (Q3+1.5IQR, IQR is the interquartile range, IQR=Q3-Q1) and the lowest datum (Q1-1.5IQR), respectively. Dots above and below the error bars indicate the outliers.

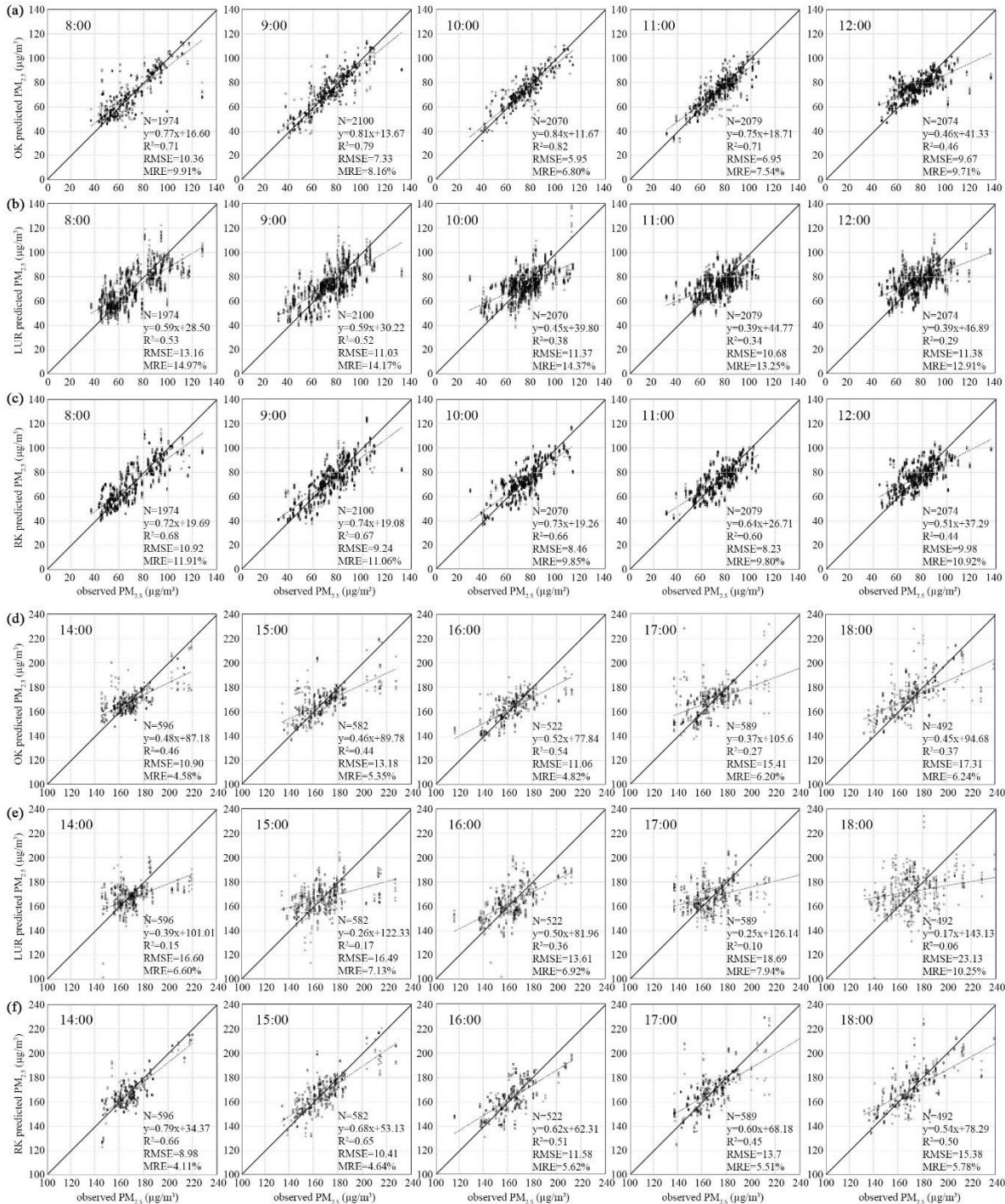

**Figure 5:** Scatterplots of repeated validating results with 90% training sites for (a) OK, Period 1; (b) LUR, Period 1; (c) RK, Period 1; (d) OK, Period 2; (e) LUR, Period 2; (f) RK, Period 2. The solid line is the 1:1 line, which is a reference.

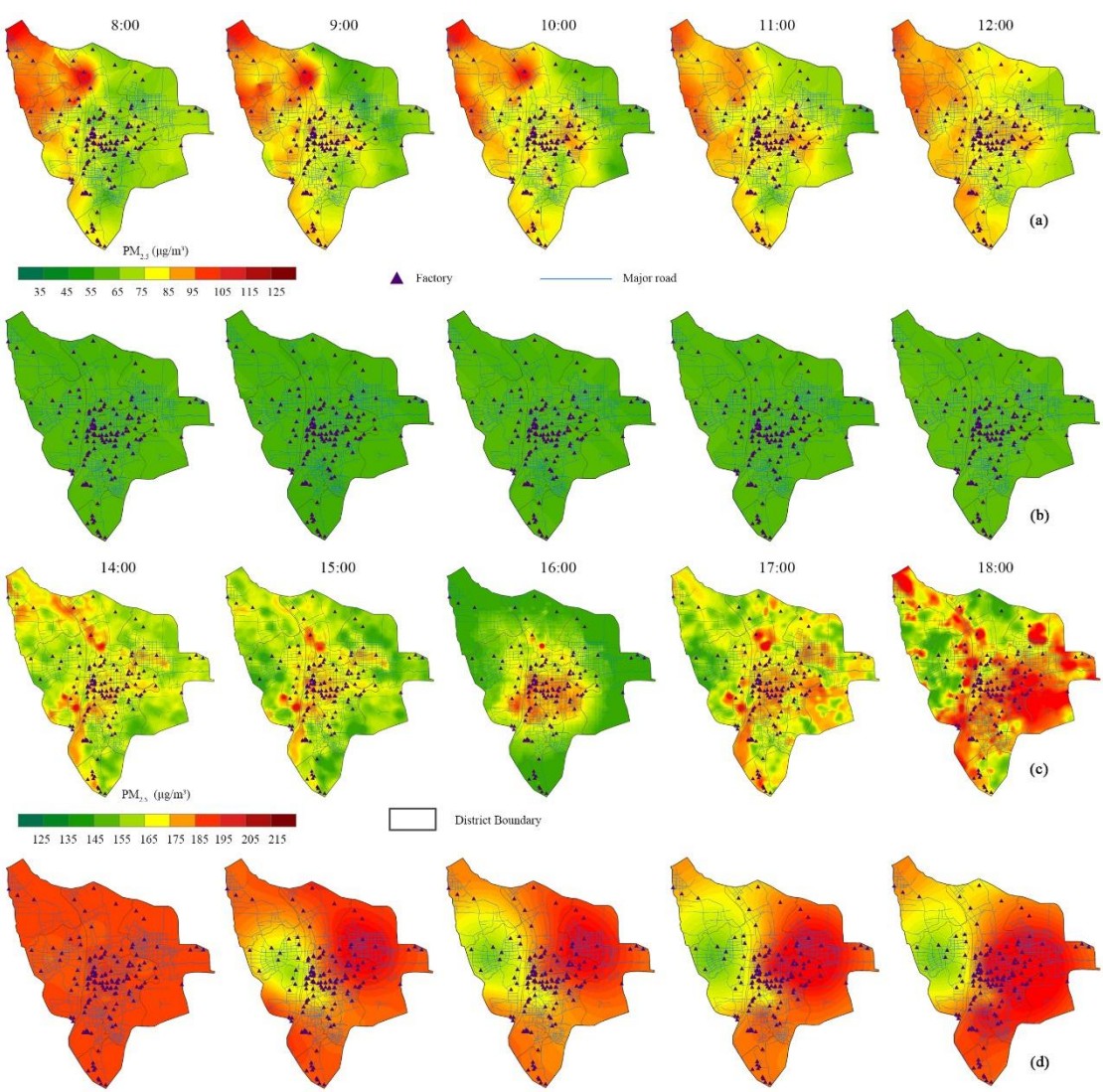

**Figure 6:** Spatial distributions of PM2.5 concentrations from crowdsourced sampling sites and national monitoring stations. (a) Period 1, crowdsourced sampling; (b) Period 1, national monitoring; (c) Period 2, crowdsourced sampling; (d) Period 2, national monitoring.