# Peer review of "Strategies of Method Selection for Fine Scale PM2.5 Mapping in an Intra-Urban Area Using Crowdsourced Monitoring"

_Atmospheric Measurement Techniques, 2018_

## Referee Comment (RC1) · Anonymous Referee #4 · 30 Jan 2019

This study used *in situ* PM2.5 measured by portable laser sir quality monitors to replace traditional PM2.5 data collected by ground monitoring stations or derived from remote sensing images and developed a new hybrid (land use regression plus geostatistical) method to map PM2.5 concentrations in an urban area. Generally, this manuscript is well organized and clearly written, even though a few of sentences need to be rephrased and more details need to be supplemented. I recommend the editor to accept this manuscript after a minor or moderate revision.

The authors developed a hybrid model in which the deterministic component of the PM2.5 concentration was fitted by LUR and the stochastic component (i.e. residual) was interpolated by kriging. Thus this is a typical LUR based REGRESSION kriging but not universal kriging. Please see Liu et al. (2018). Incorrectly naming the method is my biggest concern for the manuscript.

**Liu, Y. et al., 2018. Improve ground-level PM2.5 concentration mapping using a random forests-based geostatistical approach. Environmental Pollution, 235, 272-282.**

I am afraid that the Abstract from line 16 to 27 is not clear for a new reader especially who has not read the Method section. What do the "Period 1" and "Period 2" represent?

(Page 2, line 19) The authors should cite Liu et al. (2018) that is a typical study combining two technologies to estimate PM2.5 concentrations.

In the Measurement Instrument section, the authors may add more details for their portable air quality monitors, e.g. the company producing the equipment and other practical uses of the portable monitor.

(Page 4, lines 13-20). The sentences here are unclear and the authors may need to rewrite them. "*Sampling was carried out in two time periods in the winter of 2015…*" I am wondering

whether the authors can provide a specific time periods (e.g. from November 1 to December 31) to replace "the winter". "*The second period was between 14:00 and 18:00, when Orange warning signals of haze were released by Changsha Meteorology Bureau…*" I guess Orange warning signal was not released every day, but from your last sentence "*The first period was between 8:00 and 12:00, representing a light-polluted period*" it seems the Orange warning signal is released every afternoon. So please make it clear whether you measured PM2.5 concentrations during the two time slots all days or only Orange days. Additional, I suggest using "time slots" to replace "time periods". The "period" may be used for the days when you collected the PM2.5 concentration samples.

(Page 4, line 20). "*The official observations at 10 national monitoring  **stations**.*"

(Page 6, lines 21-22) "*Clearly, the average PM2.5 concentrations of Period 2 were **two times** higher than those of Period 1…*" I wonder why the authors emphasized "two times" higher here. It gave me a deep impression that "two times" implied something, but I have not seen any explanation for the "two times" in the following text. I would simply say: the average PM2.5 concentrations of Period 2 were much higher than …

(Page 9, lines 1-10) I cannot accept the authors' discussion in this paragraph whatsoever. Compared with the authors' cheaper potable air pollution monitors, I more trust instruments from national monitoring stations. "*This suggests the inconvenient **truth** (what a strong word! It is just a possible.) that the exposure risk remains relatively high for the public when official air pollution levels are "Good" and "Moderate" and this risk …*" I completely understand what the authors intend to express, but if the government intentionally falsified the air quality data, it was more likely to lower the heavy- rather than light-pollution data. I thought of another possibility: the authors' portable monitors were not sensitive for the low PM2.5 concentrations and are prone

to be saturated in the heavy-pollution days. In that case, it will also get the result the authors showed in the manuscript. The authors intended to emphasize that the large error (difference) on PM2.5 concentrations over the city is due to the relatively small number of national monitoring stations and thus their method using portable monitors to collect PM2.5 data is useful. However, based on the authors' statement, large differences on PM2.5 concentrations have existed even if concentrations are measured by the instruments of the national monitoring stations and the portable equipments of the authors at the same location.

I suggest the authors cautiously using some very strong adjectives and adverbs, such as clearly, significantly, tremendous, etc. (Abstract, line 25) "This method selection strategy provides **solid experimental** evidence for method selection of …" I will say "this study provides **empirical** evidence for …" Although generally clear for me, it is better to further polish the English of this manuscript, especially in the Results and Discussion sections.

---

## Referee Comment (RC2) · Anonymous Referee #3 · 4 Feb 2019

In this manuscript, the authors presented strategies of method selection for efficiently and effectively PM2.5 concentration mapping with increasing training sites based on a crowdsourcing sampling campaign. This study found that Ordinary Kriging (OK) interpolation performed best under conditions with non-peak traffic situation in light-polluted period, the Universal Kriging (UK) modeling performed better for conditions with the peak traffic and relatively few sampling sites in heavy-polluted period, and the Land Use Regression (LUR) model demonstrated limited ability in the estimation PM2.5 concentrations at very fine scale. Overall, the the manuscript is well-written and scientifically sounds good, and can be accepted after minor revision.

[Figure]

The authors should really redefine all acronyms in conclusions...Conclusions should broadly read as if the reader hadn't read the rest of the paper. Thus, the authors reintroduce everything, including hypothesis and research plan.

---

## Referee Comment (RC3) · Anonymous Referee #2 · 4 Feb 2019

General comments: Overall this is an interesting paper comparing methods for estimating spatial concentrations of PM2.5 using crowd sourced low-cost sensor measurements. I think it will be highly valuable for many researchers in the field interested in spatial variation. However, I think there is a lack of discussion of the limitations of low-cost optical particle sensors especially with the limited performance evaluation presented in this manuscript. I suggest major revisions for this paper. There are a number of places where the text is unclear and the authors should take care to thoroughly edit the next draft of this paper.

Specific comments:

[Figure]

Abstract: It's not clear what the different periods are referring to, morning versus afternoon? Line 19: I don't think that a range is the best statistic to show that 2 sets of numbers are "clearly different". Line 48: What do you mean by: "and a promising access to the prevention of exposure risks for individuals in their daily life."

Page 3 line 24-25: What does "data consistency" mean? Can you please elaborate. Also, where do you get the resolution data from? The manufacturer? Lab studies? Please cite.

Page 3 Line 30: Why would you only select 30 monitors to collocate? Without the collocation data from the other monitors you have no idea what the bias is of the other measurements.

Section 2.2.1: Can you mention if these monitors or internal sensors are commercially available or have been evaluated in any other studies, etc. Oh, I see in the supplement they are SDL307 but I think this may be important to add to the text.

Page 3 Lines 28-29: This is confusing to me. I don't see K factors anywhere when I look at the figure. Please clarify this sentence and/or move the figure reference to a more appropriate location.

Page 3 Line 30-Page 4 Line 3: I think the performance needs more discussion. How do the monitors compare to each other? If you are looking at spatial variability, bias/error between different monitors will be important. Were all monitors at the reference site for the same period? Is this 1-hr data shown in the plot or some other averaging time? Knowing the bias of individual monitors is very important because it will help determine at what threshold you can say there is likely spatial variation versus just bias in the sensor measurements. In addition, RH is known to significantly influence optical PM measurements. RH should be reported throughout. If RH is >75% during one of the periods (1,2, or comparison) this may be an issue. In addition, you have no data above ∼100 ug/m3 but during your second period the concentrations are in the 170-180 range. I think it is important to know how the sensors perform at these

high concentrations if you are going to try to draw conclusions. Has any previous work evaluated these sensors at high concentrations? You cannot assume that just because they work well from the 40-100 range they will work the same below and above that.

Page 5 lines 17-18: Meteorological data with a spatial resolution of roughly 0.4 sites per 100 km2 (wind speed, atmospheric pressure, relative humidity, temperature) that -I think it might be clearer to just list the number of stations you had in total over your sampling area.

Section 2.1.2: I'm not clear how this data is crowdsourced can you please include more information about how each monitor got to each monitoring point.

Page 6 lines 19-21: Is this the highest and lowest one-hour average from a single site and single monitor? Why are these and the times they occurred important?

Line 20: These what? Averages?

Line 20-21: I don't know what the numbers in parenthesis are please clarify

Lines 25 and 26: Is there more traffic at noon than at morning rush hour? Also is the average concentration at the different hours significantly different?

Figure 3. Does each of these points represent a single monitor? Why are they fewer monitors during period 2?

Line 13: I don't understand what you are comparing that increased. What is the first set of numbers versus the seconded set of numbers?

Line 14: What do you mean significant and steady decrease? Decreased by hour by the same amount?

Page 7 Line 30: Since readers can see the individual R2 on the figure it may be easier to digest if you just include an average or range instead of so many lists of numbers.

Page 8 Line 5: I read this paragraph a couple times and I'm still a bit confused which

method performs the best. Can you add a summary sentence at the end just stating the conclusion? Or reorganize more clearly.

Section 3.3: Can you clarify: did you use 90% training sites for only the sensor measurements and then only 90% of the reference stations? As far as I could tell previously you only used withholding from the sensor data and didn't evaluate the models using the reference data?

Page 8 line 9: "Significant difference can be found between two sources," what do you mean?

Page 8 Line 13: What do you mean three-step growth?

Page 8 Line 15: I don't understand based on the figure it seems like there are almost no factories and roads in the top left corner but that is where most of the pollution is.

Page 8 Line 30-Page 9 Line 3: I think you need to mention though the limitations of low-cost monitors and the inaccuracies in these measurements compared to federal methods.

Page 9 Line 10: It seems likely the low-cost sensors may have been saturated at the high concentrations and this may have led to the difference between the sensors and the reference methods.

Technical corrections:

Suggest rewording the title for clarity, possibly: Strategies of method selection for Fine Scale PM2.5 mapping in an intra-urban area using crowdsourced monitoring

Fine particulate matter (particulate matter singular remove s)

Line 9: to "the" public – there are a number of grammatical errors throughout the text and I have not had a chance to identify them all in this review. Please review for grammar.

Page 6 Line 20 ug/m3 formatting

Page 7 Line 4: Remove "had" assuming you are talking about this work where the sites experienced extreme PM

---

## Referee Comment (RC4) · Anonymous Referee #1 · 5 Feb 2019

Xu et al describe measurements and spatial modeling of PM2.5. Measurements were conducted with hand-held optical particle monitors. The spatial modeling compared multiple methods: ordinary kriging, universal kriging, and land use regression.

The paper suffers from several critical flaws and is not publishable in its current form. Below I outline five major problems with the manuscript.

Major Issue #1: I do not know what the authors mean by a "crowdsourced" data collection. The authors seem to define crowdsourcing in lines 27-28 of page 2, but "Crowdsourcing activities based on informal social networks and web 2.0 technologies that allowed citizens themselves to produce geospatial data among others" seems more

[Figure]

like corporate jargon than a useful explanation of crowdsourcing.

The sampling approach seems to be short-term saturation sampling - many volunteers simultaneously sampled at predetermined locations. This sampling approach does not fit my personal notion of crowdsourcing, which would be a more informal data collection leveraging people's normal movements throughout the day. Sending an army of students to collect data in an organized fashion seems less like "crowdsourcing" and more like a sampling campaign. In that sense, this study has little distinction from the large literature on distributed air quality sampling.

What would be the value or longer-term viability of this or a similar sampling approach? This paper focuses on two short sampling periods of a few hours each, so the data are unlikely representative of long-term spatial patterns. Do the authors expect to deploy an army of distributed samplers on a semi-regular basis in order to build up a dataset capable of reproducing longer-term trends? Or to send out volunteers daily to make daily maps? I don't see how the "crowdsourced" aspect of this adds value or novelty; instead it seems like crowdsourcing is being used as a buzzword.

Major Issue #2: Data quality. Figure 1 shows one short-term comparison between the handheld PM monitors and the regulatory monitors. While there is generally good agreement, there is a fair amount of scatter among the handheld monitors. This scatter is to be expected given the low cost and the use of optical particle detection. However, the authors do not address how uncertainty in the measurements potentially impacts the mapping. Nor do they seem to account for uncertainty in the measurements or make any efforts to correct the measurements (e.g., based on hygroscopic growth).

Table 3 and section 3.1 - the crowdsourced data read higher PM than the regulatory data. The authors have not convinced me that this is not an artifact of the sensors they have chosen. During some hours there is significant difference between the mean "crowdsourced" PM and the mean regulatory PM. Since the overall spatial extent of the two sampling domains (regulatory and crowdsourced) is roughly similar, I would expect

similar mean concentrations from each dataset.

Line 30 on page 8 calls the national monitoring sites "inaccurate." I am not familiar with regulatory measurement policies in China, but if they are anything like the US and Europe, the accuracy standard is high. The spatial pattern derived from these few monitors may be erroneous, but the specific measurements are accurate.

Major Issue #3: Site selection and sampling strategy The description of the sampling strategy is insufficient. Were all samplers deployed simultaneously at all sites in Table 1? How were the sampling times defined and chosen? What are significant differences between period 1 and period 2?

Table 1 - A better description of each type of site is needed. For example, Dust surfaces seem to be defined as "dust surfaces," which is not helpful to readers. What qualifies as a dust surface? Some entries in this table have "A" and "U". What do those designations mean?

Major Issue #4: Modeling and interpretation. The modeling aspect of this paper is not novel. Since the sampling method seems to be a straightforward saturation sampling campaign, using the resulting data to build spatial models is not a novel contribution. Numerous papers have already done this for PM2.5, as noted by the authors.

One main conclusion seems to be that the modeling approaches work. This is not all that novel - it is more a statistical finding than an atmospheric measurement technique. Numerous papers have shown that LUR and kriging models can be fit to spatially distributed measurements.

Another conclusion is that the models work better when provided with more training sites. Again this seems like an obvious outcome, especially for the kriging approaches.

A more relevant analysis would be to evaluate if the models (and measurements) make physical dense. In Figure 5 there is a PM hotspot in the northwestern part of the domain on Day 1 and in the center of the domain on Day 2. Do these hotspots make sense

given the distribution of sources and the climatology?

Major Issue #5: The paper needs a thorough review and edit for English grammar. There are many grammar errors (too many to count or enumerate here), and in other places the language is hard to follow.

---

## Author Comment (AC1) · 4 Mar 2019

Answer to Referee #4

We thank the referee for his/her very careful review, and his/her constructive suggestions. In the following, we answer his/her specific questions. In order to facilitate the reference to the questions and proposed changes, we use the following color coding:

Color coding:
reviewer comment
our answer
proposed change in manuscript
* * *
This study used in situ PM2.5 measured by portable laser sir quality monitors to replace traditional PM2.5 data collected by ground monitoring stations or derived from remote sensing images and developed a new hybrid (land use regression plus geostatistical) method to map PM2.5 concentrations in an urban area. Generally, this manuscript is well organized and clearly written, even though a few of sentences need to be rephrased and more details need to be supplemented. I recommend the editor to accept this manuscript after a minor or moderate revision.

The authors developed a hybrid model in which the deterministic component of the PM2.5 concentration was fitted by LUR and the stochastic component (i.e. residual) was interpolated by kriging. Thus this is a typical LUR based REGRESSION kriging but not universal kriging. Please see Liu et al. (2018). Incorrectly naming the method is my biggest concern for the manuscript.

Liu, Y. et al., 2018. Improve ground-level PM2.5 concentration mapping using a random forests-based geostatistical approach. Environmental Pollution, 235, 272-282.

Response: the naming of this method followed Mercer et al. (Atmospheric Environment 2011). They proposed a 2-step approach in which simple kriging is applied to the residuals from LUR. This approach is similar but not identical to UK. Thus, we agree with the reviewer that the Regression Kriging is more appropriate and thank him/her for the suggestion. We implemented the changes in the revised manuscript.

Mercer, L. D., Szpiro, A. A., Sheppard, L., Lindström, J., Adar, S. D., Allen, R. W., Avol, EL., Oron, A. P., Larson, T., Liu, L. J., and Kaufman, J. D.: Comparing universal kriging and land-use regression for predicting concentrations of gaseous oxides of nitrogen (NOx) for the multi-ethnic study of atherosclerosis and air pollution (MESA Air), Atmos Environ, 45, 4412–4420, doi:10.1016/j.atmosenv.2011.05.043, 2011.

I am afraid that the Abstract from line 16 to 27 is not clear for a new reader

especially who has not read the Method section. What do the "Period 1" and "Period 2" represent?

Response: "Period 1" and "Period 2" represent the light-polluted period and heavy-polluted period. We rewrote these confusing sentences and replaced lines 15 –27 on Page 1 by:

During this process, $PM_{2.5}$ concentrations were measured by laser air quality monitors and uploaded by a group of volunteers via their smart phone applications during two periods. Three extensively employed modelling methods (ordinary kriging (OK), land use regression (LUR), and regression kriging (RK) were adopted to evaluate the performance. An interesting finding is that $PM_{2.5}$ concentrations in micro-environments significantly varied in the intra-urban area. These local $PM_{2.5}$ variations can be effectively identified by crowdsourced sampling rather than national air quality monitoring stations (light-polluted period: $(69.67\pm18.81) – (76.45\pm14.55)$ µg m$^{-3}$ vs. $(36.9\pm10.97) – (41.2\pm8.68)$ µg m$^{-3}$; heavy-polluted period: $(162.72\pm15.96) – (171.89\pm21.5)$ µg m$^{-3}$ vs. $(177.8\pm16.91) – (188.3\pm22.4)$ µg m$^{-3}$).. The selection of models for fine scale $PM_{2.5}$ concentration mapping should be adjusted according to the changing sampling and pollution circumstances. Generally, OK interpolation performs best in conditions with non-peak traffic situations during a light-polluted period (hold-out validation $R^2$: 0.47–0.82), while the RK modelling can perform better during the heavy-polluted period (0.32–0.68) and in conditions with peak traffic and relatively few sampling sites (less than ~100) during the light-polluted period (0.40–0.69). Additionally, the LUR model demonstrates limited ability in estimating $PM_{2.5}$ concentrations on very fine spatial and temporal scales in this study (0.04–0.55), which challenges the traditional point about the good performance of the LUR model for air pollution mapping. This method selection strategy provides empirical evidence for the best method selection for $PM_{2.5}$ mapping using crowdsourced monitoring, and this provides a promising way to reduce the exposure risks for individuals in their daily life.

(Page 2, line 19) The authors should cite Liu et al. (2018) that is a typical study combining two technologies to estimate PM2.5 concentrations.

Response: Liu et al. (2018) adopted a random forests-based regression kriging approach which integrates recent advancements of machine learning with conventional kriging methods in geostatistics. We thank the reviewer for the suggestion and cited this article in the revised manuscript.

In the Measurement Instrument section, the authors may add more details for their portable air quality monitors, e.g. the company producing the equipment and other practical uses of the portable monitor.

Response: more details were added as the reviewer asked. Replaced lines 23 –29 on Page 3 by:

The portable laser air quality monitor SDL307 (produced by NOVA FITNESS Co., Ltd.) is employed to perform sampling. The monitor manual can be downloaded from http://www.inovafitness.com/index.html. This monitor can be conveniently carried with a total size of 25×34×14 cm (Fig. 1a). According to the test report provided by the Center for Building Environment Test at Tsinghua University, the maximum relative error of this monitor is ±20% compared with a regulatory monitor in the 20–1000 μg m$^{-3}$ range and has a resolution of 0.1 μg m$^{-3}$. The concentration of particulate matter is measured using the light-scattering method (Fig. 1b). The monitor contains a special laser module, and the signals are recorded by a photoelectric receptor when particulate matter passes through laser light. The count and size of particulate matter are then analysed by a microcomputer after the signals are amplified and converted. Their mass concentrations are calculated based on the conversion factor between the light-scattering method and the tapered element oscillating microbalance technology.

(Page 4, lines 13-20). The sentences here are unclear and the authors may need to rewrite them. "Sampling was carried out in two time periods in the winter of 2015…" I am wondering whether the authors can provide a specific time periods (e.g. from November 1 to December 31) to replace "the winter". "The second period was between 14:00 and 18:00, when Orange warning signals of haze were released by Changsha Meteorology Bureau…" I guess Orange warning signal was not released every day, but from your last sentence "The first period was between 8:00 and 12:00, representing a light-polluted period" it seems the Orange warning signal is released every afternoon. So please make it clear whether you measured PM2.5 concentrations during the two time slots all days or only Orange days. Additional, I suggest using "time slots" to replace "time periods". The "period" may be used for the days when you collected the PM2.5 concentration samples.

Response: In fact, due to the difficulties in implementing the campaign (e.g. the financial burdens of volunteers' recruitment and the extensive investment of time and efforts for technology part and procedures to ensuring data quality), we only carried out this sampling between 8:00 and 12:00 on December 24 and 14:00 and 18:00 on December 25. In the first period, the official air pollution levels were "Good" and "Moderate", in the second period, the Changsha Meteorology Bureau released an Orange warning signal of haze (i.e. the official air pollution level was "Heavily Polluted"). We rewrote these confusing sentences and replaced lines 16 – 25 on Page 4 by:

Sampling was performed in two time periods in the winter of 2015 to examine the effect of air quality grades on the mapping results. The first period fell between

8:00 and 12:00 on December 24. In this period, the official air pollution levels were "Good" and "Moderate" (i.e., Period 1, light-polluted period). The weather was overcast with occasional rain or drizzle, and the relative humidity (RH) ranged from 95% to 98%. The second period extended between 14:00 and 18:00 on December 25, when an orange warning signal of haze (i.e., official air pollution level was "Heavily Polluted") was released by the Changsha Meteorology Bureau (i.e., Period 2, heavy-polluted period). The weather was cloudy with some sunshine, and the RH ranged from 39%–43%.

Before sampling started, every volunteer received one monitor and went to the corresponding area. At each potential monitoring site, the volunteer lifted the monitor (~2 metres above the ground) and held it for at least 60 seconds to measure the PM2.5 concentration. The observations were uploaded twice to four times hourly using a smart phone application (App) that we developed. The geographic coordinates of the sampling sites were also uploaded. For each hour, we eliminated the sampling sites with less than three observations. The valid observations were then averaged at each site. As some volunteers quit after the sampling of the first period, the sampling sites in period 2 were concentrated in the central study area. A total of 179-208 samples were successfully collected at each hour in Period 1, and 105-118 samples were successfully collected in Period 2. The official observations at 10 national monitoring stations in the study area were also obtained (China Environmental Monitoring Center, CEMC: http://106.37.208.233:20035/) and averaged for comparison purposes.

(Page 4, line 20). "The official observations at 10 national monitoring sites stations."

Response: corrected.

(Page 6, lines 21-22) "Clearly, the average PM2.5 concentrations of Period 2 were two times higher than those of Period 1…" I wonder why the authors emphasized "two times" higher here. It gave me a deep impression that "two times" implied something, but I have not seen any explanation for the "two times" in the following text. I would simply say: the average PM2.5 concentrations of Period 2 were much higher than …

Response: we thank the reviewer for the suggestion and rewrote this sentence.

(Page 9, lines 1-10) I cannot accept the authors' discussion in this paragraph whatsoever. Compared with the authors' cheaper potable air pollution monitors, I more trust instruments from national monitoring stations. "This suggests the inconvenient truth (what a strong word! It is just a possible.) that the exposure risk remains relatively high for the public when official air pollution levels are "Good" and "Moderate" and this risk …" I completely understand what the authors intend to express, but if the government intentionally falsified the air quality data, it was more likely to lower the heavy- rather than light-pollution data. I thought of

another possibility: the authors' portable monitors were not sensitive for the low PM2.5 concentrations and are proneto be saturated in the heavy-pollution days. In that case, it will also get the result the authors showed in the manuscript. The authors intended to emphasize that the large error (difference) on PM2.5 concentrations over the city is due to the relatively small number of national monitoring stations and thus their method using portable monitors to collect PM2.5 data is useful. However, based on the authors' statement, large differences on PM2.5 concentrations have existed even if concentrations are measured by the instruments of the national monitoring stations and the portable equipments of the authors at the same location.

Response: we agree with the reviewer that the instruments of national monitoring stations are more accurate and reliable than the potable air pollution monitors, and that is the reason why we conducted the comparison experiments between laser air quality monitors and the national monitoring instruments at the same positions and heights before and after the crowdsourcing sampling. The point we intend to make is that the crowdsourced $PM_{2.5}$ measurements demonstrated obvious spatial variation between urban microenvironments, and these variations can hardly be disclosed by sparse national air quality monitoring stations. The difference of hourly $PM_{2.5}$ concentrations between the two types of instruments in sampling campaign is possibly because of the different sampling heights and the change of the major pollution sources in the study area. We thank the reviewer for pointing out this issue. We rewrote these confusing sentences and replaced lines 25 –30 on Page 8 and lines 1 –13 on Page 9 by:

The number of sampling sites were 18 and 10 per 100 km² for Period 1 and Period 2, respectively. These data comprise a considerable improvement compared with a density of approximately 0.015 sites per 100 km² in the national air quality monitoring network in China. As expected, crowdsourced $PM_{2.5}$ measurements demonstrated detailed spatial variation among urban microenvironments, and these variations can hardly be disclosed by sparse national air quality monitoring stations. This finding suggests that crowdsourced sampling can effectively improve the density of $PM_{2.5}$ monitoring at a rather low monetary cost and can be supportive of the short-term air pollution exposure assessment for epidemiologic studies at a fine scale. To explore the spatial variation in the $PM_{2.5}$ concentration for various urban microenvironments and compare with the national air quality measurements, the crowdsourced monitoring is assumed to cover a certain number of areas. However, persuading the general public in these areas to continuously observe and upload $PM_{2.5}$ concentrations during their activities of daily living through a designed study is difficult. We employed a batch of volunteers to model their behaviours on the general public's behaviour and simultaneously collect data. This approach is a preliminary practice of crowdsourced monitoring and can be further developed and improved in the long-term exposure assessment at the fine scale in the future with the progress in lowcost wearable air quality monitors and automatic processing techniques of crowdsourced data.

The hourly $PM_{2.5}$ concentrations between crowdsourced sampling sites and national monitoring stations were rather different; this difference varied as the official air quality level changed. The crowdsourced $PM_{2.5}$ concentrations were substantially larger than the national concentrations in Period 1 (light-polluted) and slightly lower in Period 2 (heavy-polluted). One possible reason is that the national monitoring stations in the study area were installed on the roofs of mid-rise buildings (i.e., ~15 m) with ventilation and spaciousness, while crowdsourced sampling was conducted on the real ground (i.e., ~2 m). The change in the major pollution sources and meteorological conditions in the study area may contribute to the difference between two periods; the major contribution of local sources, especially the vehicle emission and the very high RH (95%–98%) during the light-polluted period, may cause the accumulation of $PM_{2.5}$ near the ground; and the sources of long-range transport of regional pollution during the heavy-polluted period can increase the concentration of $PM_{2.5}$ on the upper layer. This finding suggests that the air pollution exposure risk may remain relatively high for the public on the ground in some urban microenvironments, even when official air pollution levels are "Good" and "Moderate" and sensitive groups should consider reducing some outdoor activities. The results confirm the necessity of developing real-ground high-density crowdsourced $PM_{2.5}$ monitoring networks. Although the low-cost sensor and the use of optical particle detection of monitors in sampling may cause inaccuracies in measurements, we have attempted to minimise the uncertainty by disusing the relatively inaccurate monitors (MRE>5%) used in preliminary indoor and outdoor experiments. Comparison experiments between laser air quality monitors and the national monitoring instruments were also conducted at the same positions and heights for two time slots; the weather conditions and air quality scenarios of the two time slots were similar to the two sampling periods (i.e., overcast with light rain, RH≥76%: December 20–22 vs. Period 1; cloudy with sunshine, RH≤67%: December 29–31 vs. Period 2). The relatively good agreement between the hourly PM2.5 concentrations of laser monitors and those of national instruments had guaranteed the reliability of sampling data to a certain extent. The relative humidity may have slightly influenced the crowdsourced $PM_{2.5}$ concentrations in the light-polluted period since December 20–22 yielded a slightly lower $R^2$ and RMSE than those of December 29–31 but a higher MRE than that of December 29–31. However, the relative error of $PM_{2.5}$ observations in preliminary and comparison experiments were generally small and fluctuated without distinct trends and leading factors. During the following procedure of mapping method selection, three methods were performed with the same dataset, which caused a limited influence of uncertainty in measurements on the method comparison results; therefore, we did not correct the measurements in this study. However, more efforts are needed in crowdsourced measurements correction and uncertainty analysis in air pollution concentration mapping at high resolution for accurate exposure assessment in the

future.

I suggest the authors cautiously using some very strong adjectives and adverbs, such as clearly, significantly, tremendous, etc. (Abstract, line 25) "This method selection strategy provides solid experimental evidence for method selection of …" I will say "this study provides empirical evidence for …" Although generally clear for me, it is better to further polish the English of this manuscript, especially in the Results and Discussion sections.

Response: we thank the reviewer for the suggestion. Meanwhile, this manuscript was edited for proper English language, grammar, punctuation, spelling, and overall style by one or more of the highly qualified native English speaking editors at American Journal Experts. The certificate may be verified at www.aje.com/certificate with a certificate verification key of E57E-12C6-6B0F-0300-999B.

---

## Author Comment (AC2) · 4 Mar 2019

Anonymous Referee #3 In this manuscript, the authors presented strategies of method selection for efficiently and effectively PM2.5 concentration mapping with increasing training sites based on a crowdsourcing sampling campaign. This study found that Ordinary Kriging (OK) interpolation performed best under conditions with non-peak traffic situation in lightpolluted period, the Universal Kriging (UK) modeling performed better for conditions with the peak traffic and relatively few sampling sites in heavy-polluted period, and the Land Use Regression (LUR) model demonstrated limited ability in the estimation PM2.5 concentrations

at very fine scale. Overall, the the manuscript is well-written and scientifically sounds good, and can be accepted after minor revision.The authors should really redefine all acronyms in conclusions. . .Conclusions should broadly read as if the reader hadn't read the rest of the paper. Thus, the authors reintroduce everything, including hypothesis and research plan.

Response: Thank you very much for your thoughtful comments and suggestions. The conclusions were changed as: This study presented strategies of method selection for efficient PM2.5 concentration mapping with an increasing number of training sites using crowdsourced monitoring. The results confirmed that PM2.5 concentrations in microenvironments varied across the intra-urban area in China's cities. These variations can be clearly disclosed by the crowdsourced PM2.5 sampling rather than the national air quality monitoring sites. The selection of models for fine scale PM2.5 concentration mapping should be adjusted with changing sampling and pollution circumstances. Generally, ordinary kriging (OK) interpolation performs the best in conditions with non-peak traffic situations in the light-polluted period, while regression kriging (RK) can perform better in the heavy-polluted period and conditions with peak traffic and relatively few sampling sites in the light-polluted period. Additionally, note that the LUR model demonstrates a limited ability in estimating PM2.5 concentrations at very fine scale in this study. This method selection strategy provides empirical evidence for the method selection of PM2.5 mapping using crowdsourced monitoring and a promising way to reduce the exposure risks for individuals in their daily lives.

---

## Author Comment (AC3) · 4 Mar 2019

Answer to Referee #2

We thank the referee for his/her very careful review, and his/her constructive suggestions. In the following, we answer his/her specific questions. In order to facilitate the reference to the questions and proposed changes, we use the following color coding:

Color coding:
reviewer comment
our answer
proposed change in manuscript
* * *
General comments: Overall this is an interesting paper comparing methods for estimating spatial concentrations of PM2.5 using crowd sourced low-cost sensor measurements. I think it will be highly valuable for many researchers in the field interested in spatial variation. However, I think there is a lack of discussion of the limitations of low-cost optical particle sensors especially with the limited performance evaluation presented in this manuscript. I suggest major revisions for this paper. There are a number of places where the text is unclear and the authors should take care to thoroughly edit the next draft of this paper.

Specific comments:
Abstract: It's not clear what the different periods are referring to, morning versus afternoon? Line 19: I don't think that a range is the best statistic to show that 2 sets of numbers are "clearly different". Line 48: What do you mean by: "and a promising access to the prevention of exposure risks for individuals in their daily life."

Response: the statistic of range was replaced by Mean±SD. Meanwhile, we rewrote these confusing sentences and replaced lines 15 –27 on Page 1 by:

Fine particulate matter (PM$_{2.5}$) is of great concern to the public due to its significant risk to human health. Numerous methods have been developed to estimate spatial PM$_{2.5}$ concentrations in unobserved locations due to the sparse number of fixed monitoring stations. Due to an increase in low-cost sensing for air pollution monitoring, crowdsourced monitoring of fine exposure control has been gradually introduced into cities. However, the optimal mapping method for conventional sparse fixed measurements may not be suitable for this new high-density monitoring approach. This study presents a crowdsourced sampling campaign and strategies of method selection for hundred metre-scale level PM$_{2.5}$ mapping in an intra-urban area of China. During this process, PM$_{2.5}$ concentrations were measured by laser air quality monitors and uploaded by a group of volunteers via their smart phone applications during two periods. Three extensively employed modelling methods (ordinary kriging (OK), land use regression (LUR), and regression kriging (RK) were adopted to evaluate the

performance. An interesting finding is that $PM_{2.5}$ concentrations in micro-environments significantly varied in the intra-urban area. These local $PM_{2.5}$ variations can be effectively identified by crowdsourced sampling rather than national air quality monitoring stations (light-polluted period: (69.67±18.81) – (76.45±14.55) µg m$^{-3}$ vs. (36.9±10.97) – (41.2±8.68) µg m$^{-3}$; heavy-polluted period: (162.72±15.96) – (171.89±21.5) µg m$^{-3}$ vs. (177.8±16.91) – (188.3±22.4) µg m$^{-3}$).. The selection of models for fine scale $PM_{2.5}$ concentration mapping should be adjusted according to the changing sampling and pollution circumstances. Generally, OK interpolation performs best in conditions with non-peak traffic situations during a light-polluted period (hold-out validation $R^2$: 0.47–0.82), while the RK modelling can perform better during the heavy-polluted period (0.32–0.68) and in conditions with peak traffic and relatively few sampling sites (less than ~100) during the light-polluted period (0.40–0.69). Additionally, the LUR model demonstrates limited ability in estimating $PM_{2.5}$ concentrations on very fine spatial and temporal scales in this study (0.04–0.55), which challenges the traditional point about the good performance of the LUR model for air pollution mapping. This method selection strategy provides empirical evidence for the best method selection for $PM_{2.5}$ mapping using crowdsourced monitoring, and this provides a promising way to reduce the exposure risks for individuals in their daily life.

Page 3 line 24-25: What does "data consistency" mean? Can you please elaborate. Also, where do you get the resolution data from? The manufacturer? Lab studies? Please cite.

Response: we rewrote these confusing sentences and more details about this monitor were added.

Replaced lines 23 –29 on Page 3 by:

The portable laser air quality monitor SDL307 (produced by NOVA FITNESS Co., Ltd.) is employed to perform sampling. The monitor manual can be downloaded from http://www.inovafitness.com/index.html. This monitor can be conveniently carried with a total size of 25×34×14 cm (Fig. 1a). According to the test report provided by the Center for Building Environment Test at Tsinghua University, the maximum relative error of this monitor is ±20% compared with a regulatory monitor in the 20–1000 µg m$^{-3}$ range and has a resolution of 0.1 µg m$^{-3}$. The concentration of particulate matter is measured using the light-scattering method (Fig. 1b). The monitor contains a special laser module, and the signals are recorded by a photoelectric receptor when particulate matter passes through laser light. The count and size of particulate matter are then analysed by a microcomputer after the signals are amplified and converted. Their mass concentrations are calculated based on the conversion factor between the light-scattering method and the tapered element oscillating microbalance technology.

Page 3 Line 30: Why would you only select 30 monitors to collocate? Without the collocation data from the other monitors you have no idea what the bias is of the other measurements.

Response: In fact, the 86 portable laser air quality monitors we used in the sampling were selected from 115 monitors through preliminary indoor and outdoor experiments. The relative errors between each other were no larger than 5%, which guaranteed the reliability of sampling data of the other measurements to a certain extent. Sentences about this were added in **2.1.1 Measurement instrument**. Under the circumstance that the national monitoring stations do not have enough room for more laser monitors to conduct the comparison experiments, we only randomly selected 30 monitors.

Section 2.2.1: Can you mention if these monitors or internal sensors are commercially available or have been evaluated in any other studies, etc. Oh, I see in the supplement they are SDL307 but I think this may be important to add to the text.

Response: added.

Page 3 Lines 28-29: This is confusing to me. I don't see K factors anywhere when I look at the figure. Please clarify this sentence and/or move the figure reference to a more appropriate location.

Response: made the changes as the reviewer suggested in the revised manuscript.

Page 3 Line 30-Page 4 Line 3: I think the performance needs more discussion. How do the monitors compare to each other? If you are looking at spatial variability, bias/error between different monitors will be important. Were all monitors at the reference site for the same period? Is this 1-hr data shown in the plot or some other averaging time? Knowing the bias of individual monitors is very important because it will help determine at what threshold you can say there is likely spatial variation versus just bias in the sensor measurements. In addition, RH is known to significantly influence optical PM measurements. RH should be reported throughout. If RH is >75% during one of the periods (1,2, or comparison) this may be an issue. In addition, you have no data above ~100 ug/m3 but during your second period the concentrations are in the 170-180 range. I think it is important to know how the sensors perform at these high concentrations if you are going to try to draw conclusions. Has any previous work evaluated these sensors at high concentrations? You cannot assume that just because they work well from the 40-100 range they will work the same below and above that.

Response: Comparison experiments between laser air quality monitors and the national monitoring instruments were also conducted at the same positions and heights for two time slots; the weather conditions (including RH) and air quality scenarios of the two time slots were similar to the two sampling periods. In the

previous version of the manuscript, we thought one comparison result is enough to demonstrate the reliability of sampling data to a certain extent, we thank the reviewer for pointing out the inadequacies. Sentences about data quality and Figure 1d were added.

On the one hand, the relative error of $PM_{2.5}$ observations in preliminary and comparison experiments were generally small and fluctuated without distinct trends and leading factors which make it hard to correct. On the other hand, the main purpose of this study was to propose strategies of method selection for fine scale $PM_{2.5}$ mapping using crowdsourced monitoring, as the three methods we compared were performed with the same sampling dataset, the uncertainty in measurements associated with monitors and RH may cause a limited influence on the method comparison results. We therefore did not correct the measurements in this study. We agree with the reviewer that more efforts are needed in crowdsourced measurements correction and uncertainty analysis in air pollution concentration mapping at high resolution for accurate exposure assessment. Sentences about this were added in the section of Discussion.

Replaced lines 30 on Page 3 and lines 1 –3 on Page 4 by:

To ensure the data quality of this monitor, we placed 115 laser air quality monitors in the same environment and continuously observed them for one week during each of the four seasons. If the relative error between the observation of one monitor and the average observations of the other monitors exceeded 5%, this monitor fell into disuse. This procedure was conducted both indoors and outdoors. Subsequently, 86 monitors with rather stable performance and a small difference between each observation remained. In addition, we randomly selected 30 portable laser air quality monitors to compare with the national monitoring instruments to further guarantee the reliability of the sampling data. First, for ease of operation, three national air quality monitoring stations were selected. Second, for each station, 10 monitors were observed next to the national monitoring instrument (~15 metres above the ground in the study area) from 8:00 to 20:00 on December 20–22, 2015 and from 8:00 to 20:00 on December 29–31, 2015. The weather on December 20–22 was overcast with patchy drizzle and light rain at times, and the relative humidity (RH) ranged from 77% to 94%, while the weather on December 29–31 was cloudy with some sunshine and a RH that ranged from 38%–67%.

The scatter plots and descriptive statistics of the valid hourly average $PM_{2.5}$ concentrations from the laser air quality monitors and the national monitoring instruments were presented in Fig. 1c and Fig. 1d. The hourly average $PM_{2.5}$ concentrations for two types of instruments generally showed good agreement with a correlation coefficient R2 of 0.89 on December 20–22 and 0.90 on December 29–31. The root-mean-square-errors (RMSE) for the former time period was lower than the RMSE for the latter time period (5.63 µg m$^{-3}$ vs. 5.94 µg m$^{-3}$), while the mean relative error (MRE) was higher than the MRE for the latter

time period (6.37% vs. 3.82%). The latter time period demonstrated a smaller difference in hourly average PM2.5 concentrations between laser air quality monitors and the national monitoring instruments with mean values and standard deviations (SD) of 72.99±16.45 µg m$^{-3}$ vs. 71.89±15.28 µg m$^{-3}$ and 129.93±18.33 µg m$^{-3}$ vs. 129.33±17.50 µg m$^{-3}$.

[Figure]

Figure 1: Principle and accuracy of measurement instrument. Y and X are laser air quality monitors and national monitoring instruments, respectively. The black dots, blue dots and red dots indicate PM$_{2.5}$ observations with relative error of <10%, 10%–20%, and >20%, respectively, between two instruments. The black dotted line and red dotted line are the 1:1 line and 1:1.2 line as references.

Replaced lines 4 –13 on Page 9 by:

The hourly PM$_{2.5}$ concentrations between crowdsourced sampling sites and national monitoring stations were rather different; this difference varied as the official air quality level changed. The crowdsourced PM$_{2.5}$ concentrations were substantially larger than the national concentrations in Period 1 (light-polluted) and slightly lower in Period 2 (heavy-polluted). One possible reason is that the national monitoring stations in the study area were installed on the roofs of mid-rise buildings (i.e., ~15 m) with ventilation and spaciousness, while crowdsourced sampling was conducted on the real ground (i.e., ~2 m). The change in the major pollution sources and meteorological conditions in the study

area may contribute to the difference between two periods; the major contribution of local sources, especially the vehicle emission and the very high RH (95%–98%) during the light-polluted period, may cause the accumulation of $PM_{2.5}$ near the ground; and the sources of long-range transport of regional pollution during the heavy-polluted period can increase the concentration of $PM_{2.5}$ on the upper layer. This finding suggests that the air pollution exposure risk may remain relatively high for the public on the ground in some urban microenvironments, even when official air pollution levels are "Good" and "Moderate" and sensitive groups should consider reducing some outdoor activities. The results confirm the necessity of developing real-ground high-density crowdsourced $PM_{2.5}$ monitoring networks. Although the low-cost sensor and the use of optical particle detection of monitors in sampling may cause inaccuracies in measurements, we have attempted to minimise the uncertainty by disusing the relatively inaccurate monitors (MRE>5%) used in preliminary indoor and outdoor experiments. Comparison experiments between laser air quality monitors and the national monitoring instruments were also conducted at the same positions and heights for two time slots; the weather conditions and air quality scenarios of the two time slots were similar to the two sampling periods (i.e., overcast with light rain, RH≥76%: December 20–22 vs. Period 1; cloudy with sunshine, RH≤67%: December 29–31 vs. Period 2). The relatively good agreement between the hourly $PM_{2.5}$ concentrations of laser monitors and those of national instruments had guaranteed the reliability of sampling data to a certain extent. The relative humidity may have slightly influenced the crowdsourced $PM_{2.5}$ concentrations in the light-polluted period since December 20–22 yielded a slightly lower $R^2$ and RMSE than those of December 29–31 but a higher MRE than that of December 29–31. However, the relative error of $PM_{2.5}$ observations in preliminary and comparison experiments were generally small and fluctuated without distinct trends and leading factors. During the following procedure of mapping method selection, three methods were performed with the same dataset, which caused a limited influence of uncertainty in measurements on the method comparison results; therefore, we did not correct the measurements in this study. However, more efforts are needed in crowdsourced measurements correction and uncertainty analysis in air pollution concentration mapping at high resolution for accurate exposure assessment in the future.

Page 5 lines 17-18: Meteorological data with a spatial resolution of roughly 0.4 sites per 100 km2 (wind speed, atmospheric pressure, relative humidity, temperature) that-I think it might be clearer to just list the number of stations you had in total over your sampling area.

Response: changed the sentence as the reviewer suggested:

Meteorological data including wind speed, atmospheric pressure, relative humidity, and temperature of 107 sites in and around the sampling area, which may affect the dispersion of $PM_{2.5}$, were also obtained.

Section 2.1.2: I'm not clear how this data is crowdsourced can you please include more

information about how each monitor got to each monitoring point.

Response: In order to explore the spatial variation of PM$_{2.5}$ concentration for various urban microenvironment and compare with the national air quality measurements, the crowdsourced monitoring is assumed to cover a certain number of areas. However, persuading the general public in these areas to continuously observe and upload PM$_{2.5}$ concentrations during their activities of daily living through a designed study is difficult. We therefore employed a batch of volunteers to model their behaviours on the general public's behaviour and and simultaneously collect data. Due to the difficulties in implementing the campaign (e.g. the financial burdens of volunteers' recruitment and the extensive investment of time and efforts for technology part and procedures to ensuring data quality), we only carried out this sampling for two short sampling periods. We believe it is a preliminary practice of crowdsourced monitoring and can be further developed and improved by progress in low-cost wearable air quality monitors and automatic processing techniques of crowdsourced data. We rewrote the **2.1.3 Sampling and data processing,** sentences about this were added.

Replaced lines 16 –25 on Page 4 by:

Sampling was performed in two time periods in the winter of 2015 to examine the effect of air quality grades on the mapping results. The first period fell between 8:00 and 12:00 on December 24. In this period, the official air pollution levels were "Good" and "Moderate" (i.e., Period 1, light-polluted period). The weather was overcast with occasional rain or drizzle, and the relative humidity (RH) ranged from 95% to 98%. The second period extended between 14:00 and 18:00 on December 25, when an orange warning signal of haze (i.e., official air pollution level was "Heavily Polluted") was released by the Changsha Meteorology Bureau (i.e., Period 2, heavy-polluted period). The weather was cloudy with some sunshine, and the RH ranged from 39%–43%.

Before sampling started, every volunteer received one monitor and went to the corresponding area. At each potential monitoring site, the volunteer lifted the monitor (~2 metres above the ground) and held it for at least 60 seconds to measure the PM2.5 concentration. The observations were uploaded twice to four times hourly using a smart phone application (App) that we developed. The geographic coordinates of the sampling sites were also uploaded. For each hour, we eliminated the sampling sites with less than three observations. The valid observations were then averaged at each site. As some volunteers quit after the sampling of the first period, the sampling sites in period 2 were concentrated in the central study area. A total of 179-208 samples were successfully collected at each hour in Period 1, and 105-118 samples were successfully collected in Period 2. The official observations at 10 national monitoring stations in the study area were also obtained (China Environmental Monitoring Center, CEMC:

http://106.37.208.233:20035/) and averaged for comparison purposes.

Replaced lines 25 –30 on Page 8 and lines 1 –3 on Page 9 by:

The number of sampling sites were 18 and 10 per 100 km$^2$ for Period 1 and Period 2, respectively. These data comprise a considerable improvement compared with a density of approximately 0.015 sites per 100 km$^2$ in the national air quality monitoring network in China. As expected, crowdsourced $PM_{2.5}$ measurements demonstrated detailed spatial variation among urban microenvironments, and these variations can hardly be disclosed by sparse national air quality monitoring stations. This finding suggests that crowdsourced sampling can effectively improve the density of $PM_{2.5}$ monitoring at a rather low monetary cost and can be supportive of the short-term air pollution exposure assessment for epidemiologic studies at a fine scale. To explore the spatial variation in the $PM_{2.5}$ concentration for various urban microenvironments and compare with the national air quality measurements, the crowdsourced monitoring is assumed to cover a certain number of areas. However, persuading the general public in these areas to continuously observe and upload $PM_{2.5}$ concentrations during their activities of daily living through a designed study is difficult. We employed a batch of volunteers to model their behaviours on the general public's behaviour and simultaneously collect data. This approach is a preliminary practice of crowdsourced monitoring and can be further developed and improved in the long-term exposure assessment at the fine scale in the future with the progress in low-cost wearable air quality monitors and automatic processing techniques of crowdsourced data.

Page 6 lines 19-21: Is this the highest and lowest one-hour average from a single site and single monitor? Why are these and the times they occurred important?
Line 20: These what? Averages?
Line 20-21: I don't know what the numbers in parenthesis are please clarify
Lines 25 and 26: Is there more traffic at noon than at morning rush hour? Also is the average concentration at the different hours significantly different?

Response: this is the highest and lowest one-hour average for all crowdsourced sampling sites, we tended to present the rather large range of crowdsourced $PM_{2.5}$ observations. We rewrote the confusing sentence.
"These" are the maximum and minimum values of crowdsourced $PM_{2.5}$ concentrations. Rewrote the confusing sentence.
"numbers in parenthesis" are the mean values and SD of the $PM_{2.5}$ concentrations and the maximum and minimum values of national monitoring $PM_{2.5}$ concentrations. Rewrote the confusing sentence.
There may be more traffic at morning than at noon, the higher $PM_{2.5}$ concentrations at noon than at morning may relate to the peaked cooking emission of stir-fry at noon. The average concentration at the different hours in the same period is rather close. Sentence about these were added or replaced.

Replaced lines 16 –26 on Page 6 by:

Table 3 shows the descriptive statistics of hourly $PM_{2.5}$ concentrations for the crowdsourced sampling sites and the national monitoring stations. Generally, the statistics differed. For Period 1, the mean values and SD of the $PM_{2.5}$ concentrations for the crowdsourced sampling sites ranged from (69.67±18.81) to (76.45±14.55) µg m$^{-3}$. These values were substantially higher than those for the national monitoring stations (i.e., (36.9±10.97) – (41.2±8.68) µg m$^{-3}$). The maximum and minimum values of crowdsourced $PM_{2.5}$ concentrations were higher than the national values. However, the mean values and SD of $PM_{2.5}$ concentrations of the crowdsourced sites are lower than those of the national stations in period 2. The former values ranged from (162.72±15.96) µg m$^{-3}$ to (171.89±21.5) µg m$^{-3}$, while the latter values ranged from (177.8±16.91) µg m$^{-3}$ to (188.3±22.4) µg m$^{-3}$. Although the minimum values of crowdsourced $PM_{2.5}$ concentrations were also lower than those of the national stations, the maximum values were higher. The average $PM_{2.5}$ concentrations of Period 2 were substantially higher than those of Period 1, and the highest values occurred when traffic and emissions from cooking had peaked (i.e., 12:00 and 18:00) for both periods.

Figure 3. Does each of these points represent a single monitor? Why are they fewer monitors during period 2?

Response: each point represents a single sampling site with no less than three observations for each hour. Because some volunteers quit after the sampling of the first period, sampling sits in period 2 were mainly concentrated in the central study area and thus fewer than in period 1. Sentences about this were added in section **2.1.3 Sampling and data processing** as mentioned before.

Line 13: I don't understand what you are comparing that increased. What is the first set of numbers versus the seconded set of numbers?
Line 14: What do you mean significant and steady decrease? Decreased by hour by the same amount?

Response: the first set of numbers are the average validation $R^2$ of OK with the smallest number of training sites for each hour; the seconded set of numbers are the average validation $R^2$ of OK with the largest number of training sites for each hour; rewrote these confusing sentences.
We thank the reviewer for pointing out the fault in line 14; rewrote the confusing sentence.

Replaced lines 7 –29 on Page 7 by:

The box plots of Fig. 4 show the variation in the hold-out validation $R^2$ for the three mapping approaches in relation to the number of training sites. The average and standard deviation of the RMSE and MRE between the observed concentration and predicted

concentration of $PM_{2.5}$ in the hold-out validation were presented in the Supporting Information (Table S3–S4). The average values and variability ranges of $R^2$ for OK, LUR and RK were positively associated with an increase in the number of training sites. RK performed best in Period 2 and at 8:00 and 12:00 of Period 1 with training sites less than ~100. The LUR demonstrated the poorest performance for both periods of the models tested.

For Period 1, the $PM_{2.5}$ estimating accuracy was generally highest at 9:00 and lowest at 12:00. The average validation $R^2$ ranges for different number training sites of OK at 8:00, 9:00, 10:00, 11:00 and 12:00 were 0.58–0.72, 0.56–0.78, 0.51–0.82, 0.47–0.71, and 0.24–0.48, respectively. Compared with OK, the accuracy of LUR was substantially lower. The ranges were 0.26–0.55, 0.29–0.54, 0.16–0.40, 0.16–0.36, and 0.24–0.34. The average $R^2$ for RK were weakly smaller than OK at 9:00, 10:00, and 11:00 with ranges of 0. 59–0. 69, 0. 50–0.66, and 0. 48–0.60, respectively. The average $R^2$ of RK at 8:00 and 12:00 were higher than OK when less than ~100 sampling sites were divided into training datasets (8:00: 0.65–0.69 vs. 0.58–0.68; 12:00: 0.40–0.44 vs. 0.24–0.41). For Period 2, the validation $R^2$ from high to low followed the sequence RK > OK > LUR. The average validation $R^2$ for a different number of training sites of OK were considerably lower in Period 1. The ranges at 14:00, 15:00, 16:00, 17:00 and 18:00 were 0.25–0.49, 0.34–0.50, 0.40–0.59, 0.27–0.39, and 0.18–0.27, respectively. The average $R^2$ of LUR were even lower; the lowest values were 0.08, 0.07, 0.15, 0.06, and 0.04, and the highest values were 0.22, 0.25, 0.42, 0.22, and 0.16, respectively. Combining OK and LUR, the performance of RK improved with an average R2 that ranged from 0.43, 0.44, 0.43, 0.36, and 0.32 to 0.60, 0.68, 0.52, 0.54, and 0.57.

Page 7 Line 30: Since readers can see the individual R2 on the figure it may be easier to digest if you just include an average or range instead of so many lists of numbers.
Page 8 Line 5: I read this paragraph a couple times and I'm still a bit confused which method performs the best. Can you add a summary sentence at the end just stating the conclusion? Or reorganize more clearly.

Response: we thank the reviewer for the suggestion. These confusing sentences were rewritten.

Replaced lines 30 –33 on Page 7 and lines 1 –5 on Page 8 by:

Fig. 5 shows scatterplots of holdout-validation results with 90% training sites. For Period 1, the lowest total R2 of OK and the highest total R2 of OK were 0.46 for 12:00 and 0.82 for 10:00 (Fig. 5a), respectively, while R2 of RK were lower with the range of 0.44–0.68 (Fig. 5c); they were both higher than the LUR (0.29–0.53, Fig. 5b). Correspondingly, the RMSE and MRE from low to high were OK (5.95–10.36; 6.80%–9.91%) < RK (8.23–10.92; 9.80%–11.91%) < LUR (10.68–13.16; 12.91%–14.97%). For Period 2, however, the RK presented the highest accuracy with a R2 that ranged from 0.45 (17:00) to 0.66 (14:00) (Fig. 5f). The OK ranked second ($R^2$: 0.27–0.54, Fig. 5d), while the LUR achieved the poorest performance ($R^2$: 0.06–0.36, Fig. 5e).

Section 3.3: Can you clarify: did you use 90% training sites for only the sensor measurements and then only 90% of the reference stations? As far as I could tell previously you only used withholding from the sensor data and didn't evaluate the models using the reference data?

Response: the method that performed best with 90% training sites was chosen as the mapping method. Using this method, the spatial distributions of the PM$_{2.5}$ concentration for each hour were estimated with all samples. Spatial distributions of PM$_{2.5}$ concentration for each hour with measurements of 10 national monitoring stations were estimated using the same method for comparison.
We rewrote these confusing sentences in section **2.3 PM$_{2.5}$ concentration mapping** and corresponding results.

Replaced lines 10 –13 on Page 6 by:

The method that performed best with 90% training sites was chosen as the mapping method. Using this method, the spatial distributions of the PM$_{2.5}$ concentration for each hour were estimated with all samples. In this study, nearest neighbour distances between two sampling sites ranged from 15 to 60 metres for Period 1 and 54 to 98 metres for Period 2. Considering the resolutions of the potential predictors, 100 metres was used as the mapping grid size. The spatial distributions of the PM$_{2.5}$ concentration for each hour with measurements of 10 national monitoring stations were estimated using the same method for comparison.

Page 8 line 9: "Significant difference can be found between two sources," what do you mean?
Page 8 Line 13: What do you mean three-step growth?

Response: It means that the hourly PM$_{2.5}$ concentrations for the crowdsourced sampling sites and the national monitoring stations were rater different.
It means gradual growth.
We rewrote these confusing sentences and replaced lines 7 –20 on Page 8 by:

Fig. 6a and Fig. 6b reveal the spatial distributions of OK interpreted PM$_{2.5}$ concentrations for Period 1 from the crowdsourced sampling sites and the national monitoring stations, respectively. Fig. 6c and Fig. 6d demonstrate the spatial distributions of the RK estimated PM$_{2.5}$ concentrations for Period 2. The crowdsourced hourly PM$_{2.5}$ concentration maps demonstrate more detailed intra-urban variations than the national monitoring maps, especially for Period 1.
For Period 1, crowdsourced PM$_{2.5}$ concentrations generally increased from south-east to north-west with multiple hot spots. In the central and south regions of the study area, areas with a larger number of factories that experience a relatively higher PM$_{2.5}$ concentration than other areas. The national monitoring PM$_{2.5}$ concentrations, however,

were less than 55 µg m$^{-3}$ with limited spatial variation. For Period 2, with the exception of 14:00, the national monitoring PM$_{2.5}$ concentration maps showed high-east and low-west patterns. PM$_{2.5}$ concentrations of central Yuelu district were rather low (<175 µg m$^{-3}$). Crowdsourced PM$_{2.5}$ concentrations demonstrate extensive cold spots of PM$_{2.5}$ concentrations in southern Changsha County and the southern Kaifu district, while southern Yuelu and western Tianxin with a high-density of factories and roads were hot spots of PM$_{2.5}$ concentration.

Page 8 Line 15: I don't understand based on the figure it seems like there are almost no factories and roads in the top left corner but that is where most of the pollution is.

Response: relatively high concentration in the northwest corner of the study area with few factories in Period 1 may be attributed to the dust deposition from construction activities promoted by a high RH in this newly developed zone. Sentences addressing this were added in the section of Discussion.

As the crowdsourced PM$_{2.5}$ concentrations maps revealed, areas with a larger number of factories and high-density of roads experienced relatively higher PM$_{2.5}$ concentrations, while areas with high levels of green vegetation cover had lower PM$_{2.5}$ concentrations. The relatively high concentration in the northwest corner of the study area with few factories in Period 1 may be attributed to the dust deposition from construction activities promoted by a high RH in this newly developed zone. This finding suggests that optimising the distribution of land use may improve the air quality to some extent and strengthening the control of local emission may be the primary way to reduce pollution in the light-polluted period. As the urban air quality grade has an important effect on the spatial distribution of samples (spatial autocorrelation, and heterogeneity), which may also be affected by sample size, the mechanism for this influence is somewhat equivocal and needs further research.

Page 8 Line 30-Page 9 Line 3: I think you need to mention though the limitations of low-cost monitors and the inaccuracies in these measurements compared to federal methods. Page 9 Line 10: It seems likely the low-cost sensors may have been saturated at the high concentrations and this may have led to the difference between the sensors and the reference methods.

Response: we thank the reviewer for the suggestion. Sentences about this were added.

Replaced lines 4 –13 on Page 9 by:

The hourly PM$_{2.5}$ concentrations between crowdsourced sampling sites and national monitoring stations were rather different; this difference varied as the official air quality level changed. The crowdsourced PM$_{2.5}$ concentrations were substantially larger than the national concentrations in Period 1 (light-polluted)

and slightly lower in Period 2 (heavy-polluted). One possible reason is that the national monitoring stations in the study area were installed on the roofs of mid-rise buildings (i.e., ~15 m) with ventilation and spaciousness, while crowdsourced sampling was conducted on the real ground (i.e., ~2 m). The change in the major pollution sources and meteorological conditions in the study area may contribute to the difference between two periods; the major contribution of local sources, especially the vehicle emission and the very high RH (95%–98%) during the light-polluted period, may cause the accumulation of PM$_{2.5}$ near the ground; and the sources of long-range transport of regional pollution during the heavy-polluted period can increase the concentration of PM$_{2.5}$ on the upper layer. This finding suggests that the air pollution exposure risk may remain relatively high for the public on the ground in some urban microenvironments, even when official air pollution levels are "Good" and "Moderate" and sensitive groups should consider reducing some outdoor activities. The results confirm the necessity of developing real-ground high-density crowdsourced PM$_{2.5}$ monitoring networks. Although the low-cost sensor and the use of optical particle detection of monitors in sampling may cause inaccuracies in measurements, we have attempted to minimise the uncertainty by disusing the relatively inaccurate monitors (MRE>5%) used in preliminary indoor and outdoor experiments. Comparison experiments between laser air quality monitors and the national monitoring instruments were also conducted at the same positions and heights for two time slots; the weather conditions and air quality scenarios of the two time slots were similar to the two sampling periods (i.e., overcast with light rain, RH≥76%: December 20–22 vs. Period 1; cloudy with sunshine, RH≤67%: December 29–31 vs. Period 2). The relatively good agreement between the hourly PM2.5 concentrations of laser monitors and those of national instruments had guaranteed the reliability of sampling data to a certain extent. The relative humidity may have slightly influenced the crowdsourced PM$_{2.5}$ concentrations in the light-polluted period since December 20–22 yielded a slightly lower $R^2$ and RMSE than those of December 29–31 but a higher MRE than that of December 29–31. However, the relative error of PM$_{2.5}$ observations in preliminary and comparison experiments were generally small and fluctuated without distinct trends and leading factors. During the following procedure of mapping method selection, three methods were performed with the same dataset, which caused a limited influence of uncertainty in measurements on the method comparison results; therefore, we did not correct the measurements in this study. However, more efforts are needed in crowdsourced measurements correction and uncertainty analysis in air pollution concentration mapping at high resolution for accurate exposure assessment in the future.

Technical corrections:
Suggest rewording the title for clarity, possibly: Strategies of method selection for Fine Scale PM2.5 mapping in an intra-urban area using crowdsourced monitoring

Response: changed as the reviewer suggested.

Fine particulate matter (particulate matter singular remove s)

Response: corrected.

Line 9: to "the" public – there are a number of grammatical errors throughout the text and I have not had a chance to identify them all in this review. Please review for grammar.
Page 6 Line 20 ug/m3 formatting
Page 7 Line 4: Remove "had" assuming you are talking about this work where the sites experienced extreme PM

Response: we thank the reviewer for the suggestion. Meanwhile, this manuscript was edited for proper English language, grammar, punctuation, spelling, and overall style by one or more of the highly qualified native English speaking editors at American Journal Experts. The certificate may be verified at www.aje.com/certificate with a certificate verification key of E57E-12C6-6B0F-0300-999B.

---

## Author Comment (AC4) · 4 Mar 2019

Answer to Referee #1

We thank the referee for his/her very careful review, and his/her constructive suggestions. In the following, we answer his/her specific questions. In order to facilitate the reference to the questions and proposed changes, we use the following color coding:

Color coding:
reviewer comment
our answer
proposed change in manuscript
* * *
Xu et al describe measurements and spatial modeling of PM2.5. Measurements were conducted with hand-held optical particle monitors. The spatial modeling compared multiple methods: ordinary kriging, universal kriging, and land use regression. The paper suffers from several critical flaws and is not publishable in its current form. Below I outline five major problems with the manuscript.

Major Issue #1: I do not know what the authors mean by a "crowdsourced" data collection. The authors seem to define crowdsourcing in lines 27-28 of page 2, but "Crowdsourcing activities based on informal social networks and web 2.0 technologies that allowed citizens themselves to produce geospatial data among others" seems more like corporate jargon than a useful explanation of crowdsourcing.

Response: we rewrote this sentence as:

Crowdsourced monitoring that enables citizens to produce geospatial data is constantly growing and shows considerable potential (Heipke, 2010). Large and diverse groups of people who lack formal training can easily describe their environments with a mobile phone or smart phone and upload data via informal social networks and web technology.

The sampling approach seems to be short-term saturation sampling - many volunteers simultaneously sampled at predetermined locations. This sampling approach does not fit my personal notion of crowdsourcing, which would be a more informal data collection leveraging people's normal movements throughout the day. Sending an army of students to collect data in an organized fashion seems less like "crowdsourcing" and more like a sampling campaign. In that sense, this study has little distinction from the large literature on distributed air quality sampling.
What would be the value or longer-term viability of this or a similar sampling approach? This paper focuses on two short sampling periods of a few hours each, so the data are unlikely representative of long-term spatial patterns. Do the authors expect to deploy an army of distributed samplers on a semi-regular basis in order to build up a dataset capable of reproducing longer-term trends? Or to send out volunteers daily to make daily maps? I

don't see how the "crowdsourced" aspect of this adds value or novelty; instead it seems like crowdsourcing is being used as a buzzword.

Response: In order to explore the spatial variation of PM$_{2.5}$ concentration for various urban microenvironment and compare with the national air quality measurements, the crowdsourced monitoring is assumed to cover a certain number of areas. However, persuading the general public in these areas to continuously observe and upload PM$_{2.5}$ concentrations during their activities of daily living through a designed study is difficult. We therefore employed a batch of volunteers to model their behaviours on the general public's behaviour and and simultaneously collect data. Due to the difficulties in implementing the campaign (e.g. the financial burdens of volunteers' recruitment and the extensive investment of time and efforts for technology part and procedures to ensuring data quality), we only carried out this sampling for two short sampling periods. We agree with the reviewer that this sampling approach is not a complete crowdsourcing activity, but we believe it is a preliminary practice of crowdsourced monitoring and can be further developed and improved by progress in low-cost wearable air quality monitors and automatic processing techniques of crowdsourced data. We rewrote the **2.1.3 Sampling and data processing,** sentences about this were added. Meanwhile, it has to be claimed that the focus of this study is the 'strategies of methods' (e.g. LUR, OK) selection under crowdsourced monitoring data rather than the discoveries of long-term spatial patterns of PM$_{2.5}$ concentrations. Sentences about this were claimed in the **Introduction**.

Replaced lines 16 –25 on Page 4 by:

Sampling was performed in two time periods in the winter of 2015 to examine the effect of air quality grades on the mapping results. The first period fell between 8:00 and 12:00 on December 24. In this period, the official air pollution levels were "Good" and "Moderate" (i.e., Period 1, light-polluted period). The weather was overcast with occasional rain or drizzle, and the relative humidity (RH) ranged from 95% to 98%. The second period extended between 14:00 and 18:00 on December 25, when an orange warning signal of haze (i.e., official air pollution level was "Heavily Polluted") was released by the Changsha Meteorology Bureau (i.e., Period 2, heavy-polluted period). The weather was cloudy with some sunshine, and the RH ranged from 39%–43%.
Before sampling started, every volunteer received one monitor and went to the corresponding area. At each potential monitoring site, the volunteer lifted the monitor (~2 metres above the ground) and held it for at least 60 seconds to measure the PM2.5 concentration. The observations were uploaded twice to four times hourly using a smart phone application (App) that we developed. The geographic coordinates of the sampling sites were also uploaded. For each hour, we eliminated the sampling sites with less than three observations. The valid observations were then averaged at each site. As some volunteers quit after the

sampling of the first period, the sampling sites in period 2 were concentrated in the central study area. A total of 179-208 samples were successfully collected at each hour in Period 1, and 105-118 samples were successfully collected in Period 2. The official observations at 10 national monitoring stations in the study area were also obtained (China Environmental Monitoring Center, CEMC: http://106.37.208.233:20035/) and averaged for comparison purposes.

Replaced lines 25 –30 on Page 8 and lines 1 –3 on Page 9 by:

The number of sampling sites were 18 and 10 per 100 km$^2$ for Period 1 and Period 2, respectively. These data comprise a considerable improvement compared with a density of approximately 0.015 sites per 100 km$^2$ in the national air quality monitoring network in China. As expected, crowdsourced PM$_{2.5}$ measurements demonstrated detailed spatial variation among urban microenvironments, and these variations can hardly be disclosed by sparse national air quality monitoring stations. This finding suggests that crowdsourced sampling can effectively improve the density of PM$_{2.5}$ monitoring at a rather low monetary cost and can be supportive of the short-term air pollution exposure assessment for epidemiologic studies at a fine scale. To explore the spatial variation in the PM$_{2.5}$ concentration for various urban microenvironments and compare with the national air quality measurements, the crowdsourced monitoring is assumed to cover a certain number of areas. However, persuading the general public in these areas to continuously observe and upload PM$_{2.5}$ concentrations during their activities of daily living through a designed study is difficult. We employed a batch of volunteers to model their behaviours on the general public's behaviour and simultaneously collect data. This approach is a preliminary practice of crowdsourced monitoring and can be further developed and improved in the long-term exposure assessment at the fine scale in the future with the progress in low-cost wearable air quality monitors and automatic processing techniques of crowdsourced data.

Major Issue #2: Data quality. Figure 1 shows one short-term comparison between the handheld PM monitors and the regulatory monitors. While there is generally good agreement, there is a fair amount of scatter among the handheld monitors. This scatter is to be expected given the low cost and the use of optical particle detection. However, the authors do not address how uncertainty in the measurements potentially impacts the mapping. Nor do they seem to account for uncertainty in the measurements or make any efforts to correct the measurements (e.g., based on hygroscopic growth).

Response: Although the low-cost sensor and the use of optical particle detection of monitors used in sampling may cause inaccuracies in measurements, we have tried to minimum the uncertainty by disusing the relatively inaccurate monitors (MRE>5%) through preliminary indoor and outdoor experiments. Comparison experiments between laser air quality monitors and the national monitoring instruments were also conducted at the same positions and heights for two time slots; the weather conditions and air quality scenarios of the two time slots were

similar to the two sampling periods. In the previous version of the manuscript, we thought one comparison result is enough to demonstrate the reliability of sampling data to a certain extent, we thank the reviewer for pointing out the inadequacies. Sentences about data quality and Figure 1d were added.

Replaced lines 23 –30 on Page 3 and lines 1 –3 on Page 4 by:

The portable laser air quality monitor SDL307 (produced by NOVA FITNESS Co., Ltd.) is employed to perform sampling. The monitor manual can be downloaded from http://www.inovafitness.com/index.html. This monitor can be conveniently carried with a total size of 25×34×14 cm (Fig. 1a). According to the test report provided by the Center for Building Environment Test at Tsinghua University, the maximum relative error of this monitor is ±20% compared with a regulatory monitor in the 20–1000 µg m$^{-3}$ range and has a resolution of 0.1 µg m$^{-3}$. The concentration of particulate matter is measured using the light-scattering method (Fig. 1b). The monitor contains a special laser module, and the signals are recorded by a photoelectric receptor when particulate matter passes through laser light. The count and size of particulate matter are then analysed by a microcomputer after the signals are amplified and converted. Their mass concentrations are calculated based on the conversion factor between the light-scattering method and the tapered element oscillating microbalance technology.

To ensure the data quality of this monitor, we placed 115 laser air quality monitors in the same environment and continuously observed them for one week during each of the four seasons. If the relative error between the observation of one monitor and the average observations of the other monitors exceeded 5%, this monitor fell into disuse. This procedure was conducted both indoors and outdoors. Subsequently, 86 monitors with rather stable performance and a small difference between each observation remained. In addition, we randomly selected 30 portable laser air quality monitors to compare with the national monitoring instruments to further guarantee the reliability of the sampling data. First, for ease of operation, three national air quality monitoring stations were selected. Second, for each station, 10 monitors were observed next to the national monitoring instrument (~15 metres above the ground in the study area) from 8:00 to 20:00 on December 20–22, 2015 and from 8:00 to 20:00 on December 29–31, 2015. The weather on December 20–22 was overcast with patchy drizzle and light rain at times, and the relative humidity (RH) ranged from 77% to 94%, while the weather on December 29–31 was cloudy with some sunshine and a RH that ranged from 38%–67%.

The scatter plots and descriptive statistics of the valid hourly average PM$_{2.5}$ concentrations from the laser air quality monitors and the national monitoring instruments were presented in Fig. 1c and Fig. 1d. The hourly average PM$_{2.5}$ concentrations for two types of instruments generally showed good agreement with a correlation coefficient R2 of 0.89 on December 20–22 and 0.90 on December 29–31. The root-mean-square-errors (RMSE) for the former time

period was lower than the RMSE for the latter time period (5.63 µg m⁻³ vs. 5.94 µg m⁻³), while the mean relative error (MRE) was higher than the MRE for the latter time period (6.37% vs. 3.82%). The latter time period demonstrated a smaller difference in hourly average PM2.5 concentrations between laser air quality monitors and the national monitoring instruments with mean values and standard deviations (SD) of 72.99±16.45 µg m⁻³ vs. 71.89±15.28 µg m⁻³ and 129.93±18.33 µg m⁻³ vs. 129.33±17.50 µg m⁻³.

[Figure]

Figure 1: Principle and accuracy of measurement instrument. Y and X are laser air quality monitors and national monitoring instruments, respectively. The black dots, blue dots and red dots indicate PM2.5 observations with relative error of <10%, 10%–20%, and >20%, respectively, between two instruments. The black dotted line and red dotted line are the 1:1 line and 1:1.2 line as references.

On the one hand, the relative error of PM2.5 observations in preliminary and comparison experiments were generally small and fluctuated without distinct trends and leading factors which make it hard to correct. On the other hand, the main purpose of this study was to propose strategies of method selection for fine scale PM2.5 mapping using crowdsourced monitoring, as the three methods we compared were performed with the same sampling dataset, the uncertainty in measurements may cause a limited influence on the method comparison results. We therefore did not correct the measurements in this study. We agree with the

reviewer that more efforts are needed in crowdsourced measurements correction and uncertainty analysis in air pollution concentration mapping at high resolution for accurate exposure assessment. Sentences about this were added in the section of Discussion.

Table 3 and section 3.1 - the crowdsourced data read higher PM than the regulatory data. The authors have not convinced me that this is not an artifact of the sensors they have chosen. During some hours there is significant difference between the mean "crowdsourced" PM and the mean regulatory PM. Since the overall spatial extent of the two sampling domains (regulatory and crowdsourced) is roughly similar, I would expect similar mean concentrations from each dataset.
Line 30 on page 8 calls the national monitoring sites "inaccurate." I am not familiar with regulatory measurement policies in China, but if they are anything like the US and Europe, the accuracy standard is high. The spatial pattern derived from these few monitors may be erroneous, but the specific measurements are accurate.

Response: we agree with the reviewer that the instruments of national monitoring stations are more accurate and reliable than the potable air pollution monitors, and that is the reason why we conducted the comparison experiments between laser air quality monitors and the national monitoring instruments at the same positions and heights before and after the crowdsourcing sampling. The point we intend to make is that the crowdsourced $PM_{2.5}$ measurements demonstrated obvious spatial variation between urban microenvironments, and these variations can hardly be disclosed by sparse national air quality monitoring stations. In fact, the overall spatial extent of the two sampling domains (regulatory and crowdsourced) is relatively different according to the Figure 3, the color rendering may be the reason why the difference is not so significant. We therefore summarized the statistics of $PM_{2.5}$ concentration. The difference of hourly $PM_{2.5}$ concentrations between the two types of instruments in sampling campaign is possibly because of the different sampling heights and the change of the major pollution sources in the study area. We thank the reviewer for pointing out this issue. We rewrote these confusing sentences and replaced lines 25 –30 on Page 8 and lines 1 –13 on Page 9 by:

The number of sampling sites were 18 and 10 per 100 km² for Period 1 and Period 2, respectively. These data comprise a considerable improvement compared with a density of approximately 0.015 sites per 100 km² in the national air quality monitoring network in China. As expected, crowdsourced $PM_{2.5}$ measurements demonstrated detailed spatial variation among urban microenvironments, and these variations can hardly be disclosed by sparse national air quality monitoring stations. This finding suggests that crowdsourced sampling can effectively improve the density of $PM_{2.5}$ monitoring at a rather low monetary cost and can be supportive of the short-term air pollution exposure assessment for epidemiologic studies at a fine scale. To explore the spatial variation in the $PM_{2.5}$ concentration

for various urban microenvironments and compare with the national air quality measurements, the crowdsourced monitoring is assumed to cover a certain number of areas. However, persuading the general public in these areas to continuously observe and upload $PM_{2.5}$ concentrations during their activities of daily living through a designed study is difficult. We employed a batch of volunteers to model their behaviours on the general public's behaviour and simultaneously collect data. This approach is a preliminary practice of crowdsourced monitoring and can be further developed and improved in the long-term exposure assessment at the fine scale in the future with the progress in low-cost wearable air quality monitors and automatic processing techniques of crowdsourced data.

The hourly $PM_{2.5}$ concentrations between crowdsourced sampling sites and national monitoring stations were rather different; this difference varied as the official air quality level changed. The crowdsourced $PM_{2.5}$ concentrations were substantially larger than the national concentrations in Period 1 (light-polluted) and slightly lower in Period 2 (heavy-polluted). One possible reason is that the national monitoring stations in the study area were installed on the roofs of mid-rise buildings (i.e., ~15 m) with ventilation and spaciousness, while crowdsourced sampling was conducted on the real ground (i.e., ~2 m). The change in the major pollution sources and meteorological conditions in the study area may contribute to the difference between two periods; the major contribution of local sources, especially the vehicle emission and the very high RH (95%–98%) during the light-polluted period, may cause the accumulation of $PM_{2.5}$ near the ground; and the sources of long-range transport of regional pollution during the heavy-polluted period can increase the concentration of $PM_{2.5}$ on the upper layer. This finding suggests that the air pollution exposure risk may remain relatively high for the public on the ground in some urban microenvironments, even when official air pollution levels are "Good" and "Moderate" and sensitive groups should consider reducing some outdoor activities. The results confirm the necessity of developing real-ground high-density crowdsourced $PM_{2.5}$ monitoring networks. Although the low-cost sensor and the use of optical particle detection of monitors in sampling may cause inaccuracies in measurements, we have attempted to minimise the uncertainty by disusing the relatively inaccurate monitors (MRE>5%) used in preliminary indoor and outdoor experiments. Comparison experiments between laser air quality monitors and the national monitoring instruments were also conducted at the same positions and heights for two time slots; the weather conditions and air quality scenarios of the two time slots were similar to the two sampling periods (i.e., overcast with light rain, RH≥76%: December 20–22 vs. Period 1; cloudy with sunshine, RH≤67%: December 29–31 vs. Period 2). The relatively good agreement between the hourly PM2.5 concentrations of laser monitors and those of national instruments had guaranteed the reliability of sampling data to a certain extent. The relative humidity may have slightly influenced the crowdsourced $PM_{2.5}$ concentrations in the light-polluted period since December 20–22 yielded a slightly lower $R^2$ and RMSE than those of

December 29–31 but a higher MRE than that of December 29–31. However, the relative error of $PM_{2.5}$ observations in preliminary and comparison experiments were generally small and fluctuated without distinct trends and leading factors. During the following procedure of mapping method selection, three methods were performed with the same dataset, which caused a limited influence of uncertainty in measurements on the method comparison results; therefore, we did not correct the measurements in this study. However, more efforts are needed in crowdsourced measurements correction and uncertainty analysis in air pollution concentration mapping at high resolution for accurate exposure assessment in the future.

Major Issue #3: Site selection and sampling strategy. The description of the sampling strategy is insufficient. Were all samplers deployed simultaneously at all sites in Table 1? How were the sampling times defined and chosen? What are significant differences between period 1 and period 2?
Table 1 - A better description of each type of site is needed. For example, Dust surfaces seem to be defined as "dust surfaces," which is not helpful to readers. What qualifies as a dust surface? Some entries in this table have "A" and "U". What do those designations mean?

Response: Table 1 presents the rules to determine the potential $PM_{2.5}$ sampling sites that we would like to monitor. At each potential monitoring site, the volunteer lifted the monitor (~2 metres above the ground) and held it for at least 60 seconds to measure the $PM_{2.5}$ concentration. Those observations were uploaded twice to four times hourly using a smart phone application (App) that we developed. So technically, samplers were not deployed simultaneously. For each hour, we eliminated the sampling sites with less than three observations. The valid observations were then averaged at each site. Meanwhile, because some volunteers quit after the sampling of the first period, the final number of samples for each hour were different. Compare with period 1, sampling sits in period 2 were mainly concentrated in the central study area.
Dust surfaces refer to natural and artificial bare surfaces with vegetation cover less than 10% that are easy to produce atmospheric particulate matters. "U" and "A" are subset of the set of potential $PM_{2.5}$ sampling sites and the subset of the union of supporting data. More details for supporting data of site selection were added.

Replaced lines 9-14 on page 4 by:

To ensure that the sampling sites exhibit a relatively even and typical distribution for different urban microenvironments (i.e., residential community, building site, school, and park), a series of rules were designed to determine the potential $PM_{2.5}$ sampling sites based on the distribution of potential emission sources (refer to Table 1). The data that support the sampling design consist of important points of interest (POI), dust surfaces, and main road networks. POI data includes industrial parks, enterprises, factories, depots, hospitals, schools, and parks. Dust surfaces refer to natural and artificial bare surfaces

with vegetation that covers less than 10%, which easily produce atmospheric particulate matter, such as construction sites, stacked substance, and natural bare land. These data were collected from the Information Center of Land and Resources of Hunan Province. More than three observations of $PM_{2.5}$ concentrations are required every hour for each potential sampling site to improve the reliability of the sampling data. Given that the number of laser air quality monitors and the distance that a volunteer can walk in one hour are limited, only 2–4 sites can be set in the area in which a monitor can cover during the sampling. Therefore, a total of 208 potential $PM_{2.5}$ sampling sites were selected. The centre of each area covered by a monitor were numbered in sequence (i.e., 1–86). The monitors were also numbered and labelled.

Table 1 was changed as:

**Table 1.** Rules for potential $PM_{2.5}$ sampling sites selection.

| Code | Type | N | Rules |
|------|------|---|-------|
| 1 | Vertex point | 5 | U1[a] = {X[c] \| X∈(Vertex point of the boundary of sampling area ∩ Landmark)}. |
| 2 | Industrial park | 28 | A2[b] = {X \| X∈((Industrial park ∪ (Metal & cement & power industrial factories agglomeration)) – High-tech industrial park)};
 U2 = {X \| X has the largest number of factories within its 100 m buffer zone AND X∈A2}. |
| 3 | Dust surface | 13 | A3 = {X \| X∈(POI ∩ Dust surface) AND area of dust surface ranks in the top 4 of each district};
 U3 = {X \| Distance between X＞200 m AND X∈A3}. |
| 4 | Depot | 16 | U4 = {X \| X∈(Coach station ∩ Railway station)}. |
| 5 | Scenic area | 27 | A5 = {X \| X∈((Park – Neighbourhood park) ∩ well-known scenic area)};
 U5 = {X \| Distance between X＞200 m AND X∈A5}. |
| 6 | Hospital | 11 | A6 = {X \| X∈(Hospital ranks in the top 3 of each district ∪ Children's hospital ∪ Respiratory special hospital)};
 U6 = {X \| Distance between X＞200 m AND X∈A6}. |
| 7 | Residential area | 12 | A7 = {X \| Distance between X and U1＜200 m OR Distance between X and U3＜200 m, X∈ Residential area };
 U7 = {X \| Distance between X＞200 m AND X∈A7}. |
| 8 | School | 15 | U8 = {X \| Distance between X and U1＜200 m OR Distance between X and U3＜200 m, X∈School, in order of priority: Kindergarten＞Primary＞Secondary＞Universities)}. |
| 9 | Commercial area | 9 | U9 = {X \| X is the building with the highest population density, X∈Commercial area}. |
| 10 | Other important POI | 8 | U10 = {X \| X∈(Corresponding sampling site of national monitoring station ∪ Background site ∪ Museum)}. |
| 11 | Road | 56 | A11 = {X \| X∈(Junction of (Expressway ∪ Main road))};
 U11 = {X \| X is 50/100 metres away from A11 OR X∈A11}. |

| 12 | Supplementary point | 3 | U12 = {X \| X∈ POI where four neighbouring grids have no site}. |

[a]U$_i$ (*i*=1, 2, …): *i*th subset of the set of potential PM$_{2.5}$ sampling sites.

[b]A$_i$ (*i*=1, 2, …): *i*th subset of the union of supporting data.

[c]X: element belongs to the set.

Major Issue #4: Modeling and interpretation. The modeling aspect of this paper is not novel. Since the sampling method seems to be a straightforward saturation sampling campaign, using the resulting data to build spatial models is not a novel contribution. Numerous papers have already done this for PM2.5, as noted by the authors.

One main conclusion seems to be that the modeling approaches work. This is not all that novel - it is more a statistical finding than an atmospheric measurement technique. Numerous papers have shown that LUR and kriging models can be fit to spatially distributed measurements.

Another conclusion is that the models work better when provided with more training sites. Again, this seems like an obvious outcome, especially for the kriging approaches.

Response: we agree with the reviewer that the modeling methods themselves were not novel, but the main purpose of this study was to propose strategies of method selection for fine scale PM$_{2.5}$ mapping using crowdsourced monitoring, not to develop a new model. As we mentioned in the manuscript, the optimal mapping method for conventional sparse fixed measurements may not be suitable for this new high-density monitoring way, and the results were rather different from previous studies as we expected. OK interpolation performs best under conditions with non-peak traffic situation in light-polluted period, while the RK modelling can perform better in heavy-polluted period and for conditions with the peak traffic and relatively few sampling sites (less than ~100) in light-polluted period. Additionally, this study for the first time found and pointed out that the LUR model demonstrates limited ability in estimating PM$_{2.5}$ concentrations at very fine spatial and temporal scale which challenges the traditional point on LUR model's good performance in air pollution mapping.

We would not call "modeling approaches work "and "the models work better when provided with more training sites" the main conclusions of this study, but we do admit that some sentences of the abstract in previous version of the manuscript may make this wrong impression on the readers. We thank the reviewer for pointing out this issue.

We changed the abstract as:

Fine particulate matter (PM$_{2.5}$) is of great concern to the public due to its significant risk to human health. Numerous methods have been developed to estimate spatial PM$_{2.5}$ concentrations in unobserved locations due to the sparse number of fixed monitoring stations. Due to an increase in low-cost sensing for air

pollution monitoring, crowdsourced monitoring of fine exposure control has been gradually introduced into cities. However, the optimal mapping method for conventional sparse fixed measurements may not be suitable for this new high-density monitoring approach. This study presents a crowdsourced sampling campaign and strategies of method selection for hundred metre-scale level PM$_{2.5}$ mapping in an intra-urban area of China. During this process, PM$_{2.5}$ concentrations were measured by laser air quality monitors and uploaded by a group of volunteers via their smart phone applications during two periods. Three extensively employed modelling methods (ordinary kriging (OK), land use regression (LUR), and regression kriging (RK) were adopted to evaluate the performance. An interesting finding is that PM$_{2.5}$ concentrations in micro-environments significantly varied in the intra-urban area. These local PM$_{2.5}$ variations can be effectively identified by crowdsourced sampling rather than national air quality monitoring stations (light-polluted period: $(69.67\pm18.81)$ – $(76.45\pm14.55)$ μg m$^{-3}$ vs. $(36.9\pm10.97)$ – $(41.2\pm8.68)$ μg m$^{-3}$; heavy-polluted period: $(162.72\pm15.96)$ – $(171.89\pm21.5)$ μg m$^{-3}$ vs. $(177.8\pm16.91)$ – $(188.3\pm22.4)$ μg m$^{-3}$).. The selection of models for fine scale PM$_{2.5}$ concentration mapping should be adjusted according to the changing sampling and pollution circumstances. Generally, OK interpolation performs best in conditions with non-peak traffic situations during a light-polluted period (hold-out validation R$^2$: 0.47–0.82), while the RK modelling can perform better during the heavy-polluted period (0.32–0.68) and in conditions with peak traffic and relatively few sampling sites (less than ~100) during the light-polluted period (0.40–0.69). Additionally, the LUR model demonstrates limited ability in estimating PM$_{2.5}$ concentrations on very fine spatial and temporal scales in this study (0.04–0.55), which challenges the traditional point about the good performance of the LUR model for air pollution mapping. This method selection strategy provides empirical evidence for the best method selection for PM$_{2.5}$ mapping using crowdsourced monitoring, and this provides a promising way to reduce the exposure risks for individuals in their daily life.

A more relevant analysis would be to evaluate if the models (and measurements) make physical dense. In Figure 5 there is a PM hotspot in the northwestern part of the domain on Day 1 and in the center of the domain on Day 2. Do these hotspots make sense given the distribution of sources and the climatology?

Response: we thank the reviewer for the suggestion and sentences addressing this were added in the section of Discussion. The PM hotspot in the northwestern part of the domain on Day 1 may be attributed to the dust deposition from construction activities promoted by a high RH in this newly developed zone, while the PM hotspot in the center of the domain on Day 2 may relate to the larger number of factories and high-density of roads.

As the crowdsourced PM$_{2.5}$ concentrations maps revealed, areas with a larger

number of factories and high-density of roads experienced relatively higher PM$_{2.5}$ concentrations, while areas with high levels of green vegetation cover had lower PM$_{2.5}$ concentrations. The relatively high concentration in the northwest corner of the study area with few factories in Period 1 may be attributed to the dust deposition from construction activities promoted by a high RH in this newly developed zone.

Major Issue #5: The paper needs a thorough review and edit for English grammar. There are many grammar errors (too many to count or enumerate here), and in other places the language is hard to follow.

Response: we thank the reviewer for the suggestion. Meanwhile, this manuscript was edited for proper English language, grammar, punctuation, spelling, and overall style by one or more of the highly qualified native English speaking editors at American Journal Experts. The certificate may be verified at www.aje.com/certificate with a certificate verification key of E57E-12C6-6B0F-0300-999B.

---

## Author Comment (AC5) · 5 Mar 2019

Thank you very much for your thoughtful comments and suggestions. Our responses to your comments are in attached file. Please see attached file.

Please also note the supplement to this comment: https://www.atmos-meas-tech-discuss.net/amt-2018-402/amt-2018-402-AC5-supplement.pdf

---

## Author Response (AR1)

**Author's Response**

**No.**: amt-2018-402     Submitted on 16 Nov 2018

**Title**: Strategies of Method Selection for Fine Scale PM$_{2.5}$ Mapping in Intra-Urban Area Under Crowdsourcing Monitoring

**Authors**: Shan Xu, Bin Zou, Yan Lin, Xiuge Zhao, Shenxin Li, and Chenxia Hu

We would like to take this opportunity to thank the editor and the reviewers for their very careful review and constructive suggestions. We have revised the manuscript based on the suggestions and provided justifications where appropriate. Given below is a summary of the responses and revisions. Meanwhile, we misspelled the first name of one author (Hu) in the last version of the manuscript, we corrected it in the new version.

**Answer to Referee #1**

Xu et al describe measurements and spatial modeling of PM2.5. Measurements were conducted with hand-held optical particle monitors. The spatial modeling compared multiple methods: ordinary kriging, universal kriging, and land use regression. The paper suffers from several critical flaws and is not publishable in its current form. Below I outline five major problems with the manuscript.

Major Issue #1: I do not know what the authors mean by a "crowdsourced" data collection. The authors seem to define crowdsourcing in lines 27-28 of page 2, but "Crowdsourcing activities based on informal social networks and web 2.0 technologies that allowed citizens themselves to produce geospatial data among others" seems more like corporate jargon than a useful explanation of crowdsourcing.

**Response**: we rewrote this sentence as, please see the section of **Introduction, Page 1, Lines 25-28**.

The sampling approach seems to be short-term saturation sampling - many volunteers simultaneously sampled at predetermined locations. This sampling approach does not fit my personal notion of crowdsourcing, which would be a more informal data collection leveraging people's normal movements throughout the day. Sending an army of students to collect data in an organized fashion seems less like "crowdsourcing" and more like a sampling campaign. In that sense, this study has little distinction from the large literature on distributed air quality sampling.

What would be the value or longer-term viability of this or a similar sampling approach? This paper focuses on two short sampling periods of a few hours each, so the data are unlikely representative of long-term spatial patterns. Do the authors expect to deploy an army of distributed samplers on a semi-regular basis in order to build up a dataset capable of reproducing longer-term trends? Or to send out volunteers daily to make daily maps? I don't see how the "crowdsourced" aspect of this adds value or novelty; instead it seems like crowdsourcing is being used as a buzzword.

**Response**: In order to explore the spatial variation of $PM_{2.5}$ concentration for various urban microenvironment and compare with the national air quality measurements, the crowdsourced monitoring is assumed to cover a certain number of areas. However, persuading the general public in these areas to continuously observe and upload $PM_{2.5}$ concentrations during their activities of daily living through a designed study is difficult. We therefore employed a batch of volunteers to model their behaviours on the general public's behaviour and simultaneously collect data. Due to the difficulties in implementing the campaign (e.g. the financial burdens of volunteers' recruitment and the extensive investment of time and efforts for technology part and procedures to ensuring data quality), we only carried out this sampling for two short sampling periods. We agree with the reviewer that this sampling approach is not a complete crowdsourcing activity, but we believe it is a preliminary practice of crowdsourced monitoring and can be further developed and improved by progress in low-cost wearable air quality monitors and automatic processing techniques of crowdsourced data. We rewrote the **2.1.3 Sampling and data processing and Discussion,** sentences about this were added. Please see **Page 5, Lines 5-20** and **Page 9, Lines 15-27.** Meanwhile, it has to be claimed that the focus of this study is the 'strategies of methods' (e.g. LUR, OK) selection under crowdsourced monitoring data rather than the discoveries of long-term spatial patterns of $PM_{2.5}$ concentrations. Sentences about this were claimed in the **Introduction**. Please see **Page 3, Lines 6-11**.

Major Issue #2: Data quality. Figure 1 shows one short-term comparison between the handheld PM monitors and the regulatory monitors. While there is generally good agreement, there is a fair amount of scatter among the handheld monitors. This scatter is to be expected given the low cost and the use of optical particle detection. However, the authors do not address how uncertainty in the measurements potentially impacts the mapping. Nor

do they seem to account for uncertainty in the measurements or make any efforts to correct the measurements (e.g., based on hygroscopic growth).

**Response**: Although the low-cost sensor and the use of optical particle detection of monitors used in sampling may cause inaccuracies in measurements, we have tried to minimum the uncertainty by disusing the relatively inaccurate monitors (MRE>5%) through preliminary indoor and outdoor experiments. Comparison experiments between laser air quality monitors and the national monitoring instruments were also conducted at the same positions and heights for two time slots; the weather conditions and air quality scenarios of the two time slots were similar to the two sampling periods. In the previous version of the manuscript, we thought one comparison result is enough to demonstrate the reliability of sampling data to a certain extent, we thank the reviewer for pointing out the inadequacies. Sentences about data quality and **Figure 1d** were added. Please see the section of **Data and methods, Page 3, Lines 22-30 and Page 4, Lines 1-19**.

On the one hand, the relative error of $PM_{2.5}$ observations in preliminary and comparison experiments were generally small and fluctuated without distinct trends and leading factors which make it hard to correct. On the other hand, the main purpose of this study was to propose strategies of method selection for fine scale $PM_{2.5}$ mapping using crowdsourced monitoring, as the three methods we compared were performed with the same sampling dataset, the uncertainty in measurements may cause a limited influence on the method comparison results. We therefore did not correct the measurements in this study. We agree with the reviewer that more efforts are needed in crowdsourced measurements correction and uncertainty analysis in air pollution concentration mapping at high resolution for accurate exposure assessment. Sentences about this were added in the section of **Discussion, Page 9, Lines 28-32 and Page 10, Lines 1-22**.

Table 3 and section 3.1 - the crowdsourced data read higher PM than the regulatory data. The authors have not convinced me that this is not an artifact of the sensors they have chosen. During some hours there is significant difference between the mean "crowdsourced" PM and the mean regulatory PM. Since the overall spatial extent of the two sampling domains (regulatory and crowdsourced) is roughly similar, I would expect similar mean concentrations from each dataset.Line 30 on page 8 calls the national monitoring sites "inaccurate." I am not familiar with regulatory measurement policies in China, but if they are anything like the US and Europe, the

accuracy standard is high. The spatial pattern derived from these few monitors may be erroneous, but the specific measurements are accurate.

**Response**: we agree with the reviewer that the instruments of national monitoring stations are more accurate and reliable than the potable air pollution monitors, and that is the reason why we conducted the comparison experiments between laser air quality monitors and the national monitoring instruments at the same positions and heights before and after the crowdsourcing sampling. The point we intend to make is that the crowdsourced $PM_{2.5}$ measurements demonstrated obvious spatial variation between urban microenvironments, and these variations can hardly be disclosed by sparse national air quality monitoring stations. In fact, the overall spatial extent of the two sampling domains (regulatory and crowdsourced) is relatively different according to the Figure 3, the colour rendering may be the reason why the difference is not so significant. We therefore summarized the statistics of $PM_{2.5}$ concentration. The difference of hourly $PM_{2.5}$ concentrations between the two types of instruments in sampling campaign is possibly because of the different sampling heights and the change of the major pollution sources in the study area. We thank the reviewer for pointing out this issue. We rewrote these confusing sentences, please see the section of **Discussion, Page 9, Lines 28-32 and Page 10, Lines 1-22**.

Major Issue #3: Site selection and sampling strategy. The description of the sampling strategy is insufficient. Were all samplers deployed simultaneously at all sites in Table 1? How were the sampling times defined and chosen? What are significant differences between period 1 and period 2?

Table 1 - A better description of each type of site is needed. For example, Dust surfaces seem to be defined as "dust surfaces," which is not helpful to readers. What qualifies as a dust surface? Some entries in this table have "A" and "U". What do those designations mean?

**Response**: Table 1 presents the rules to determine the potential $PM_{2.5}$ sampling sites that we would like to monitor. At each potential monitoring site, the volunteer lifted the monitor (~2 metres above the ground) and held it for at least 60 seconds to measure the $PM_{2.5}$ concentration. Those observations were uploaded twice to four times hourly using a smart phone application (App) that we developed. So technically, samplers were not deployed simultaneously. For each hour, we eliminated the sampling sites with less than three observations. The valid observations were then averaged at each site. Meanwhile, because some volunteers quit after the sampling of the first period, the final number of samples for each hour were

different. Compare with period 1, sampling sits in period 2 were mainly concentrated in the central study area. In period 1, the official air pollution levels were "Good" and "Moderate", in period 2, the Changsha Meteorology Bureau released an Orange warning signal of haze (i.e. the official air pollution level was "Heavily Polluted").

5   Dust surfaces refer to natural and artificial bare surfaces with vegetation cover less than 10% that are easy to produce atmospheric particulate matters. "U" and "A" are subset of the set of potential $PM_{2.5}$ sampling sites and the subset of the union of supporting data. More details for supporting data of site selection were added.

We rewrote these confusing sentences of **Data and methods and Table 1,** please see **Page 4, Lines 25-**
10   **33 and Page 5, Lines 1-3.**

Major Issue #4: Modeling and interpretation. The modeling aspect of this paper is not novel. Since the sampling method seems to be a straightforward saturation sampling campaign, using the resulting data to build spatial models is not a novel contribution. Numerous papers have already done this for PM2.5, as noted by the authors.
15   One main conclusion seems to be that the modeling approaches work. This is not all that novel - it is more a statistical finding than an atmospheric measurement technique. Numerous papers have shown that LUR and kriging models can be fit to spatially distributed measurements. Another conclusion is that the models work better when provided with more training sites. Again, this seems like an obvious outcome, especially for the kriging approaches.

**Response**: we agree with the reviewer that the modeling methods themselves were not novel, but the
20   main purpose of this study was to propose strategies of method selection for fine scale $PM_{2.5}$ mapping using crowdsourced monitoring, not to develop a new model. As we mentioned in the manuscript, the optimal mapping method for conventional sparse fixed measurements may not be suitable for this new high-density monitoring way, and the results were rather different from previous studies as we expected. OK interpolation performs best under conditions with non-peak traffic situation in light-polluted period,
25   while the RK modelling can perform better in heavy-polluted period and for conditions with the peak traffic and relatively few sampling sites (less than ~100) in light-polluted period. Additionally, this study for the first time found and pointed out that the LUR model demonstrates limited ability in estimating $PM_{2.5}$ concentrations at very fine spatial and temporal scale which challenges the traditional point on LUR model's good performance in air pollution mapping.

We would not call "modeling approaches work "and "the models work better when provided with more training sites" the main conclusions of this study, but we do admit that some sentences of the abstract in previous version of the manuscript may make this wrong impression on the readers. We thank the reviewer for pointing out this issue. We rewrote the abstract, please see **Page 1, Lines 9-29.**

A more relevant analysis would be to evaluate if the models (and measurements) make physical dense. In Figure 5 there is a PM hotspot in the northwestern part of the domain on Day 1 and in the center of the domain on Day 2. Do these hotspots make sense given the distribution of sources and the climatology?

**Response**: we thank the reviewer for the suggestion and sentences addressing this were added in the

10  section of Discussion. The PM hotspot in the northwestern part of the domain on Day 1 may be attributed to the dust deposition from construction activities promoted by a high RH in this newly developed zone, while the PM hotspot in the center of the domain on Day 2 may relate to the larger number of factories and high-density of roads. Sentences about this were added in the section of **Discussion**, **Page 11, Lines 29-34.**

Major Issue #5: The paper needs a thorough review and edit for English grammar. There are many grammar errors (too many to count or enumerate here), and in other places the language is hard to follow.

**Response**: we thank the reviewer for the suggestion. Meanwhile, this manuscript was edited for proper English language, grammar, punctuation, spelling, and overall style by one or more of the highly qualified

20  native English speaking editors at American Journal Experts. The certificate may be verified at www.aje.com/certificate with a certificate verification key of E57E-12C6-6B0F-0300-999B.
* * *
**Answer to Referee #2**

General comments: Overall this is an interesting paper comparing methods for estimating spatial concentrations of PM2.5 using crowd sourced low-cost sensor measurements. I think it will be highly valuable for many researchers in the field interested in spatial variation. However, I think there is a lack of discussion of the limitations of lowcost optical particle sensors especially with the limited performance evaluation presented in this manuscript. I suggest major revisions for this paper. There are a number of places where the text is unclear and the authors should take care to thoroughly edit the next draft of this paper.

5    Specific comments:

Abstract: It's not clear what the different periods are referring to, morning versus afternoon? Line 19: I don't think that a range is the best statistic to show that 2 sets of numbers are "clearly different". Line 48: What do you mean by: "and a promising access to the prevention of exposure risks for individuals in their daily life."

**Response**: period 1 was between 8:00 and 12:00 on December 24; period 2 was between 14:00 and 18:00

10    on December 25. In period 1, the official air pollution levels were "Good" and "Moderate", in period 2, the Changsha Meteorology Bureau released an Orange warning signal of haze (i.e. the official air pollution level was "Heavily Polluted").

Line 19: The statistic of range was replaced by Mean±SD.

Line 48: It means that If individuals could consciously choose the location and time of their outdoor

15    activities based on detailed knowledge about the spatiotemporal variation in $PM_{2.5}$ concentration, then their health protection could be improved

We rewrote these confusing sentences of abstract, please see **Page 1, Lines 9-29.**

Page 3 line 24-25: What does "data consistency" mean? Can you please elaborate. Also, where do you get the

20    resolution data from? The manufacturer? Lab studies? Please cite.

**Response**: "data consistency" means the relative errors between monitors are rather small; resolution data came from the monitor manual provided by the manufacturer.

we rewrote these confusing sentences and more details about this monitor were added in the section of **Data and methods,** please see **Page 3, Lines 22-30**.

Page 3 Line 30: Why would you only select 30 monitors to collocate? Without the collocation data from the other monitors you have no idea what the bias is of the other measurements.

**Response**: In fact, the 86 portable laser air quality monitors we used in the sampling were selected from 115 monitors through preliminary indoor and outdoor experiments. The relative errors between each other were no larger than 5%, which guaranteed the reliability of sampling data of the other measurements to a certain extent. Sentences about this were added in **2.1.1 Measurement instrument, Page 4, Lines 1-4**. Under the circumstance that the national monitoring stations do not have enough room for more laser monitors to conduct the comparison experiments, we only randomly selected 30 monitors.

Section 2.2.1: Can you mention if these monitors or internal sensors are commercially available or have been evaluated in any other studies, etc. Oh, I see in the supplement they are SDL307 but I think this may be important to add to the text.

**Response**: added.

Page 3 Lines 28-29: This is confusing to me. I don't see K factors anywhere when I look at the figure. Please clarify this sentence and/or move the figure reference to a more appropriate location.

**Response**: made the changes as the reviewer suggested in the revised manuscript, please see **Page 3, Lines 26-30**.

Page 3 Line 30-Page 4 Line 3: I think the performance needs more discussion. How do the monitors compare to each other? If you are looking at spatial variability, bias/error between different monitors will be important. Were all monitors at the reference site for the same period? Is this 1-hr data shown in the plot or some other averaging time? Knowing the bias of individual monitors is very important because it will help determine at what threshold you can say there is likely spatial variation versus just bias in the sensor measurements. In addition, RH is known to significantly influence optical PM measurements. RH should be reported throughout. If RH is >75% during one of the periods (1,2, or comparison) this may be an issue. In addition, you have no data above ~100 ug/m3 but during your second period the concentrations are in the 170-180 range. I think it is important to know how the sensors perform at these high concentrations if you are going to try to draw conclusions. Has any previous work evaluated these sensors at high concentrations? You cannot assume that just because they work well from the 40-100 range they will work the same below and above that.

**Response**: Comparison experiments between laser air quality monitors and the national monitoring instruments were also conducted at the same positions and heights for two time slots; the weather

conditions (including RH) and air quality scenarios of the two time slots were similar to the two sampling periods. In the previous version of the manuscript, we thought one comparison result is enough to demonstrate the reliability of sampling data to a certain extent, we thank the reviewer for pointing out the inadequacies. Sentences about data quality and Figure 1d were added, please see the section of **Data and methods, Page 3, Lines 22-30 and Page 4, Lines 1-19**.

On the one hand, the relative error of $PM_{2.5}$ observations in preliminary and comparison experiments were generally small and fluctuated without distinct trends and leading factors which make it hard to correct. On the other hand, the main purpose of this study was to propose strategies of method selection for fine scale $PM_{2.5}$ mapping using crowdsourced monitoring, as the three methods we compared were performed with the same sampling dataset, the uncertainty in measurements associated with monitors and RH may cause a limited influence on the method comparison results. We therefore did not correct the measurements in this study. We agree with the reviewer that more efforts are needed in crowdsourced measurements correction and uncertainty analysis in air pollution concentration mapping at high resolution for accurate exposure assessment. Sentences about this were added in the section of **Discussion, Page 9, Lines 28-32 and Page 10, Lines 1-22**.

Page 5 lines 17-18: Meteorological data with a spatial resolution of roughly 0.4 sites per 100 km2 (wind speed, atmospheric pressure, relative humidity, temperature) that-I think it might be clearer to just list the number of stations you had in total over your sampling area.

**Response**: changed the sentence as the reviewer suggested, **Page 6, Lines 14-17**.

Section 2.1.2: I'm not clear how this data is crowdsourced can you please include more information about how each monitor got to each monitoring point.

**Response**: In order to explore the spatial variation of $PM_{2.5}$ concentration for various urban microenvironment and compare with the national air quality measurements, the crowdsourced monitoring is assumed to cover a certain number of areas. However, persuading the general public in these areas to continuously observe and upload $PM_{2.5}$ concentrations during their activities of daily living through a designed study is difficult. We therefore employed a batch of volunteers to model their behaviours on the general public's behaviour and simultaneously collect data. Due to the difficulties in implementing the

campaign (e.g. the financial burdens of volunteers' recruitment and the extensive investment of time and efforts for technology part and procedures to ensuring data quality), we only carried out this sampling for two short sampling periods. We believe it is a preliminary practice of crowdsourced monitoring and can be further developed and improved by progress in low-cost wearable air quality monitors and automatic processing techniques of crowdsourced data. We rewrote the **2.1.3 Sampling and data processing, Page 5, Lines 5-20**.

Sentences about this were added in the section of **Discussion, Page 9, Lines 15-27.**

Page 6 lines 19-21: Is this the highest and lowest one-hour average from a single site and single monitor? Why are these and the times they occurred important? Line 20: These what? Averages? Line 20-21: I don't know what the numbers in parenthesis are please clarify

Lines 25 and 26: Is there more traffic at noon than at morning rush hour? Also is the average concentration at the different hours significantly different?

**Response**: lines 19-21: this is the highest and lowest one-hour average for all crowdsourced sampling sites, we tended to present the rather large range of crowdsourced $PM_{2.5}$ observations.

Line 20: "These" are the maximum and minimum values of crowdsourced $PM_{2.5}$ concentrations. Rewrote the confusing sentence.

Line 20-21: "numbers in parenthesis" are the mean values and SD of the $PM_{2.5}$ concentrations and the maximum and minimum values of national monitoring $PM_{2.5}$ concentrations. Rewrote the confusing sentence.

Lines 25 and 26: There may be more traffic at morning than at noon, the higher $PM_{2.5}$ concentrations at noon than at morning may relate to the peaked cooking emission of stir-fry at noon. The average concentration at the different hours in the same period is rather close. Sentence about these were added or replaced.

Sentences about these were rewrote, please see the section of **Results, Page 7, Lines 14-23**.

Figure 3. Does each of these points represent a single monitor? Why are they fewer monitors during period 2?

**Response**: each point represents a single sampling site with no less than three observations for each hour. Because some volunteers quit after the sampling of the first period, sampling sits in period 2 were mainly

concentrated in the central study area and thus fewer than in period 1. Sentences about this were added in section **2.1.3 Sampling and data processing** as mentioned before, **Page 5, Lines 12-20.**

Line 13: I don't understand what you are comparing that increased. What is the first set of numbers versus the seconded set of numbers? Line 14: What do you mean significant and steady decrease? Decreased by hour by the same amount?

**Response**: the first set of numbers are the average validation $R^2$ of OK with the smallest number of training sites for each hour; the seconded set of numbers are the average validation $R^2$ of OK with the largest number of training sites for each hour; rewrote these confusing sentences.

We thank the reviewer for pointing out the fault in line 14; rewrote the confusing sentence.

Please see the section of **Results, Page 8, Lines 4-21**.

Page 7 Line 30: Since readers can see the individual R2 on the figure it may be easier to digest if you just include an average or range instead of so many lists of numbers.

Page 8 Line 5: I read this paragraph a couple times and I'm still a bit confused which method performs the best. Can you add a summary sentence at the end just stating the conclusion? Or reorganize more clearly.

**Response**: we thank the reviewer for the suggestion. These confusing sentences were rewritten.

Please see the section of **Results, Page 8, Lines 22-28**.

Section 3.3: Can you clarify: did you use 90% training sites for only the sensor measurements and then only 90% of the reference stations? As far as I could tell previously you only used withholding from the sensor data and didn't evaluate the models using the reference data?

**Response**: the method that performed best with 90% training sites was chosen as the mapping method. Using this method, the spatial distributions of the $PM_{2.5}$ concentration for each hour were estimated with all samples. Spatial distributions of $PM_{2.5}$ concentration for each hour with measurements of 10 national monitoring stations were estimated using the same method for comparison.

We rewrote these confusing sentences in section **2.3 PM₂.₅ concentration mapping** and corresponding results. Please see **Page 7, Lines 6-11**, **Page 8, Lines 30-31 and Page 9, Lines 1-2.**

Page 8 line 9: "Significant difference can be found between two sources," what do you mean? Page 8 Line 13: What do you mean three-step growth?

**Response**: line 9: It means that the hourly $PM_{2.5}$ concentrations for the crowdsourced sampling sites and the national monitoring stations were rather different.

Line 13: It means gradual growth.

We rewrote these confusing sentences. Please see **Page 9, Lines 3-10**

Page 8 Line 15: I don't understand based on the figure it seems like there are almost no factories and roads in the top left corner but that is where most of the pollution is.

**Response**: relatively high concentration in the northwest corner of the study area with few factories in Period 1 may be attributed to the dust deposition from construction activities promoted by a high RH in this newly developed zone. Sentences addressing this were added in the section of **Discussion**, **Page 11, Lines 29-34.**

Page 8 Line 30-Page 9 Line 3: I think you need to mention though the limitations of low-cost monitors and the inaccuracies in these measurements compared to federal methods. Page 9 Line 10: It seems likely the low-cost sensors may have been saturated at the high concentrations and this may have led to the difference between the sensors and the reference methods.

**Response**: we thank the reviewer for the suggestion. Sentences about this were added. Please see the section of **Discussion, Page 9, Lines 28-32 and Page 10, Lines 1-22**.

Technical corrections: Suggest rewording the title for clarity, possibly: Strategies of method selection for Fine Scale PM2.5 mapping in an intra-urban area using crowdsourced monitoring

**Response**: changed as the reviewer suggested.

Fine particulate matter (particulate matter singular remove s)
**Response**: corrected.

Line 9: to "the" public – there are a number of grammatical errors throughout the text and I have not had a chance to identify them all in this review. Please review for grammar.

Page 6 Line 20 ug/m3 formatting

5 Page 7 Line 4: Remove "had" assuming you are talking about this work where the sites experienced extreme PM

**Response**: we thank the reviewer for the suggestion. Meanwhile, this manuscript was edited for proper English language, grammar, punctuation, spelling, and overall style by one or more of the highly qualified native English speaking editors at American Journal Experts. The certificate may be verified at www.aje.com/certificate with a certificate verification key of E57E-12C6-6B0F-0300-999B.

**Answer to Referee #3**

In this manuscript, the authors presented strategies of method selection for efficiently and effectively PM2.5
15 concentration mapping with increasing training sites based on a crowdsourcing sampling campaign. This study found that Ordinary Kriging (OK) interpolation performed best under conditions with non-peak traffic situation in lightpolluted period, the Universal Kriging (UK) modeling performed better for conditions with the peak traffic and relatively few sampling sites in heavy-polluted period, and the Land Use Regression (LUR) model demonstrated limited ability in the estimation PM2.5 concentrations at very fine scale. Overall, the the manuscript
20 is well-written and scientifically sounds good, and can be accepted after minor revision.

The authors should really redefine all acronyms in conclusions: : :Conclusions should broadly read as if the reader hadn't read the rest of the paper. Thus, the authors reintroduce everything, including hypothesis and research plan.

**Response**: we thank the reviewer for the suggestion. Rewrote the section of Conclusions and redefine all acronyms, please see **Page 12, Line 8-12.**

**Answer to Referee #4**

This study used in situ PM2.5 measured by portable laser sir quality monitors to replace traditional PM2.5 data collected by ground monitoring stations or derived from remote sensing images and developed a new

5  hybrid (land use regression plus geostatistical) method to map PM2.5 concentrations in an urban area. Generally, this manuscript is well organized and clearly written, even though a few of sentences need to be rephrased and more details need to be supplemented. I recommend the editor to accept this manuscript after a minor or moderate revision.

The authors developed a hybrid model in which the deterministic component of the PM2.5 concentration

10  was fitted by LUR and the stochastic component (i.e. residual) was interpolated by kriging. Thus this is a typical LUR based REGRESSION kriging but not universal kriging. Please see Liu et al. (2018). Incorrectly naming the method is my biggest concern for the manuscript.

Liu, Y. et al., 2018. Improve ground-level PM2.5 concentration mapping using a random forests-based geostatistical approach. Environmental Pollution, 235, 272-282.

15  **Response**: the naming of this method followed Mercer et al. (Atmospheric Environment 2011). They proposed a 2-step approach in which simple kriging is applied to the residuals from LUR. This approach is similar but not identical to UK. Thus, we agree with the reviewer that the Regression Kriging is more appropriate and thank him/her for the suggestion. We implemented the changes in the revised manuscript.

20  Mercer, L. D., Szpiro, A. A., Sheppard, L., Lindström, J., Adar, S. D., Allen, R. W., Avol, EL., Oron, A. P., Larson, T., Liu, L. J., and Kaufman, J. D.: Comparing universal kriging and land-use regression for predicting concentrations of gaseous oxides of nitrogen (NOx) for the multi-ethnic study of atherosclerosis and air pollution (MESA Air), Atmos Environ, 45, 4412–4420, doi:10.1016/j.atmosenv.2011.05.043, 2011.

I am afraid that the Abstract from line 16 to 27 is not clear for a new reader especially who has not read the Method section. What do the "Period 1" and "Period 2" represent?

**Response**: "Period 1" and "Period 2" represent the light-polluted period and heavy-polluted period. We rewrote the Abstract, please see **Page 1, Lines 9-29.**

(Page 2, line 19) The authors should cite Liu et al. (2018) that is a typical study combining two technologies to estimate PM2.5 concentrations.

**Response**: Liu et al. (2018) adopted a random forests-based regression kriging approach which integrates recent advancements of machine learning with conventional kriging methods in geostatistics. We thank the reviewer for the suggestion and cited this article in the revised manuscript.

In the Measurement Instrument section, the authors may add more details for their portable air quality monitors, e.g. the company producing the equipment and other practical uses of the portable monitor.

**Response**: more details were added as the reviewer asked, please see **Page 3, Lines 22-30.**

(Page 4, lines 13-20). The sentences here are unclear and the authors may need to rewrite them. "Sampling was carried out in two time periods in the winter of 2015…" I am wondering whether the authors can provide a specific time periods (e.g. from November 1 to December 31) to replace "the winter". "The second period was between 14:00 and 18:00, when Orange warning signals of haze were released by Changsha Meteorology Bureau…" I guess Orange warning signal was not released every day, but from your last sentence "The first period was between 8:00 and 12:00, representing a light-polluted period" it seems the Orange warning signal is released every afternoon. So please make it clear whether you measured PM2.5 concentrations during the two time slots all days or only Orange days. Additional, I suggest using "time slots" to replace "time periods". The "period" may be used for the days when you collected the PM2.5 concentration samples.

**Response**: In fact, due to the difficulties in implementing the campaign (e.g. the financial burdens of volunteers' recruitment and the extensive investment of time and efforts for technology part and procedures to ensuring data quality), we only carried out this sampling between 8:00 and 12:00 on December 24 and 14:00 and 18:00 on December 25. In the first period, the official air pollution levels were "Good" and "Moderate", in the second period, the Changsha Meteorology Bureau released an

Orange warning signal of haze (i.e. the official air pollution level was "Heavily Polluted"). We rewrote these confusing sentences, please see **Page 5, Lines 5-20.**

(Page 4, line 20). "The official observations at 10 national monitoring sites stations."

**Response**: corrected.

(Page 6, lines 21-22) "Clearly, the average PM2.5 concentrations of Period 2 were two times higher than those of Period 1…" I wonder why the authors emphasized "two times" higher here. It gave me a deep impression that "two times" implied something, but I have not seen any explanation for the "two times" in the following text. I would simply say: the average PM2.5 concentrations of Period 2 were much higher than …

**Response**: we thank the reviewer for the suggestion and rewrote this sentence.

(Page 9, lines 1-10) I cannot accept the authors' discussion in this paragraph whatsoever. Compared with the authors' cheaper potable air pollution monitors, I more trust instruments from national monitoring stations. "This suggests the inconvenient truth (what a strong word! It is just a possible.) that the exposure risk remains relatively high for the public when official air pollution levels are "Good" and "Moderate" and this risk …" I completely understand what the authors intend to express, but if the government intentionally falsified the air quality data, it was more likely to lower the heavy- rather than light-pollution data. I thought of another possibility: the authors' portable monitors were not sensitive for the low PM2.5 concentrations and are proneto be saturated in the heavy-pollution days. In that case, it will also get the result the authors showed in the manuscript. The authors intended to emphasize that the large error (difference) on PM2.5 concentrations over the city is due to the relatively small number of national monitoring stations and thus their method using portable monitors to collect PM2.5 data is useful. However, based on the authors' statement, large differences on PM2.5 concentrations have existed even if concentrations are measured by the instruments of the national monitoring stations and the portable equipments of the authors at the same location.

**Response**: we agree with the reviewer that the instruments of national monitoring stations are more accurate and reliable than the potable air pollution monitors, and that is the reason why we conducted the comparison experiments between laser air quality monitors and the national monitoring instruments at the same positions and heights before and after the crowdsourcing sampling. The point we intend to make is that the crowdsourced $PM_{2.5}$ measurements demonstrated obvious spatial variation between urban microenvironments, and these variations can hardly be disclosed by sparse national air quality monitoring stations. The difference of hourly $PM_{2.5}$ concentrations between the two types of instruments in sampling campaign is possibly because of the different sampling heights and the change of the major pollution sources in the study area. We thank the reviewer for pointing out this issue. We rewrote these confusing sentences, please see the section of **Discussion, Page 9, Lines 28-32 and Page 10, Lines 1-22**.

I suggest the authors cautiously using some very strong adjectives and adverbs, such as clearly, significantly, tremendous, etc. (Abstract, line 25) "This method selection strategy provides solid experimental evidence for method selection of …" I will say "this study provides empirical evidence for …" Although generally clear for me, it is better to further polish the English of this manuscript, especially in the Results and Discussion sections.

**Response**: we thank the reviewer for the suggestion. Meanwhile, this manuscript was edited for proper English language, grammar, punctuation, spelling, and overall style by one or more of the highly qualified native English speaking editors at American Journal Experts. The certificate may be verified at www.aje.com/certificate with a certificate verification key of E57E-12C6-6B0F-0300-999B.

**Relevant changes made in the manuscript:**

[revised manuscript text omitted]

---

## Referee Report (RR1)

After major revisions I am still not convinced that this paper and the supporting research meet the standards of AMT. I think there are a number of major flaws the authors must address in order to make their paper suitable for publication.

In this revision of the paper I feel the authors have revealed a potential fatal flaw of their study:

Page 23 line 12-15: This is very surprising that volunteers just looked at their sensors 3 times in an hour and reported the value to the app. So the concentrations are not hourly averages from each monitor. This makes me question the usefulness of this study. How often does the screen update with the concentration? Is it a 1-minute average? Rolling average of the past hour? Something else? If you pulled 3 random minutes of data from an hour you would expect they might be significantly different than the average for the full hour.

Were all the performance metrics calculated in the same way just reading 3 random minutes off the screen? If not the performance metrics (Figure 1 and +/- 20% reported by Tsinghua) are not really good indicators for the expected accuracy of this work. I think the implications of this sampling methodology on the results need to be discussed.

See Zheng et al. 2018 for discussion of averaging time and low-cost sensor performance: https://www.atmos-meas-tech.net/11/4823/2018/

My other major concern is the overestimate by the sensors during period 1, and the over estimate of the sensors during period 2. During one of the evaluations periods the RH was 94% at maximum while during one of their sampling campaigns the RH was reported from 95% to 98%. Much previous work has shown that nephelometer type devices greatly over estimate during high RH events and in a nonlinear way (including: https://www.atmos-meas-tech.net/11/4823/2018/). There is no evaluation data at the RH that occurred during this project and they present no figures showing the relationship between RH and sensor error and no past work showing how this sensor performs at high RH. Maybe having accurate sensor readings is not important for the main goal of this paper (generating different models) but they have tried to draw a number of large conclusions based on the higher average from the sensor data (people should stay indoors even when government monitors say it is clean out, pollution at road level is higher). I have a similar concern during the second period where the monitors underestimate since they have presented no evaluation data and included no references that show that the sensor provides a linear response above 160 ug/m3. Low-cost sensors may show non-linear responses (http://www.aaqr.org/article/detail/AAQR-17-10-OA-0418) and it is very important to evaluate them under the conditions they will be evaluated under. Any supporting literature the authors can find about the monitor they used or the internal sensing component may support their findings but currently the authors have not cited any previous work with their monitor.

Additional comments:

Figure 1. It's not clear to me what is on the right scatter plot versus the left? Does each plot show the data from all three stations? How was the data recorded for these tests? Since for the experiment it was read off the screen is this also what was done for these? What is the averaging

time for the points shown on this plot. It would be helpful to label them as c and d so you can reference them in the text.

Page 22 line 4: please define small difference.

Abstract line 20: Not sure what all the numbers are in the parenthesis

Abstract Line 16: suggest mentioning how many hours each period was.

Abstract line 22: I don't think you can make any generalizations based on this dataset suggest: "During this project", OK interpolation performs... or something similar

Page 20 line 24: What do you mean by "fine" exposure control?

Page 21: Line 9 and 10: suggest including a citation about underestimation/mis calculation of risk

Page 21: Lines 22-27: These results from Tsinghua are super important since you are reporting data up to 260 ug/m3 but you only have a comparison range up to 160 ug/m3. Please include a citation or a figure in the supplement. Is this an ambient comparison or in the lab?

Page 21 Lines 29-30: Is this conversion factor calculated by you or by the company?

Page 21 line 25: At what averaging time is this +/- 20%

Page 27 line 32: Suggest citing previous literature that has seen significant gradients from 2 to 15 m.

Page 25 Line 15: I don't think it's helpful/scientific to state "Generally, the statistics differed". I think this is not needed

Page 27 Line 20-22: To explore the spatial variation in the... This sentence is unclear I don't understand any point you are trying to make

Page 28 line 14: You can't say this "guaranteed" their reliability

Page 28: Lines 4-7: I think this is too large of a finding based on the results you have presented here

---

## Author Response (AR2)

**Author's Response**

**No.**: amt-2018-402     Submitted on 16 Nov 2018

**Title**: Strategies of Method Selection for Fine Scale PM$_{2.5}$ Mapping in Intra-Urban Area Under Crowdsourcing Monitoring

5 **Authors**: Shan Xu, Bin Zou, Yan Lin, Xiuge Zhao, Shenxin Li, and Chenxia Hu

We would like to take this opportunity to thank the editor and the reviewers for their very careful review and constructive suggestions again. We have revised the manuscript based on the suggestions and provided justifications where appropriate. Given below is a summary of the responses and revisions.

After major revisions I am still not convinced that this paper and the supporting research meet the standards of AMT. I think there are a number of major flaws the authors must address in order to make their paper suitable for publication.

In this revision of the paper I feel the authors have revealed a potential fatal flaw of their study: Page 23 line 12-

15 15: This is very surprising that volunteers just looked at their sensors 3 times in an hour and reported the value to the app. So the concentrations are not hourly averages from each monitor. This makes me question the usefulness of this study. How often does the screen update with the concentration? Is it a 1-minute average? Rolling average of the past hour? Something else? If you pulled 3 random minutes of data from an hour you would expect they might be significantly different than the average for the full hour.

20 Were all the performance metrics calculated in the same way just reading 3 random minutes off the screen? If not the performance metrics (Figure 1 and +/- 20% reported by Tsinghua) are not really good indicators for the expected accuracy of this work. I think the implications of this sampling methodology on the results need to be discussed. See Zheng et al. 2018 for discussion of averaging time and low-cost sensor performance: https://www.atmos-meas-tech.net/11/4823/2018/

25 Author response: the laser air quality monitor has two monitoring modes, Mode One and Mode Two. Under the Mode One, the monitor observes PM$_{2.5}$ concentration in real time and measurement updates and automatically stores every second, but only has a battery life of 5 hours. Under the Mode Two, the monitor repeats the procedure of observing and storing 1 minute and then sleeping 5 minutes and have a battery life of 30 hours. As the number and the battery life of the monitor are limited, we cannot continuously observe PM$_{2.5}$ concentration at one site for

30 the whole sampling period. For all potential sites in one area that a volunteer can cover, 27–36 minutes are needed

to walk through three times. That means 24–33 minutes are left in one hour to observe 2–4 potential sites three times. Therefore, at each potential monitoring site, 3 minutes are designed for $PM_{2.5}$ concentration constantly monitoring every observation. During this process, after the first 60 seconds, we observed the screen and uploaded the measurement using a smart phone application (App) that we developed to verify the reliability of the stored data. For each hour, we eliminated the sampling sites observed less than three times. As the sites take turns measuring $PM_{2.5}$ concentration, there are at least 3 minutes of measurements every 20 minutes of every sampling hour for those left sites. The valid observations for every sampling hour (i.e. 9–12 minutes measurements) were then averaged at each site. During the comparison experiments between laser air quality monitors and the national monitoring instruments, the monitors observed in the Mode Two (i.e. observe 1 minute and then sleep 5 minute). The hourly $PM_{2.5}$ concentrations of the evaluation periods were the mean values of 10 minutes measurements. Although the sampling concentrations are only averages from ~10 minutes measurements, we believe it is generally equal to the hourly averages for the following reasons: (1) the 3–4 times of 3-minutes observation happened at rather regular intervals, which can reflect the temporal variation of $PM_{2.5}$ measurements for one location during one hour to a certain extent. (2) the monitor observed in a similar pattern in the comparison experiments with the national monitoring instruments, and the relatively good agreement the results demonstrate provides empirical evidence for this assumption.

In the last version of manuscript, the sampling methodology was not well described, we thank the reviewer very much for pointing out this issue. We rewrote the sampling methodology and sentences about this were added in the section of Discussion.

My other major concern is the overestimate by the sensors during period 1, and the overestimate of the sensors during period 2. During one of the evaluations periods the RH was 94% at maximum while during one of their sampling campaigns the RH was reported from 95% to 98%. Much previous work has shown that nephelometer type devices greatly over estimate during high RH events and in a nonlinear way (including: https://www.atmos-meas-tech.net/11/4823/2018/). There is no evaluation data at the RH that occurred during this project and they present no figures showing the relationship between RH and sensor error and no past work showing how this sensor performs at high RH. Maybe having accurate sensor readings is not important for the main goal of this paper (generating different models) but they have tried to draw a number of large conclusions based on the higher average from the sensor data (people should stay indoors even when government monitors say it is clean out, pollution at road level is higher). I have a similar concern during the second period where the monitors underestimate since

they have presented no evaluation data and included no references that show that the sensor provides a linear response above 160 ug/m3. Low-cost sensors may show non-linear responses (http://www.aaqr.org/article/detail/AAQR-17-10-OA-0418) and it is very important to evaluate them under the conditions they will be evaluated under. Any supporting literature the authors can find about the monitor they used

5 or the internal sensing component may support their findings but currently the authors have not cited any previous work with their monitor.

Author response: as the reviewer suggested, we analysed the relationship between RH and the ratio of laser monitor measurements to the national instrument measurements for two evaluation periods (December 20–22 and 29–31) (see Supporting Information Figure S1a and S1b). In general, the ratio of laser monitor measurements to the

10 national instrument measurements roughly increased exponentially with the increase in the RH for December 20–22 (RH: 77%–94%) ($R^2$=0.22), while the ratio was uncorrelated with RH for December 29–31 (RH: 38%–67%) ($R^2$=0.05). This is consistent with the findings of Zheng et al. (2018) and Badura et al. (2018), who discovered that low-cost sensors tend to overestimate the $PM_{2.5}$ concentrations when RH is high ($>\sim$80%). When RH correction is made by empirical equation for December 20–22, the $R^2$ between hourly $PM_{2.5}$ concentration from laser monitor

15 measurements and the national instrument measurements improved from 0.89 to 0.9. The agreement between laser monitor and the national instrument was rather good and the improvement after RH correction was insignificant; the potential effect of RH on hourly $PM_{2.5}$ concentration during very high RH events could be consistent because of the low inter-sensor variability (i.e. the measurement difference under the same condition, <5%) of sampling monitors selected from the preliminary experiments and the small spatial variability of RH in intra-urban area. In

20 view of the above-mentioned reasons, we believe the hourly $PM_{2.5}$ concentration for light-polluted sampling period could generally disclose the air pollution variation of different urban microenvironments although the very high RH conditions (RH>95%) were not experienced in the evaluation period. The point of the suggestion about people should stay indoors we intend to make is that it might be sensible for the sensitive groups try to avoid activities in those urban microenvironments with relatively high air pollution exposure risk even when official air pollution

25 levels are "Good" and "Moderate", but still, we agree with the reviewer that the measurements were not accurate enough to support this suggestion and thank the reviewer very much for pointing out this issue. We rewrote the related sentences in the section of Discussion.

Although the monitor experienced little high concentration environments (>160 ug/m$^3$) in the evaluation period, the monitor was compared with the regulatory monitor at concentration levels of 106, 212 and 454 ug/m$^3$ in the

30 test of Tsinghua University and the relative errors were rather low (<20%) and demonstrated similar patterns between concentration levels (Test report, http://www.inovafitness.com/a/minyongchanpin/jianceyilei/2015/0522/

31.html). In consideration of this and the low inter-sensor variability of sampling monitors, we assume the responses of sampling monitor to the national monitoring instrument in heavy-polluted sampling period is consistent in the study area and the spatial variation of air pollution could be revealed to some extent. As the three methods we compared were performed with the same sampling dataset, the uncertainty in measurements associated

5  with the monitor, RH and high concentration environments may cause a limited influence on the method comparison results. We therefore did not correct the measurements in this study. However, we agree with the reviewer that more efforts are needed in the intended use environment evaluating, uncertainty analysis and bias correction of low-cost sensors for applications that requires more accurate measurements in the future, such as very high-resolution mapping of air pollution and accurate exposure assessment, especially under extreme weather

10  conditions and very high and very low concentration environments. Sentences about this were added in the section of Discussion.

Additional comments:

25  Figure 1. It's not clear to me what is on the right scatter plot versus the left? Does each plot show the data from all three stations? How was the data recorded for these tests? Since for the experiment it was read off the screen is this also what was done for these? What is the averaging time for the points shown on this plot. It would be helpful to label them as c and d so you can reference them in the text.

Author response: the left scatter plot is the hourly $PM_{2.5}$ concentrations from the thirty laser air quality monitors

30  and the three national monitoring instruments for December 20–22 (i.e. the first evaluation period with a weather condition similar to that of light-polluted sampling period), the right scatter plot is the hourly $PM_{2.5}$ concentrations

for December 29–31 (i.e. the second evaluation period with a weather condition similar to that of heavy-polluted sampling period). For these tests, the monitors were observed in their Mode Two, namely, the monitor repeats the procedure of observing and storing 1 minute and then sleeping 5 minutes. The hourly $PM_{2.5}$ concentrations of the evaluation periods were the mean values of 10 minutes measurements of each hour. Thank the reviewer for the suggestion. We labelled them as Fig. 1c and Fig. 1d and referenced them in the text.

Page 22 line 4: please define small difference.

Author response: the absolute value of relative error between the observation of one monitor and the average observations of the other monitors no more than 5%.

Abstract line 20: Not sure what all the numbers are in the parenthesis

Author response: mean$\pm$standard deviation of hourly $PM_{2.5}$ concentrations for crowdsourced sampling sites and the national air quality monitoring stations. But after the revision of this manuscript, we found that the numbers are not suitable in the parenthesis, we rewrote the confusing sentence.

Abstract Line 16: suggest mentioning how many hours each period was.

Author response: Thanks for the suggestion. Added.

Abstract line 22: I don't think you can make any generalizations based on this dataset suggest: "During this project", OK interpolation performs… or something similar

Author response: Thanks for the suggestion. Revised.

Page 20 line 24: What do you mean by "fine" exposure control?

Author response: Thanks for the careful review. wrong expression, deleted "fine".

Page 21: Line 9 and 10: suggest including a citation about underestimation/mis calculation of risk

Author response: Thanks for the suggestion. Added.

Page 21: Lines 22-27: These results from Tsinghua are super important since you are reporting data up to 260 ug/m3 but you only have a comparison range up to 160 ug/m3. Please include a citation or a figure in the supplement. Is this an ambient comparison or in the lab?

Author response: the tests from Tsinghua were conducted in the lab. The test report was cited in the revised manuscript.

Page 21 Lines 29-30: Is this conversion factor calculated by you or by the company?

Author response: it was calculated by the company, and the mass concentrations are automatically calculated using a built-in algorithm. Rewrote the confusing sentence.

Page 21 line 25: At what averaging time is this +/- 20%

Author response: hour.

Page 27 line 32: Suggest citing previous literature that has seen significant gradients from 2 to 15m.

Author response: sentence about this and related reference were added.

Author response: The point we intend to make is that it might be sensible for the sensitive groups try to avoid activities in those urban microenvironments with relatively high air pollution exposure risk even when official air pollution levels are "Good" and "Moderate", but still, we agree with the reviewer that the measurements were not accurate enough to support this suggestion and thank the reviewer very much for pointing out this issue. We rewrote the related sentences in the section of Discussion.

**Relevant changes made in the manuscript:**

[revised manuscript text omitted]

---

## Author Response (AR3)

**Author's Response**

**No.**: amt-2018-402     Submitted on 16 Nov 2018

**Title**: Strategies of Method Selection for Fine Scale PM$_{2.5}$ Mapping in Intra-Urban Area Under Crowdsourcing Monitoring

**Authors**: Shan Xu, Bin Zou, Yan Lin, Xiuge Zhao, Shenxin Li, and Chenxia Hu

We are sincerely grateful for the very careful review and constructive suggestions the editor and the reviewer provided. We have revised the manuscript based on the suggestions and provided justifications where appropriate. Given below is a summary of the responses and revisions.

After a second round of major revisions I still think the authors have some flaws and inconsistencies in their discussion that need to be addressed. In the last draft I had 2 major concerns, 1) the sensor "hour averages" are actually a few much shorter time periods averaged and assumed to be equivalent to the 1-hour measurement and 2) that the results from the high RH period may be biased high.

The first concern remains unclear as they now report that 9-12, 1-second sensor measurements were used to calculate a 1-hour average, but this is not described consistently through the paper where initially they are reported as 1 second measurements and then they are referred to as 1-minute measurements for the rest of the paper.

Essentially in this draft the authors claim they will not correct for RH because although it will bias their measurements it should be a uniform bias across the study area. I think that this is acceptable.

Please see my specific comments below.

Major concerns:

In this draft the authors reveal that the sensor has 2 operating modes. During evaluation sensors collected a minute of data and then slept for 5 minutes. During the crowdsourced collection the screen updates concentrations every second and saves 1-second data. This means that a few 1-second measurements were used for the crowdsourcing campaign. The authors have not provided us reason to believe taking 10-minutes of measurements per hour as was done during the evaluation stage would be comparable to 9 seconds of measurements during the crowd sourced campaign.

Author response: the laser monitor has two monitoring modes, Mode One and Mode Two. Under the Mode One, the monitor observes PM$_{2.5}$ concentration in real time and measurement updates and automatically stores every second. This means that 9-12-minutes measurements, a total of 540–720 seconds of measurements were used for

the crowdsourcing campaign for every sampling site per hour. Under the Mode Two, the monitor repeats the procedure of observing and storing every second for 1 minute and then sleeping 5 minutes, that is, 10-minutes of measurements, a total of 600-seconds of measurements per hour during the evaluation stage. Although the hourly sampling concentrations are only averages from 9-12-minutes, a total of 540–720 seconds of measurements, we believe it is generally equal to the hourly averages for the following reasons: (1) the 3–4 times of 3-minutes observation happened at rather regular intervals, which can reflect the temporal variation of $PM_{2.5}$ measurements for one location during one hour to a certain extent. (2) there are 10-minutes, a total of 600-seconds of measurements per hour for each monitor during the comparison experiments with the national monitoring instruments; the numbers of measurements per hour for crowdsourced campaign and comparison experiments are close and averages from the 600-seconds of measurements during the comparison experiments demonstrate a relatively good agreement with the national hourly $PM_{2.5}$ observations ($R^2$: 0.89–0.90), which provides empirical evidence for this assumption. (3) several studies have employed sampling periods of 10-minutes level to measure the within-day variability of PM concentrations (Godri et al., 2010; Griffiths et al., 2018) and few studies have proven that the performance characteristics of some low-cost sensors on 1 min and 1 h scales were rather similar (Zheng et al., 2018). However, it needs to be noted that this assumption was proposed under the circumstance that the weather condition and emission sources of the study area demonstrate no extreme variation in short-term. It is inapplicable to some special cases such as a dust storm when the $PM_{2.5}$ can increase tens and even hundreds of times high concentration within a few minutes (Zhang et al., 2010).

We thank the editor and the reviewer for pointing out the inadequacy of the explanation and discussion about the measurements between evaluation and crowdsourcing stage. We rewrote some confusing sentences and discussion about this issue was added as well.

Given below are the screenshots of the link.

[Figure]

[Figure]

Figure S1b: What is X and Y in the equation? A quick check shows that the equation printed on the plot is not the equation printed (y=-.001x+1.508 at 40% y should be 1.45 and it is <1.1)

5  Author response: X is the relative humidity and Y is the ratio of hourly PM$_{2.5}$ concentration of laser monitor to national instrument. We apologize for making such a mistake during the composition of figures, we corrected the Figure S1, and sentences about the figure were added. The careful work of the reviewer is greatly appreciated.

Given below are the screenshots of the relationships between relative humidity and the ratio of hourly PM$_{2.5}$ concentration of laser monitor to national instrument for December 20-22 and December 29-31.

[Figure]

Page 12 line 18: You say 3 minutes but on page 10 line 20-21 you say that the measurement updates and automatically stores every second. So, this means they are recording 3 one-second readings not 3 one-minute readings.

Author response: under the Mode One, the monitor observes $PM_{2.5}$ concentration in real time and measurement updates and automatically stores every second. This means that there is a total of 180 one-second of measurements if we continuously observe $PM_{2.5}$ for 3-minutes in Mode One.

Page 17 line 11: Again, this contradicts page 10 line 20-21.

Author response: rewrote the confusing sentences.

Page 17 line 14: But if one is second measurements and one is minute measurements this may not be a good comparison

Author response: rewrote the confusing sentences.

Editorial comments:

Page 17 line 7: should be monitors (not monitor)

Author response: corrected.

Page 18 line 2: Suggest using lightly polluted instead of light polluted so it is clear you aren't talking about light pollution from lights or the sun.

Author response: thanks for the suggestion. Corrected.

**Relevant changes made in the manuscript:**

[revised manuscript text omitted]

---

## Author Response (AR4)

**Author's Response**

**No.**: amt-2018-402     Submitted on 16 Nov 2018

**Title**: Strategies of Method Selection for Fine Scale PM$_{2.5}$ Mapping in Intra-Urban Area Under Crowdsourcing Monitoring

**Authors**: Shan Xu, Bin Zou, Yan Lin, Xiuge Zhao, Shenxin Li, and Chenxia Hu

We would like to take this opportunity to thank the editors for their very careful work. We have checked the manuscript and prepared the files according to the submission guidelines. All files shall be uploaded as required. We certify that (a) Research is original and has not been published elsewhere; (b) Presented material is that of the authors'; (c) Research is conducted with scientific integrity and known norms of science; and (d) there is not any duplication. If you need further information, please feel free to contact us at 210010@csu.edu.cn.

Thank you again for your consideration of this manuscript.

Warmest regards,

Bin Zou, Ph.D

Professor of GIS applied urban environment simulation and analysis

Mobile phone: +86(0) 18607312100

Email: 210010@csu.edu.cn

Address: No. 932, South Lushan Road, Yuelu District, Changsha City, China, 410086

cc: co-authors